# STABLE AND SCALABLE DEEP PREDICTIVE CODING NETWORKS WITH META-PREDICTION ERRORS

**Myoung Hoon Ha[1]**  **Hyunjun Kim[2],†**  **Yoondo Sung[3],†**
**Youngha Jo[4],‡**  **Min S. Kang[5],‡**  **Sang Wan Lee[1,3,6,7],\***
[1]Center for Neuroscience-inspired AI, KAIST    [2]Dept. of ECE, Seoul National University
[3]Dept. of Bio and Brain Engineering, KAIST    [4]LG CNS
[5]School of Electrical Engineering, KAIST    [6]Brain and Cognitive Sciences, KAIST
[7]Kim Jaechul Graduate School of AI, KAIST
{mh.ha.soar, youngha908}@gmail.com  doctor3390@snu.ac.kr
{ydsung, minseok21, sangwan}@kaist.ac.kr

## ABSTRACT

Predictive Coding Networks (PCNs) offer a biologically inspired alternative to conventional deep neural networks. However, their scalability is hindered by severe training instabilities that intensify with network depth. Through dynamical mean-field analyses, we identify two fundamental pathologies that impede deep PCN training: (1) prediction error (PE) imbalance that leads to uneven learning across layers, characterized by error concentration at network boundaries; and (2) exploding and vanishing prediction errors (EVPE) sensitive to weight variance. To address these challenges, we propose Meta-PCN, a unified framework that incorporates two synergistic components: (1) a loss based on meta-prediction error, which minimizes PEs of PEs to linearize the nonlinear inference dynamics; and (2) weight regularization that employs normalization to regulate weight variance and mitigate EVPE. Extensive experimental validation on CIFAR-10/100 and TinyImageNet demonstrates that Meta-PCN achieves statistically significant improvements over conventional PCNs, outperforming backpropagation in most tested configurations, while preserving the local learning rules of PCNs.

## 1 INTRODUCTION

Predictive coding (PC) represents a theoretical framework for understanding cortical information processing. It encompasses fundamental functions such as learning, prediction, encoding, and memorization. As neural architectures, PCNs implement this framework and offer a compelling alternative to backpropagation-based learning. PCNs are grounded in PC theory (Srinivasan et al., 1982; Mumford, 1992; Rao & Ballard, 1999; Friston, 2005) and formalized through the free-energy framework (Friston, 2010; Bogacz, 2017; Bastos et al., 2012). They employ purely local learning rules that respect biological constraints while enabling massive parallelization (Millidge et al., 2022b; Salvatori et al., 2022; Song et al., 2020). This positions them as promising candidates for neuromorphic computing (Schuman et al., 2017; Sacramento et al., 2018). However, PCNs face a critical limitation. As network depth increases, their training becomes progressively unstable. This creates a formidable barrier to scalability (Millidge et al., 2022b). The underlying mechanisms driving this instability have remained poorly understood, hindering the practical deployment of deep PCNs in complex applications.

To address these fundamental challenges, we conduct a rigorous analysis of PCN inference dynamics using dynamical mean-field theory (DMFT) (Sompolinsky et al., 1988; Poole et al., 2016; Schoenholz et al., 2017). Our theoretical investigation (detailed in Section 3) reveals two distinct yet interconnected pathologies that impede deep PCN scalability:
**(1) PE Imbalance** (Pinchetti et al., 2024; Qi et al., 2025; Goemaere et al., 2025; Innocenti et al., 2025): Errors concentrate in boundary layers (input/output) while vanishing in intermediate

---

*Corresponding author.   †Equal second-author contribution.   ‡Equal contribution.

layers. This creates a characteristic imbalanced distribution. This results in gradient starvation in mid-layers and prevents effective learning.

**(2) EVPE**: We identify exponential growth and decay patterns in latent states and PEs during inference. These dynamics are controlled by temporal scaling factors that depend critically on weight variance. This leads to training instabilities analogous to classical exploding and vanishing gradients (Bengio et al., 1994; Hochreiter, 1998; Pascanu et al., 2012; Arjovsky et al., 2016), though arising during inference rather than backpropagation.

To systematically address these pathologies, we propose Meta-PCN (Section 4). This unified framework incorporates two complementary solutions operating synergistically. First, we introduce a novel objective based on meta-prediction error (meta-PE). This objective linearizes the nonlinear equilibrium system by minimizing PEs of PEs. Second, we implement a normalization of weight variance. This controls variance and suppresses exponential behaviors. Through these two complementary solutions, our framework enables stable training of deep PCNs. We achieve substantial improvements in inference stability, convergence speed, and classification performance. These improvements are obtained while preserving the local learning rules that underpin PCNs' relevance to neuromorphic applications (see Appendix A for a discussion of biological plausibility).

Extensive experimental validation on CIFAR-10, CIFAR-100, and TinyImageNet demonstrates that Meta-PCN achieves substantial performance improvements, with 12–79% gains over conventional PCNs across all tested architectures (Figure 7). Notably, Meta-PCN outperforms backpropagation in 29 out of 30 configurations by an average of 2.15% (22 statistically significant; see Appendix H), while adhering to local learning constraints. These results suggest that Meta-PCN is a viable framework for scaling PC to deeper architectures while preserving the local learning rules relevant to neuromorphic computing.

## 2 PREDICTIVE CODING NETWORKS

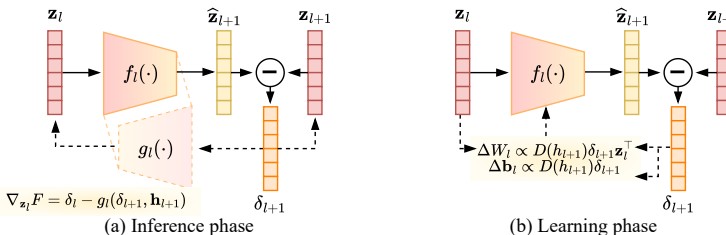

(a) Inference phase      (b) Learning phase

Figure 1: Inference and learning phases of local PC modules. (a) Inference phase: the PEs ($\boldsymbol{\delta}_{l+1}$) are calculated and the latent states ($\mathbf{z}_l$) are updated. This process is repeated until it reaches the final inference step $T$. (b) Learning phase: the weight and bias parameters ($W_l$ and $\mathbf{b}_l$) are updated.

**PCN Architecture and Objective.** PC proposes that the brain continuously generates predictions of the external environment and refines internal representations by minimizing PEs—a computational mechanism that extends beyond perception to reward-guided decision-making (Lee & Seymour, 2019; Lee et al., 2019). PCNs implement this principle by connecting local PC modules in a hierarchical chain structure, as illustrated in Figure 1. The forward pass generates predictions for subsequent layers, while the backward pass minimizes local PEs (Whittington & Bogacz, 2017; Millidge et al., 2022a). Each layer $l$ with the latent state $\mathbf{z}_l \in \mathbb{R}^{N_l}$ produces predictions via $\hat{\mathbf{z}}_{l+1} = f_l(\mathbf{z}_l) = \phi(W_l \mathbf{z}_l + \mathbf{b}_l)$, where $f_l(\cdot)$ represents the forward prediction function, $W_l$ denotes the weight matrix, $\mathbf{b}_l$ is the bias vector, and $\phi(\cdot)$ is the activation function. The main goal is to minimize PEs for each layer, $\boldsymbol{\delta}_l = \mathbf{z}_l - \hat{\mathbf{z}}_l$, quantified using the notion of free energy:

$$\mathcal{F} = \frac{1}{2} \sum_{l=2}^{L} \|\boldsymbol{\delta}_l\|_2^2.$$

This optimization is subject to the boundary conditions $\mathbf{z}_1 = \mathbf{x}$ (input) and $\mathbf{z}_L = \mathbf{y}$ (target).

**Inference and Learning Dynamics.** PCNs employ a dual optimization process that alternates between inference and learning phases. During the inference phase, latent states evolve according to the fixed-point iteration: $\mathbf{z}^{t+1} = \mathbf{z}^t - \eta \cdot \nabla_{\mathbf{z}} \mathcal{F}(\mathbf{z}^t)$, where the global latent state vector $\mathbf{z}$ is the concatenation of all layer-wise latent states and $\eta$ denotes the inference rate. The gradient computation

involves forward and backward operations that facilitate bidirectional information flow throughout the network. The forward prediction function $f_l(\mathbf{z}_l) = \phi(W_l\mathbf{z}_l + \mathbf{b}_l)$ propagates information from lower to higher layers, while the backward operation $g_l(\boldsymbol{\delta}_{l+1}, \mathbf{h}_{l+1}) = W_l^\top D(\mathbf{h}_{l+1})\boldsymbol{\delta}_{l+1}$ transmits error signals from higher to lower layers, where $\mathbf{h}_{l+1} = W_l\mathbf{z}_l + \mathbf{b}_l$ denotes the pre-activation and $D(\mathbf{h}_{l+1}) = \mathrm{diag}(\phi'(\mathbf{h}_{l+1}))$ represents the diagonal matrix of activation derivatives. The gradient naturally decomposes into bottom-up error and top-down feedback components: $\nabla_{\mathbf{z}_l}\mathcal{F} = \boldsymbol{\delta}_l - g_l(\boldsymbol{\delta}_{l+1}, \mathbf{h}_{l+1})$. During the learning phase, parameters are updated according to $\nabla_{W_l}\mathcal{F} = -D(\mathbf{h}_{l+1})\boldsymbol{\delta}_{l+1}\mathbf{z}_l^\top$ and $\nabla_{\mathbf{b}_l}\mathcal{F} = -D(\mathbf{h}_{l+1})\boldsymbol{\delta}_{l+1}$.

**Challenges in Achieving Equilibrium.** The equilibrium condition $\nabla_{\mathbf{z}}\mathcal{F} = 0$ yields $\boldsymbol{\delta}_l = g_l(\boldsymbol{\delta}_{l+1}, \mathbf{h}_{l+1})$, which resembles a standard error backpropagation algorithm (Whittington & Bogacz, 2017; Millidge et al., 2022a). However, achieving equilibrium during inference presents substantial computational challenges. Equivalence to backpropagation holds only under specific conditions, such as downweighting the output-layer error term to zero (Whittington & Bogacz, 2017). The complex interplay between inference and learning dynamics makes understanding deep PCN behavior particularly challenging, motivating systematic analyses of deep PCN pathologies (Section 3).

## 3 DEEP PCN INSTABILITY: UNCOVERING FUNDAMENTAL PATHOLOGIES

This section provides a comprehensive investigation into the internal inference dynamics in deep PCNs. We employ DMFT to mathematically characterize the underlying mechanisms driving these instabilities (Section 3.1). Our analysis reveals that deep PCNs suffer from two fundamental pathologies that severely impede practical scalability. These pathologies are PE imbalance (Section 3.2) and EVPE (Section 3.3).

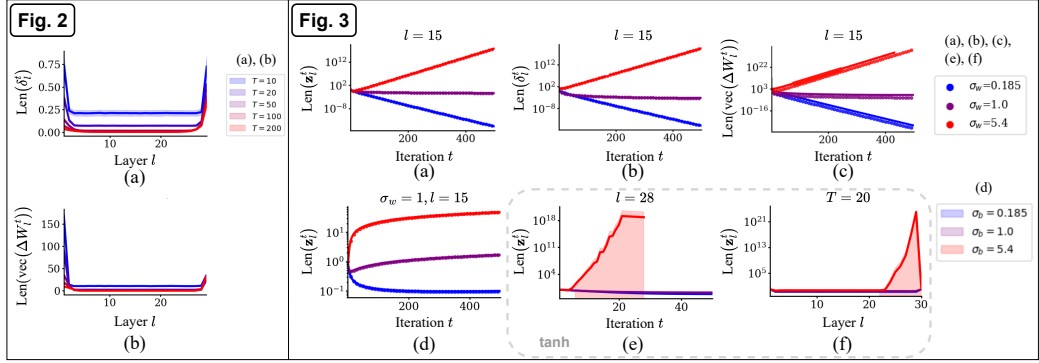

Figure 2: The layer-wise distributions of lengths in PCNs: (a) $\mathrm{len}(\boldsymbol{\delta}_l^t)$, and (b) $\mathrm{len}(\mathrm{vec}(\Delta W_l^t))$. *Setting:* A randomly initialized linear PCN with Gaussian-sampled inputs, outputs, and initial latent states ($\mathbf{z}_l^0 \sim \mathcal{N}(\mathbf{0}, I)$); no training is involved, purely for analyzing inference dynamics (see Appendices C.1 and E for details). We set $L = 30$, the terminal inference step is $T = 200$, and the latent dimension of each layer is set to 100. The inference rate is set to $\eta = 0.05$. $\sigma_w$ and $\sigma_b$ are set to 1 and 0.1, respectively.

Figure 3: Dynamics of $\mathrm{len}(\mathbf{z}_l^t)$, $\mathrm{len}(\boldsymbol{\delta}_l^t)$, and $\mathrm{len}(\mathrm{vec}(\Delta W_l^t))$. Dotted lines represent theoretical predictions, while solid lines correspond to empirical observations from linear PCN experiments. *Subfigures:* Dynamics of (a) $\mathrm{len}(\mathbf{z}_l^t)$, (b) $\mathrm{len}(\boldsymbol{\delta}_l^t)$, and (c) $\mathrm{len}(\mathrm{vec}(\Delta W_l^t))$ of linear PCNs across $\sigma_w \in \{0.185, 1.0, 5.4\}$. (d) Dynamics of $\mathrm{len}(\mathbf{z}_l^t)$ across $\sigma_b \in \{0.185, 1.0, 5.4\}$. (e) $\mathrm{len}(\mathbf{z}_l^t)$ of nonlinear (tanh) PCNs across $\sigma_w \in \{0.185, 1.0, 5.4\}$. (f) Layer-wise distributions of $\mathrm{len}(\mathbf{z}_l^t)$ of nonlinear (tanh) PCNs at $t = 20$. Further analysis for nonlinear cases can be found in Appendix K. *Setting:* In (a), (b), (c), and (d), $l = 15$, while in (e), $l = 28$. The other settings are the same as in Figure 2.

### 3.1 ANALYTICAL METHODOLOGY: DYNAMICAL MEAN-FIELD THEORY APPROACH

**Motivation.** PCNs present unique analytical challenges due to their dynamically evolving latent states during the inference phase. This makes it difficult to characterize these dynamics using conventional analytical methodologies. To address this challenge, we develop an analytical framework

based on the DMFT approach (Poole et al., 2016; Schoenholz et al., 2017). This framework provides mathematical tractability for the complex inference dynamics of PCNs.

**Length-based Statistical Framework.** Following the methodology established by Poole et al. (2016), we construct a statistical framework for analyzing PCN dynamics through length-based measures. We define the length of a vector $\mathbf{x}$ as the mean squared element: $\text{len}(\mathbf{x}) = \langle x_i^2 \rangle = \frac{1}{N} \sum_{i=1}^{N} x_i^2$, where $x_i$ denotes the $i$-th element of the vector $\mathbf{x}$, and $N$ represents the vector dimension. In this analysis, weights and biases are assumed to be drawn i.i.d. as $w_{i,j}^l \sim \mathcal{N}(0, \frac{\sigma_w^2}{N_l})$ and $b_i^l \sim \mathcal{N}(0, \sigma_b^2)$. Our analysis systematically tracks three fundamental length variables. These are the lengths of latent states $\text{len}(\mathbf{z}_l^t)$, the lengths of PEs $\text{len}(\boldsymbol{\delta}_l^t)$, and the lengths of weight updates $\text{len}(\text{vec}(\Delta W_l^t))$ at layer $l$ and time step $t$, where $\text{vec}(\cdot)$ denotes the vectorization operator that transforms matrix columns into a single vector and the weight update $\text{vec}(\Delta W_l^t) \propto \nabla_{\text{vec}(W_l)} \mathcal{F}$. To mathematically characterize PCN dynamics, we introduce three interaction matrices that capture the intricate relationships between network components. The Latent Self-Interaction Matrix $P^t$ captures inter-layer latent state interactions through $P_{l+k,l}^t = \frac{1}{N} \mathbf{z}_{l+k}^t{}^\top M_{l+k-1:l} \mathbf{z}_l^t$. Here $M_{l+k-1:l} = M_{l+k-1} M_{l+k-2} \cdots M_l$ represents the product of normalized weight matrices and $M_l = \frac{1}{\sigma_w} W_l$ denotes the normalized weight matrix. The framework additionally incorporates bias-latent interactions via $B_{l,l-k}^t = \frac{1}{N} \mathbf{b}_{l-1}^\top M_{l:l-k} \mathbf{z}_{l-k}^t$. Bias self-interactions are characterized through the constant matrix $\Gamma = \sigma_b^2 I$. The interaction matrices enable the systematic derivation of length dynamics for each network component. The temporal evolution is governed by the interaction matrices themselves, which evolve according to the dynamics of the latent state updates: $\mathbf{z}_l^{t+1} = \nu M_{l-1} \mathbf{z}_{l-1}^t + \kappa \mathbf{z}_l^t + \nu M_l^\top \mathbf{z}_{l+1}^t + \eta \mathbf{b}_{l-1} - \nu M_l^\top \mathbf{b}_l$. Here, $\nu = \eta \sigma_w$ governs the information propagation rate, and $\kappa = 1 - \eta(1 + \sigma_w^2)$ controls self-interaction dynamics. The latent state length, $\text{len}(\mathbf{z}_l^t) = P_{l,l}^t$, directly corresponds to the diagonal elements of the inter-layer latent state interactions matrix $P$. The PE length emerges from the interaction between these components according to: $\text{len}(\boldsymbol{\delta}_l^t) = \mathbf{c}_q^\top P_{l-1:l,l-1:l}^t \mathbf{c}_q - 2 B_{l,l-1:l}^t \mathbf{c}_q + \Gamma_{l-1,l-1}$, where $\mathbf{c}_q = [-\sigma_w, 1]^\top$. The weight update length follows directly as $\text{len}(\text{vec}(\Delta W_l^t)) = \text{len}(\boldsymbol{\delta}_{l+1}^t) \cdot \text{len}(\mathbf{z}_l^t)$. The complete mathematical derivations are presented in Appendix C, including the evolution equations along with a detailed analysis of length dynamics. We emphasize that these results are derived under simplifying assumptions: i.i.d. Gaussian weights and biases, linear (or linearized) forward and backward maps, uniform layer widths, and the large-width limit. These assumptions do not perfectly match the deep, nonlinear convolutional architectures used in our experiments (Section 6), leaving an inherent gap between theory and practice.

## 3.2 PROBLEM 1: IMBALANCED PREDICTION ERRORS

**Empirical Observations.** The first fundamental pathology of deep PCNs manifests as PE imbalance across network layers. As shown in Figure 2, PEs exhibit a characteristic imbalanced distribution. PEs decrease substantially in intermediate layers during inference. This results in significant error concentration at the boundary layers, while errors vanish in intermediate layers. This phenomenon arises from the fundamental constraints on information propagation in PCNs. Information propagates from $k$ layers away at a rate of $\mathcal{O}(\nu^k)$, where $\nu = \eta \sigma_w$, as established in our length-based framework. With typical values of $\nu \leq 1.0$ (varying with inference rate and weight variance), this exponential decay causes inference to terminate prematurely before information effectively reaches the middle layers.

**Theoretical Analysis: Imbalanced Error Distribution and Gradient Starvation.** The mathematical foundation of this imbalance can be understood through the effects of boundary conditions and the dynamics of error propagation. With the input layer clamped as $\mathbf{z}_1 = \mathbf{x}$ during inference, the PE $\boldsymbol{\delta}_2 = \mathbf{z}_2 - \hat{\mathbf{z}}_2 = \mathbf{z}_2 - \phi(W_1 \mathbf{x} + \mathbf{b}_1)$ contains a forcing term. This term continuously reintroduces residual errors near the input boundary. Similarly, clamping the output layer as $\mathbf{z}_L = \mathbf{y}$ yields $\boldsymbol{\delta}_L = \mathbf{y} - \phi(W_{L-1} \mathbf{z}_{L-1} + \mathbf{b}_{L-1})$. This typically remains nonzero and acts as a persistent error source at the output boundary. The equilibrium condition $\boldsymbol{\delta}_l = W_l^\top D(\mathbf{h}_{l+1}) \boldsymbol{\delta}_{l+1}$ implies the spectral norm bound $\|\boldsymbol{\delta}_l\| \leq \|W_l\|_2 \|\phi'(\mathbf{h}_{l+1})\|_\infty \|\boldsymbol{\delta}_{l+1}\|$. The total bound across layers yields the product bound $\|\boldsymbol{\delta}_l\| \leq \left( \prod_{j=l}^{L-1} \|W_j\|_2 \|\phi'(\mathbf{h}_{j+1})\|_\infty \right) \|\boldsymbol{\delta}_L\|$. When the terms in this product are less than unity, PEs decay geometrically as they propagate downward from the output boundary.

Combined with the term fixed at the input boundary, this creates the characteristic U-shaped error profile.

**The PE Dilemma: Error Minimization Impedes Training.** As PEs distribute unevenly across layers, certain layers experience $\boldsymbol{\delta}_{l+1} \approx 0$. Meanwhile, the per-layer weight derivatives follow $\nabla_{W_l}\mathcal{F} = -D(\mathbf{h}_{l+1})\boldsymbol{\delta}_{l+1}\mathbf{z}_l^\top$ for $2 \leq l \leq L-1$. Consequently, when $\boldsymbol{\delta}_{l+1} \approx 0$, the gradient $\nabla_{W_l}\mathcal{F} \approx 0$ and parameter updates vanish, leading to gradient starvation (Pezeshki et al., 2021). Thus, near-zero PEs at layer $l+1$ directly eliminate learning signals for weights $W_l$, compromising the learning capacity of deep PCNs. This presents a fundamental paradox. While PEs constitute the primary objective to minimize, reducing them to near-zero in any layer disrupts learning signal propagation. This reveals that learning signals are transmitted through $\boldsymbol{\delta}$ values, and their elimination blocks this critical information flow. The equilibrium condition's delta relationship $(\boldsymbol{\delta}_l = g_l(\boldsymbol{\delta}_{l+1}, \mathbf{h}_{l+1}))$ indicates not PE shrinkage, but rather the necessity of maintaining learning signal transmission through $\boldsymbol{\delta}$ terms. This insight underpins our meta-PE-based loss design (Section 4.1), which provides a principled solution to the inherent trade-off between minimizing error and preserving the learning signal.

### 3.3 PROBLEM 2: EXPLODING AND VANISHING PREDICTION ERRORS

**Characterization.** EVPE represents a phenomenon distinct from the classical exploding or vanishing parameter gradients observed in backpropagation (Bengio et al., 1994; Hochreiter, 1998; Pascanu et al., 2012; Arjovsky et al., 2016). Our DMFT analysis, corroborated by empirical measurements on deep PCNs (Figure 3), reveals multiplicative scaling patterns that manifest across inference iteration $t$. Specifically, we observe a scaling relationship of the form $\|\boldsymbol{\delta}_l^{t+1}\| \approx \tau_t(\sigma_w)\|\boldsymbol{\delta}_l^t\|$. The multiplicative factors $\tau_t(\sigma_w)$ are governed primarily by the weight variance $\sigma_w^2$ and the effective gain $\|\phi'(\mathbf{h})\|_\infty$. When $\tau_t > 1$, the corresponding quantities experience geometric growth; conversely, when $\tau_t < 1$, geometric decay occurs. Critically, stable dynamics (factors approaching unity) emerge only within a narrow interval of $\sigma_w$, specifically near one. This stable region contracts as network depth increases, making proper initialization exponentially more difficult for deeper architectures. Although nonlinear activations such as tanh and ReLU impose the constraint $\|\phi'\|_\infty \leq 1$, thereby attenuating these multiplicative effects, they do not eliminate the underlying geometric scaling behavior.

**Distinction from Classical Exploding/Vanishing Gradients.** The implications of EVPE extend beyond classical gradient pathologies due to the unique structure of PC weight updates. Since PC parameter updates follow $\nabla_{W_l}\mathcal{F} = -D(\mathbf{h}_{l+1})\boldsymbol{\delta}_{l+1}\mathbf{z}_l^\top$, we have the proportionality relationship: $\|\text{vec}(\Delta W_l^t)\| \propto \|\boldsymbol{\delta}_{l+1}^t\|\|\mathbf{z}_l^t\|\|\phi'(\mathbf{h}_{l+1}^t)\|_\infty$. This direct coupling implies that the exploding and vanishing of latent states and PEs are immediately reflected in the magnitude of the parameter updates. Notably, this instability arises during the inference phase itself, before any parameter updates—a fundamental distinction from traditional analyses that localize explosion or vanishing behavior to backpropagated loss gradients alone. In practical terms, large values of $\tau_t$ result in rapidly increasing magnitudes $\|\mathbf{z}_l^t\|$ or $\|\boldsymbol{\delta}_l^t\|$, thereby inducing proportionally larger updates that can destabilize training. Conversely, small multiplicative factors yield near-zero updates, leading to training stagnation and ineffective learning.

## 4 META-PCN: A UNIFIED FRAMEWORK FOR DEEP PCN STABILIZATION

Meta-PCN represents a comprehensive framework that addresses the two fundamental pathologies identified in Section 3 through a synergistic combination of complementary techniques. Our approach systematically targets each instability mechanism with tailored solutions that work in concert to enable stable deep PCN training. The framework comprises two core components that address the identified pathologies as follows:

• **Addressing PE Imbalance (Problem 1):** We employ a dual approach that combines (i) the meta-PE objective to linearize the nonlinear equilibrium system (Section 4.1) and (ii) systematic weight regularization to control error propagation patterns (Section 4.2).
• **Mitigating EVPE (Problem 2):** We implement comprehensive weight regularization strategies that control the multiplicative scaling factors $\tau_t(\sigma_w)$, thereby preventing geometric growth and decay patterns during inference (Section 4.2).

Each solution component is designed to preserve local learning properties, while systematically addressing the mathematical and computational challenges that have hindered scalability to deep architectures.

## 4.1 Inference as Meta-Prediction Error Minimization

**Motivation.** Conventional PC objectives suffer from fundamental structural limitations. First, as demonstrated in Section 3.2, the free-energy formulation creates a paradoxical situation where minimizing PEs leads to gradient starvation, eliminating learning signals and blocking their propagation through the network. Second, a critical train-test mismatch emerges in practice: while model predictions at evaluation time rely on direct feed-forward computation without inference, training depends entirely on latent state updates through the iterative inference process. We propose a novel objective that addresses these fundamental issues while preserving core PC principles.

**Loss Based on Meta-Prediction Errors.** Our approach redesigns the PCN objective to transform the underlying inference dynamics. While conventional PCNs initialize latent states with feed-forward predictions, we fix these predictions during inference: $\hat{\mathbf{z}}_l^{(t)} = f_{l-1}(\hat{\mathbf{z}}_{l-1}^{(0)}) = \mathbf{c}_l$, where $\mathbf{c}_l :=$ $\phi(\mathbf{h}_l^{(0)})$ and $\mathbf{h}_l^{(0)}$ represents the initial pre-activation. Note that freezing feed-forward predictions is inherently tied to the meta-objective; see Appendix I.4 for clarification and additional ablations. This modification effectively linearizes the nonlinear stationarity system $F(\mathbf{z}) = \nabla_{\mathbf{z}}\mathcal{F}(\mathbf{z}) = 0$ around the feed-forward initialization point. By introducing error $\tilde{\boldsymbol{\delta}}_l := \mathbf{z}_l - \mathbf{c}_l$, we obtain a layer-wise linear surrogate:

$$\tilde{F}_l(\mathbf{z}) = \tilde{\boldsymbol{\delta}}_l - g_l(\tilde{\boldsymbol{\delta}}_{l+1}, \mathbf{h}_{l+1}^{(0)}) = (\mathbf{z}_l - \mathbf{c}_l) - W_l^\top D(\mathbf{h}_{l+1}^{(0)})(\mathbf{z}_{l+1} - \mathbf{c}_{l+1}),$$

for $2 \leq l \leq L - 1$, establishing a linear fixed-point relationship between consecutive layer errors. Importantly, this represents a linearization of the equilibrium map $F(\mathbf{z})$. The Meta-PCN equilibrium approximates the standard PCN equilibrium only when inference trajectories remain near the feed-forward initialization; the two do not coincide in general. Building on this linearization, we define a loss function:

$$\mathcal{J}(\tilde{\boldsymbol{\delta}}) = \frac{1}{2} \sum_{l=2}^{L-1} \|\tilde{\boldsymbol{\delta}}_l - g_l(\tilde{\boldsymbol{\delta}}_{l+1}^*, \mathbf{h}_{l+1}^{(0)})\|_2^2.$$

Since PEs propagate top-down under feed-forward initialization, we treat the top-down transmitted signal $\tilde{\boldsymbol{\delta}}_{l+1}^*$ as a stabilized error. Conceptually, $\tilde{\boldsymbol{\delta}}_{l+1}^*$ denotes the top-down error that would be present once inference converges to a fixed point. In practice, we approximate it using the current estimate $\tilde{\boldsymbol{\delta}}_{l+1}^{(t)}$, analogous to bootstrapping in temporal-difference learning. The conceptual innovation lies in treating $g_l(\cdot)$ as a function that predicts the model's feed-forward PE $\tilde{\boldsymbol{\delta}}_l$ using the stabilized error signals as input. Therefore, $\mathcal{J}$ minimizes PEs of PEs—a meta-level objective that transforms the learning dynamics. Since $\partial\tilde{\boldsymbol{\delta}}_l/\partial\mathbf{z}_l = I$ and $\partial\tilde{\boldsymbol{\delta}}_{l+1}/\partial\mathbf{z}_{l+1} = I$, the gradient of $\mathcal{J}$ with respect to $\mathbf{z}_l$ equals the linearized stationarity map: $\nabla_{\mathbf{z}_l}\mathcal{J} = \tilde{\boldsymbol{\delta}}_l - g_l(\tilde{\boldsymbol{\delta}}_{l+1}^*, \mathbf{h}_{l+1}^{(0)})$. Consequently, minimizing $\mathcal{J}$ drives the linearized equilibrium residual to zero, providing a tractable linear surrogate for the original nonlinear PC inference problem. This approach directly addresses the motivational problems. First, rather than directly minimizing PEs and causing gradient starvation, this objective encourages PEs to follow the delta relationship, ensuring balanced error propagation across layers in conjunction with weight regularization. Second, by fixing feed-forward predictions, we mitigate the train-test mismatch while transmitting learning signals via PEs. Importantly, iterative inference over latent states $\mathbf{z}_l$ is preserved: only the predictions $\mathbf{c}_l$ are frozen, while the latent states continue to evolve through $\tilde{\boldsymbol{\delta}}_l = \mathbf{z}_l - \mathbf{c}_l$.

**Mitigation of PE Imbalance and Enhanced Convergence.** Our analysis in Section 3 demonstrates that layer-wise PEs satisfy the spectral bound $\|\boldsymbol{\delta}_l\| \leq \|W_l\|_2 \|\phi'(\mathbf{h}_{l+1})\|_\infty \|\boldsymbol{\delta}_{l+1}\|$. When these multiplicative factors are less than unity, the resulting product bound yields geometric decay from the output boundary, which—combined with the anchored input forcing—creates the characteristic U-shaped error profile. The meta-PE objective addresses this imbalance through its dependence on the operator $W_l^\top D(\mathbf{h}_{l+1}^{(0)})$. By regulating the scale of this operator, we directly control the amplification and attenuation factors in the spectral bound. Furthermore, the transformation from

solving a nonlinear system to a linear surrogate provides substantial convergence improvements. The dynamics exhibit more predictable and stable behavior compared to the original nonlinear inference, enabling more efficient convergence to surrogate equilibrium states while preserving the essential PC principles.

**Parameter Learning.** While the meta-PE objective $\mathcal{J}$ governs the inference dynamics over latent states, parameter updates follow the conventional PC loss: $\mathcal{L}(\theta) = \frac{1}{2} \sum_{l=2}^{L-1} \|\mathbf{z}_l^{(T)} - f_{l-1}(\mathbf{z}_{l-1}^{(0)}; \theta)\|_2^2.$* This separation allows the meta-objective to stabilize inference while preserving the standard PC learning rule for parameters (see Appendix F for details).

## 4.2 WEIGHT REGULARIZATION

To address the identified pathologies, Meta-PCN employs a variance-based normalization strategy that provides computationally efficient spectral control while maintaining stable weight distributions across network layers. We introduce a practical alternative to direct spectral norm computation that leverages the relationship between weight variance and spectral properties. For a weight matrix $W$ with dimensions $(m, n)$ and variance $\sigma_w^2 = \text{Var}(W)$, we apply direct normalization:

$$W \leftarrow \frac{W}{(\sqrt{m} + \sqrt{n})\sigma_w}.$$

This approach draws upon random matrix theory. For weight matrices with i.i.d. entries having zero mean and variance $\sigma_w^2$, the spectral norm satisfies $\|W\|_2 \approx (\sqrt{m} + \sqrt{n})\sigma_w$. Our variance-based scaling ensures $\|W_{\text{normalized}}\|_2 \approx 1$, serving as a computationally efficient proxy for spectral control. The theoretical foundation rests on the conservative upper bound $\|W\|_2 \leq \|W\|_F \approx \sqrt{mn}\sigma_w$ and the random matrix theory result $\|W\|_2 \approx (\sqrt{m} + \sqrt{n})\sigma_w$, enabling accurate spectral norm approximation. For implementation, we determine operator dimensions as follows: linear layers use $m = d_{\text{out}}, n = d_{\text{in}}$; convolutional layers employ $m = C_{\text{out}}, n = C_{\text{in}} \cdot k_H \cdot k_W$. This approach offers several advantages: low computational cost with parallel computation, no additional parameters, uniform application across layer types, and robust spectral control by maintaining $\|W\|_2$ near unity. We note that the i.i.d. assumption does not fully capture the structured nature of convolutional filters or the evolving weight distributions during training; a complete theoretical treatment for such structured operators remains an important direction for future work. The variance normalization framework directly addresses the exponential scaling behaviors identified in our analysis while preserving computational efficiency. By regulating $\sigma_w^2$ to maintain spectral norms near unity, this approach simultaneously mitigates both EVPE and PE imbalance pathologies, providing an effective solution for deep architectures. The complete training procedure is summarized in Algorithm 1 (Appendix D).

## 5 META-PCN RESOLVES TWO FUNDAMENTAL PATHOLOGIES

This section demonstrates that Meta-PCN successfully addresses the two fundamental pathologies identified in Section 3. The experiments are conducted under identical conditions to enable direct comparison with the problematic behaviors observed in conventional PCNs (see Appendix E for experimental details).

**Balanced PEs.** We replicate the PE imbalance analysis from Section 3.2 using the Meta-PCN framework. Figure 4 shows the layer-wise PE distribution under Meta-PCN. Unlike the characteristic U-shaped error concentration observed in Figure 2, Meta-PCN exhibits well-balanced error distributions across all network layers. PEs maintain relatively uniform magnitudes across layers without excessive concentration at boundary layers. Note that since the output layer is clamped ($\mathbf{z}_L = \mathbf{y}$), the boundary error $\boldsymbol{\delta}_L$ remains fixed; inference redistributes this error through latent layers. The weight update magnitudes correspondingly show balanced distributions, ensuring that learning signals reach all layers effectively. Consequently, the meta-PE objective and weight regularization regulate the operator $W_l^\top D(\mathbf{h}_{l+1}^{(0)})$ to maintain relatively uniform scaling across layers, preventing the geometric decay patterns that created boundary-layer dominance in conventional PCNs.

---

*For classification tasks, the squared error at the output layer is replaced with a cross-entropy loss.

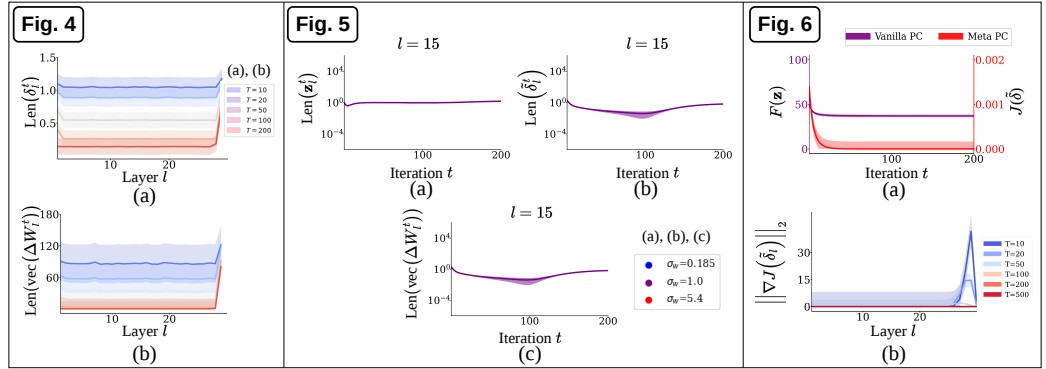

Figure 4: The layer-wise distributions of lengths in Meta-PCNs: (a) $\text{len}(\boldsymbol{\delta}_l^t)$, and (b) $\text{len}(\text{vec}(\Delta W_l^t))$. *Setting:* same as in Figure 2.

Figure 5: Dynamics of (a) $\text{len}(\mathbf{z}_l^t)$, (b) $\text{len}(\boldsymbol{\delta}_l^t)$, and (c) $\text{len}(\text{vec}(\Delta W_l^t))$ across $\sigma_w \in \{0.185, 1.0, 5.4\}$ in Meta-PCNs. Due to weight variance normalization, the three $\sigma_w$ cases overlap and appear as a single trajectory. *Setting:* same as in Figure 3.

Figure 6: Convergence dynamics and equilibrium analysis during inference. (a) Objective function convergence showing Meta-PCN's meta-PE objective $\mathcal{J}$ over $\tilde{\boldsymbol{\delta}}$ (right y-axis) and conventional PCN's standard PE objective $\mathcal{F}$ over $\mathbf{z}$ (left y-axis) in the inference phase. (b) Layer-wise convergence to delta relationships in Meta-PCN, measured by the gradient norm of the objective function.

**Stable PEs.** Figure 5 demonstrates that Meta-PCN substantially mitigates the exponential growth and decay patterns that characterized EVPE in conventional PCNs. The temporal dynamics remain stable with controlled magnitudes across all weight variance settings. The latent state lengths (Figure 5a), PE lengths (Figure 5b), and weight update lengths (Figure 5c) all maintain stable trajectories without the dramatic exponential scaling observed in Figure 3. Because we normalize weight variance to enforce a uniform scale across all conditions, the three different cases ($\sigma_w \in \{0.185, 1.0, 5.4\}$) overlap and are visually indistinguishable, appearing as a single trajectory in the figure. This is an expected result since variance normalization maintains the $\sigma_w$ of weight matrices at the same scale regardless of the initial $\sigma_w$. Therefore, the multiplicative scaling factors $\tau_t(\sigma_w)$ that previously caused geometric growth or decay are now effectively controlled through the variance-based weight regularization strategy. This regulation of multiplicative factors eliminates the root cause of EVPE while preserving essential PC dynamics. All quantities remain within manageable ranges, preventing both explosion and vanishing dynamics.

**Enhanced Convergence Properties.** We evaluate convergence improvements by comparing the inference dynamics of conventional PCN and Meta-PCN, measuring the reduction in their respective objectives. Although a direct comparison is inherently limited by the differences in objectives, Figure 6 reveals substantial convergence improvements achieved by Meta-PCN. The reduction of the objective in Meta-PCN exhibits rapid and definitive convergence to zero, in contrast to the slow convergence dynamics of conventional PCNs (Figure 6a). Layer-wise gradient norm analysis (Figure 6b) demonstrates the rapid resolution of high meta-PEs initially observed in some layers, indicating quick convergence to equilibrium. Overall, Meta-PCN achieves better convergence than conventional PCNs while maintaining stability across all layers. The linearized meta-PE-based objective empirically demonstrates improved convergence properties compared to the original nonlinear equilibrium system (Figure 6). The transformation from a nonlinear to a linearized system addresses a fundamental convergence limitation of conventional PC inference.

## 6 META-PCN ENABLES SCALABLE DEEP LEARNING

**Experimental Setup.** We evaluate Meta-PCN on CIFAR-10, CIFAR-100, and TinyImageNet datasets using VGG and ResNet architectures of varying depths. We directly compare backpropagation (BP), conventional PCNs with only feed-forward initialization, and the complete Meta-PCN framework. Our three-way comparison is conducted under identical experimental settings except for algorithmic differences: the underlying architectures and all shared hyperparameters (optimizer, learning rate, batch size, etc.) are kept exactly the same across BP, PCN, and Meta-PCN; only the objective function and update rules differ (see Appendix E for details). Meta-PCN does not introduce

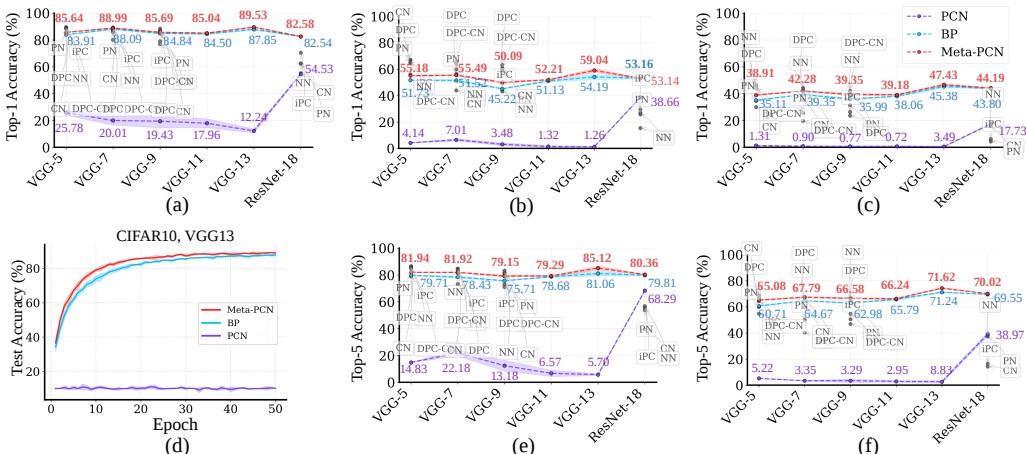

Figure 7: Classification accuracy across network architectures (VGG-5, 7, 9, 11, 13, & ResNet-18) and datasets (CIFAR-10/100 and TinyImageNet). Performance comparison of Meta-PCN, back-propagation (BP), and PCN on (a) CIFAR-10 (Top-1 acc.), (b) CIFAR-100 (Top-1 acc.), (c) Tiny-ImageNet (Top-1 acc.), (d) Training dynamics on CIFAR-10 with VGG-13 showing test accuracy evolution over 50 epochs, (e) CIFAR-100 (Top-5 acc.), and (f) TinyImageNet (Top-5 acc.). Mean accuracy values over 5 runs are annotated for each method, architecture, and dataset (see Appendix H for standard deviations and statistical significance tests). Scattered points indicate literature-reported results for PCN variants: positive/negative/centered nudging (PN/NN/CN) (Pinchetti et al., 2024), incremental PC (iPC) (Salvatori et al., 2023b), and Deep PCN (DPC) / DPC with centered nudging (DPC-CN) (Qi et al., 2025).

additional hyperparameters beyond those in conventional PCNs (inference rate, number of inference steps, optimizer settings); its computational overhead is analyzed in Appendix G. Additionally, we compare various PC variants by directly incorporating values from their respective studies. We note that some recent PCN implementations (Pinchetti et al., 2024) employ auxiliary training heuristics such as GELU activations, momentum during inference, and no weight decay—choices orthogonal to the core PC mechanism. Our experiments deliberately use a standard training protocol to isolate the effect of Meta-PCN's structural improvements rather than method-specific tuning.

**Performance Across Network Depths and Datasets.** Figure 7 presents classification accuracy across different architectures and datasets. The results reveal distinct performance patterns consistent with our theoretical predictions about PCN scalability limitations and Meta-PCN's effectiveness in addressing them. On CIFAR-10 (Figure 7a), conventional PCNs exhibit severe performance degradation with increasing depth, achieving only 10–20% accuracy across most architectures. In contrast, Meta-PCN demonstrates strong stability, maintaining 80–90% accuracy across all tested depths. Notably, within our controlled experimental setting, Meta-PCN outperforms BP across most architectures, with improvements ranging from 0.04% (ResNet-18) to 1.73% (VGG-5), four of which are statistically significant, suggesting that local learning rules need not compromise competitive performance (see Table 2 in Appendix H). Similar patterns emerge on CIFAR-100 (Figure 7b) and TinyImageNet (Figure 7c). Across all 30 dataset–architecture–metric configurations, Meta-PCN outperforms BP in 29 cases (22 statistically significant), with an average improvement of 2.15%. The sole exception is CIFAR-100 ResNet-18 Top-1, where BP marginally leads by 0.02% ($p=0.84$); see Appendix H for full results.

Our experimental design employs stringent controls to ensure identical conditions across BP, PCN, and Meta-PCN. This rigorous standardization may result in subdued BP performance relative to literature reports employing method-specific optimizations; likewise, literature-reported PCN variants consistently underperformed their corresponding BP implementations in those studies, underscoring the importance of controlled comparative evaluation. Despite this conservative comparison protocol, in deeper architectures (VGG-13, ResNet-18), Meta-PCN matches or exceeds BP without requiring architecture-specific tuning, suggesting that the proposed theoretical resolution provides scalability advantages.

**Training Dynamics and Stability.** Figure 7d illustrates learning trajectories over 50 training epochs, comparing the three approaches on a representative architecture and dataset (VGG-13 on CIFAR-10). The analysis reveals fundamental differences in optimization behavior that complement

our framework. Conventional PCNs exhibit minimal learning progression, plateauing at approximately 12% accuracy throughout training. This behavior is consistent with the gradient starvation phenomenon identified in our theoretical analysis. Meta-PCN, by contrast, demonstrates smooth and monotonic improvement, closely paralleling BP's learning trajectory and achieving higher final accuracy in this configuration (Meta-PCN 89.53% vs. BP 87.85%).

Ablation results are reported in Appendix I. The ablation confirms that the meta-PE objective is essential (removing it drops accuracy from 89.5% to 10.0%), while weight regularization provides a statistically significant but moderate improvement (1.3%).

## 7 RELATED WORK

With increasing depth, PCN benchmarks report deteriorating accuracy, skewed layer-wise prediction-error and energy profiles, and growing imbalances in relaxation speeds (Pinchetti et al., 2024; Kinghorn et al., 2023; Qi et al., 2025). Symptomatic remedies tune schedules or normalizers; for instance, interleaving state and weight updates improves robustness and speed but does not give depth-agnostic guarantees for stable inference (Salvatori et al., 2023b). Theoretical reinterpretations cast PC updates as implicit stochastic gradients or connect PCNs to backpropagation and equilibrium methods, clarifying step-size stability while offering few constructive prescriptions for removing depth-induced pathologies (Alonso et al., 2022; Millidge et al., 2022a).

Some depth-focused methods reduce error-distortion accumulation but rely on temporally non-local updates that do not enforce contraction (Qi et al., 2025). Others reparameterize objectives to optimize directly in error space, mitigating exponential signal attenuation (Goemaere et al., 2025), or adopt depth-aware parameterization and learning-rate scaling to train deep residual PCNs (Innocenti et al., 2025). Collectively, these advances boost performance yet still fall short of general, architecture-agnostic guarantees of stable inference.

A separate line of work has explored theoretical connections between predictive coding and backpropagation. Whittington & Bogacz (2017) showed that PC updates can approximate backpropagation under certain conditions, and Millidge et al. (2022a) extended this analysis to arbitrary computation graphs. While these studies clarify when PC and BP produce similar learning signals, they do not address the stability pathologies that arise in deep PCNs. Our meta-PE objective is complementary: rather than establishing equivalence to BP, it is designed to stabilize inference dynamics by minimizing prediction errors of prediction errors, directly targeting the EVPE and PE imbalance phenomena identified in our analysis.

Our contribution is to treat stability as the primary design objective and to provide a unified, theory-guided remedy for depth-induced pathologies in PCNs. Using a dynamical mean-field, length-based analysis, we show that PE imbalance and EVPE arise from boundary conditions and the spectral statistics of the weights, and we use this account to design Meta-PCN: a meta-PE objective, together with variance-based weight normalization that controls the effective spectral norms. These components enforce contraction and scale separation during relaxation while preserving fully local learning, yielding stable and scalable training across the architectures and datasets tested. We emphasize that feed-forward-based latent initialization is standard practice in the PCN literature and not part of our claimed contribution; our novelty lies in the DMFT-guided diagnosis of these pathologies and the resulting meta-PE plus variance-normalization framework.

## 8 CONCLUSION

This study identifies two fundamental pathologies that impede the scalability of deep PCNs. Through DMFT analysis, we analytically characterize how PE imbalance and EVPE constitute the core obstacles to deep PCN training. We propose the Meta-PCN framework that systematically addresses these problems through two synergistic components: (1) a meta-PE-based loss function that linearizes the nonlinear equilibrium system to provide stable dynamics, and (2) variance-based weight regularization that suppresses EVPE. Sustained improvements across diverse datasets and architectures, coupled with stable training dynamics, position Meta-PCN as a promising step toward a practical alternative to backpropagation-based optimization while preserving the local learning properties relevant to neuromorphic implementations.

## ACKNOWLEDGMENTS

We thank Jung Young Kim for valuable advice and discussions on biological plausibility. This research was partly supported by the MSIT (Ministry of Science, ICT), Korea, under the Global Research Support Program in the Digital Field program (RS-2024-00436680) supervised by the IITP (Institute for Information & Communications Technology Planning & Evaluation). This project was partly supported by Microsoft Research Asia. This work was partly supported by Global Research Cluster program of Samsung Advanced Institute of Technology. This work was partly supported by Institute for Information & Communications Technology Planning & Evaluation (IITP) grant funded by the Korea government (MSIT) (No. RS-2023-00233251, System3 reinforcement learning with high-level brain functions). This work was partly supported by Institute for Information & Communications Technology Planning & Evaluation (IITP) grant funded by the Korea government (MSIT) (RS-2019-II190075, Artificial Intelligence Graduate School Program (KAIST)). This work was partly supported by the National Research Foundation of Korea (NRF) grant funded by the Korea government (MSIT) (RS-2024-00341805). This work was partly supported by the National Research Foundation of Korea (NRF) grant funded by the Korea government (MSIT) (RS-2022-NR068758).

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

# A BIOLOGICAL PLAUSIBILITY OF META-PCNS

## A.1 BIOLOGICAL PLAUSIBILITY OF INFERENCE AS META–FREE ENERGY MINIMIZATION

Our inference objective fixes each layer's feed-forward target $c_l$ and minimizes the norm of meta-PE, $\|\tilde{\delta}_l - g_l(\tilde{\delta}_{l+1}^*, h_{l+1}^{(0)})\|_2^2$ so that the realized local error $\tilde{\delta}_l^t$ matches a top-down prediction of that error. This "error of errors" matching is consistent with hierarchical predictive coding accounts in which ascending signals progressively encode residuals of residuals, supported by a division of labor between superficial error units and deep representation units, and by laminar asymmetries in feed-forward versus feedback pathways (Rao & Ballard, 1999; Bastos et al., 2012; Keller & Mrsic-Flogel, 2018). Dendritic predictive coding further suggests that basal and apical dendritic compartments can locally compare bottom-up errors with top-down predictions, implementing error computation without requiring distinct error neurons (Mikulasch et al., 2023; Salvatori et al., 2023a). Contemporary cortical PC models operate at two complementary levels: (i) laminar hierarchies where deep layers receive predictions and superficial layers relay PEs, and (ii) local microcircuits where pyramidal cells compute PEs via dendritic comparison (Hertäg & Clopath, 2022; Denève & Machens, 2016). Meta-PCN's meta-prediction errors can be implemented by interacting error neuron populations or distinct dendritic compartments. Recent thalamocortical models with lamina-specific prediction and error pathways further support this circuitry (George et al., 2025). Taken together, these views align with our formulation as an explicit hierarchical comparison between realized errors and their predicted values, while remaining faithful to canonical message-passing in cortical microcircuits (Bastos et al., 2012). A complementary line of work interprets meta-error minimization as dynamically adjusting precision weights on errors. Predictive coding theories posit context-dependent gain control of sensory error signals, with neuromodulatory systems (notably cholinergic) and thalamocortical circuits (via the pulvinar) regulating the effective gain of error transmission in a task-dependent manner (Feldman & Friston, 2010; Moran et al., 2013; Kanai et al., 2015). More broadly, prediction errors serve as a core computational currency not only in perception but also in reward-guided decision-making, where interdisciplinary perspectives linking neuroscience and machine learning have proven fruitful (Lee & Seymour, 2019; Lee et al., 2019). Our objective reduces the mismatch between observed errors and expected precision, offering a mechanistic bridge between these theoretical proposals and graded, circuit-level control of error gain. Furthermore, clamping $c_l$ during inference is supported by the presence of hierarchical intrinsic timescales in the cortex: higher-order association areas evolve slowly, acting as quasi-fixed boundary conditions over short inference windows, while faster superficial error dynamics in lower areas relax toward these stable top-down states (Murray et al., 2014; Chaudhuri et al., 2015; Bastos et al., 2012).

## A.2 BIOLOGICAL PLAUSIBILITY OF FIXED FEED-FORWARD PREDICTIONS VIA HIERARCHICAL TIMESCALES AND WORKING MEMORY

The fixed feed-forward predictions in our inference scheme are biologically plausible given known temporal hierarchies and active memory mechanisms in the brain. Higher cortical areas exhibit significantly longer intrinsic timescales than lower sensory areas, meaning that higher-level neural activity changes more slowly over time (Murray et al., 2014; Kiebel et al., 2008). Consequently, during the brief timescales in which lower-level error neurons undergo rapid update cycles, the top-down signals from higher areas can be treated as essentially constant, providing a stable context or boundary condition for lower-level processing. In effect, Meta-PCN's choice to hold each $c_l$ fixed throughout inference mirrors the brain's own use of slowly varying higher-level activity to constrain fast iterative dynamics in subordinate regions, in line with a hierarchy of intrinsic timescales.

In addition to this passive timescale separation, the brain actively maintains persistent signals through recurrent circuitry, which underlies working memory. Empirical and computational studies demonstrate that cortical networks – especially in frontal areas – can sustain neural activity patterns even after an external stimulus is removed, via recurrent excitation and slow NMDA receptor–mediated currents that support an attractor state (Wang, 2001; 2021). This mechanism allows information (e.g., a sensory prediction) to be held "online" for hundreds of milliseconds to seconds, rather than decaying immediately. Notably, such sustained memory states are not confined to a single high-level region but are distributed across multiple interconnected areas (Mejías & Wang, 2022), implying that each layer in a hierarchy can uphold its prediction at its own timescale. Meta-PCN's feed-forward pass initialization and subsequent clamping of $c_l$ can thus be interpreted as the network

entering a regime of active maintenance: the initial top-down predictions are kept stable by internal dynamics (analogous to a global attractor state) while errors are iteratively corrected. This corresponds to an active maintenance of predictions within the network, rather than a passive snapshot or cache of activations. In contrast to backpropagation's practice of storing intermediate activations for later use – a procedure that violates the real-time, local processing constraint of biological circuits – here the information is continuously present in the neural state itself. Such sustained representations respect temporal locality and are energetically efficient, aligning with the brain's tendency to minimize surprise by keeping expected or predictable inputs in a stable active state (Friston, 2010).

While these mechanisms provide biologically grounded interpretations, we acknowledge that Meta-PCN does not fully satisfy the strictest definitions of temporal locality. We view it as a step toward more temporally local predictive coding frameworks: it demonstrates that deep PCNs can be stabilized through principled mechanisms, and future work can build on these stability properties while enforcing stricter temporal constraints.

### A.3 BIOLOGICAL PLAUSIBILITY OF WEIGHT REGULARIZATION

Our weight regularizer rescales each weight matrix $W$ multiplicatively according to its layer-wise variance, in order to keep the effective operator near unit spectral norm. This stabilizes the dynamics of the feedback term $W^\top D(\mathbf{h})$ across layers. Functionally, this mirrors divisive normalization, a canonical neural computation in which neuronal responses are divided by a pooled activity term to control gain across populations (Carandini & Heeger, 2012). Interpreting our variance-governed rescaling as a form of parameter-space gain control links it to convergent physiological mechanisms for divisive normalization observed across sensory systems and cortical areas (Carandini & Heeger, 2012). The same multiplicative scheme is consistent with global synaptic scaling, which multiplicatively adjusts all excitatory synapses to maintain target firing rates, and also with theoretical local rules that explicitly normalize weight vectors (Turrigiano et al., 1998; Oja, 1982). At finer scales, heterosynaptic plasticity and presynaptic normalization processes can constrain synaptic strength variance branchwise and along individual axonal arbors, yielding population-level effects equivalent to our matrix-level multiplicative regularization (Letellier et al., 2019; Hardingham et al., 2007).

## B DISCUSSION

### B.1 CONTRIBUTIONS

This work aims not only to document empirical instabilities in deep predictive coding networks (PCNs), but to explain them through a precise structural mechanism and to derive a unified, theory-guided remedy. Our main contributions are summarized as follows:

- **DMFT-based analysis of deep PCN dynamics.** We develop a dynamical mean-field, length-based framework for deep PCNs under Gaussian initialization, which analytically tracks the evolution of latent-state, PE, and weight-update lengths across both layer and inference iteration steps. This framework, which does not presuppose feed-forward-based initialization, already reveals the characteristic U-shaped PE profile and exponential scaling patterns in a mathematically tractable setting.

- **Theoretical reproduction and diagnosis of PE imbalance and EVPE.** Within this framework we reproduce two core pathologies of deep PCNs—PE imbalance and EVPE—and show that they arise structurally from the underlying dynamics. PE lengths decay geometrically away from boundary layers due to products of spectral factors, while EVPE appears as geometric growth or decay over inference iterations governed by multiplicative factors that depend on weight variance and activation gains. This analysis isolates the root causes in terms of boundary conditions, hierarchical layer interactions, and the spectral statistics of the weight operators, going beyond explanations based solely on discretization or numerical heuristics.

- **Meta-PCN: a unified, theory-guided remedy.** Guided by this diagnosis, we introduce Meta-PCN, which combines (i) a meta–prediction-error objective that minimizes "prediction errors of prediction errors" by replacing the standard free-energy objective with a loss equal to the linearized equilibrium residual, and (ii) a variance-based normalization

rule that controls effective spectral norms via random-matrix–motivated scaling. Together, these components enforce contraction and scale separation during relaxation while preserving fully local learning, thereby mitigating both PE imbalance and EVPE within a single, principled framework rather than via separate engineering fixes.

- **Stable and scalable deep learning with local learning rules.** Extensive experiments on CIFAR-10/100 and TinyImageNet with VGG and ResNet architectures demonstrate that Meta-PCN substantially improves inference stability, convergence speed, and classification performance. Across most configurations, Meta-PCN achieves large gains over conventional PCNs and performs competitively with backpropagation, all while maintaining the local learning rules and architectural constraints of PCNs.

## B.2 LIMITATIONS

This study has several limitations. First, Meta-PCN's performance has not reached complete parity with backpropagation. While this represents substantial progress achieved while maintaining biological constraints, it suggests that performance gaps may still exist in some application domains. Second, the current analysis focuses primarily on computer vision tasks, requiring additional validation of its generalization to other domains. Third, a comprehensive analysis is lacking regarding whether variance-based weight regularization is equally effective across all architectures and activation functions. Fourth, for very deep architectures, when the number of inference steps $T$ is smaller than the network depth $L - 1$, prediction errors cannot fully propagate from the output to input boundary within a single inference episode, leading to performance degradation. This limitation may be alleviated in streaming or continual settings where inference steps evolve with real-time data.

## B.3 FUTURE WORK

This work can be extended in several directions. First, the effectiveness of Meta-PCN should be validated on larger-scale datasets such as ImageNet and in other domains such as natural language processing with text-based architectures and reinforcement learning. Second, more sophisticated weight regularization techniques and adaptive inference schemes could be developed to further improve stability across diverse architectures. Third, reinterpreting frozen predictions as slowly updating variables and latent states as fast-updating variables may enable Meta-PCN variants naturally adapted to streaming settings and continuous-time formulations, which would also strengthen connections to biological real-time processing.

## B.4 RELATIONSHIP TO PRECISION LEARNING

In the classical predictive coding framework, precision $\Pi_l = \Sigma_l^{-1}$ is defined as the inverse covariance of latent activity and acts as a reliability weight on prediction errors, yielding the free energy $\mathcal{F} = \frac{1}{2} \sum_l \boldsymbol{\delta}_l^\top \Pi_l \boldsymbol{\delta}_l$. This scaling can partially compensate for PE imbalance across layers. Our work deliberately fixes $\Pi_l = I$ and focuses on why PE imbalance and EVPE arise in deep PCNs.

There is an important distinction between precision learning and our weight-variance control:

- Precision, defined as the inverse covariance of latent activity, rescales $\boldsymbol{\delta}_l$ directly via $\Pi_l$.

- Our variance-based normalization operates on the variance/spectrum of weights, controlling $\sigma_w^2$ and $\|W_l\|_2$ to prevent amplification/attenuation of errors.

Our DMFT analysis shows that PE variance is primarily determined by weight statistics through multiplicative scaling $\|\boldsymbol{\delta}_l^{t+1}\| \approx \tau_t(\sigma_w)\|\boldsymbol{\delta}_l^t\|$. Thus, the most direct control for the pathology is weight variance rather than precision alone. When weight-induced error variance is normalized toward identity, the optimal precision $\Pi_l = \Sigma_l^{-1}$ becomes close to $I$, making our fixed identity precision appropriate. We view precision learning and weight-variance control as complementary strategies, with their integration as future work.

## C DERIVATION OF LENGTH DYNAMICS

This section provides the complete mathematical foundation for the length dynamics framework introduced in Section 3.1. We present the theoretical aspects of how latent states evolve during the inference process, leveraging assumptions of linearity and Gaussian-distributed parameters. We focus on the statistical distribution of Gaussian samples, which serves as the foundation for understanding the Gaussian ensemble network's behavior under large-scale computations. We present a rigorous analysis of interaction matrices of latents and bias and their dynamics, offering insights into the length dynamics of latent states $\text{len}(\mathbf{z}_l^t)$, PEs $\text{len}(\boldsymbol{\delta}_l^t)$, and weight updates $\text{len}(\text{vec}(\Delta W_l^t))$ as defined in Section 3.1.

### C.1 ASSUMPTIONS AND LATENT STATE UPDATE RULE

Our primary goal is to track the changes in the length of the latent state during the inference step $t$. To perform this analysis, we adopt the following assumptions:

**Gaussian Assumption** We assume that the initial latent state at the inference step $t = 0$ is drawn i.i.d. as $z_{i,l}^0 \sim \mathcal{N}(0, 1)$. The learnable parameters, weight and bias, are drawn i.i.d. as $w_{i,j,l} \sim \mathcal{N}(0, \frac{\sigma_w^2}{N_l})$ and $b_{i,l} \sim \mathcal{N}(0, \sigma_b^2)$.

**Linearity Assumption** The correlation between variables may vary arbitrarily depending on the nonlinearity of the activation function $\phi$, making it difficult to expand interaction analytically. Therefore, we initially analyze the case where $\phi$ is a linear function. For cases involving nonlinear activation functions, empirical verification is performed to confirm the results in Appendix K. In the context of the linearity assumption, the forward and backward transformations are defined as follows:

$$f_{l-1}(\mathbf{z}_{l-1}) = W_{l-1}\mathbf{z}_{l-1} + \mathbf{b}_{l-1} \tag{1}$$

$$g_l(\mathbf{z}_{l+1}) = W_l^\top \mathbf{z}_{l+1}. \tag{2}$$

**Dimensionality Assumption** We assume that all layers share the same dimensionality. If the dimensions differ, the latent spaces must be transformed using matrices like $M$, resulting in non-generalizable cross-layer interactions.

With these assumptions, we can expand the latent state update rule as follows:

$$\begin{aligned}
\mathbf{z}_l^{t+1} &= \mathbf{z}_l^t + \Delta \mathbf{z}_l^t \\
&= \mathbf{z}_l^t + \eta\left(-\delta_l^t + W_l^\top \delta_{l+1}^t\right) \\
&= \mathbf{z}_l^t + \eta\left(-\left(\mathbf{z}_l^t - \hat{\mathbf{z}}_l^t\right) + W_l^\top\left(\mathbf{z}_{l+1}^t - \hat{\mathbf{z}}_{l+1}^t\right)\right)
\end{aligned} \tag{3}$$

The update rule can be further simplified as:

$$\begin{aligned}
\mathbf{z}_l^{t+1} &= (1 - \eta)\mathbf{z}_l^t + \eta \cdot (W_{l-1}\mathbf{z}_{l-1}^t + \mathbf{b}_{l-1}) \\
&\quad + \eta \cdot W_l^\top \mathbf{z}_{l+1}^t - \eta \cdot W_l^\top\left(W_l \mathbf{z}_l^t + \mathbf{b}_l\right) \\
&= \nu M_{l-1}\mathbf{z}_{l-1}^t + \kappa \mathbf{z}_l^t + \nu M_l^\top \mathbf{z}_{l+1}^t + \eta \mathbf{b}_{l-1} \\
&\quad - \nu M_l^\top \mathbf{b}_l
\end{aligned} \tag{4}$$

where $M = \frac{1}{\sigma_w}W$, $\nu = \eta\sigma_w$, and $\kappa = 1 - \eta(1 + \sigma_w^2)$.

### C.2 THE DISTRIBUTION OF THE PRODUCT OF GAUSSIAN SAMPLES

Before delving into the dynamics of length, given that our analysis involves the product of different forms of Gaussian samples, it is essential to review the distributional properties of products of Gaussian samples. Let $u_i \sim \mathcal{N}(0, \frac{\sigma_u^2}{N})$ and $v_i \sim \mathcal{N}(0, \frac{\sigma_v^2}{N})$. The square of $u_i$ follows a chi-square distribution, while the product $u_i \cdot v_i$ follows a normal product distribution. Our interest lies in

understanding the distribution of the following inner product values

$$\mathbf{u}^\top \mathbf{u} = \sum_{i=1}^{N} u_i^2 \quad \text{and} \quad \mathbf{u}^\top \mathbf{v} = \sum_{i=1}^{N} u_i \cdot v_i \tag{5}$$

as $N \to \infty$. Applying the Central Limit Theorem (CLT) to these values, we obtain the following:

$$\sqrt{N} \cdot \frac{\frac{\mathbf{u}^\top \mathbf{u}}{N} \mathbb{E}[u_i^2]}{\sqrt{\text{Var}(u_i^2)}} \to \mathcal{N}(0, 1), \tag{6}$$

where $\mathbb{E}[u_i^2] = \text{Var}(u_i) = \frac{\sigma_u^2}{N}$, and $\text{Var}(u_i^2) = \mathbb{E}[u_i^4] - \mathbb{E}[u_i^2]^2 = 3\frac{\sigma_u^4}{N^2} - \frac{\sigma_u^4}{N^2} = 2\frac{\sigma_u^4}{N^2}$. As a result, $\mathbf{u}^\top \mathbf{u} \to \mathcal{N}(\sigma_u^2, \frac{2\sigma_u^4}{N})$, and equivalently,

$$\mathbf{u}^\top \mathbf{u} \to \sigma_u^2 \cdot \mathcal{N}(1, \frac{2}{N}). \tag{7}$$

Similarly, applying the CLT to the cross-product yields:

$$\sqrt{N} \cdot \frac{\frac{\mathbf{u}^\top \mathbf{v}}{N} \mathbb{E}[u_i \cdot v_i]}{\sqrt{\text{Var}(u_i \cdot v_i)}} \to \mathcal{N}(0, 1), \tag{8}$$

where $\mathbb{E}[u_i \cdot v_i] = 0$, since $u_i$ and $v_i$ are independent, and $\text{Var}(u_i \cdot v_i) = \sigma_u^2 \cdot \sigma_v^2$. Hence, we obtain $\mathbf{u}^\top \mathbf{v} \to \mathcal{N}(0, \frac{\sigma_u^2 \cdot \sigma_v^2}{N})$. Equivalently, for large $N$,

$$\mathbf{u}^\top \mathbf{v} \sim \sigma_u \sigma_v \cdot \mathcal{N}(0, \frac{1}{N}), \tag{9}$$

and if $\sigma_u = \sigma_v$, this converges to $\sigma_u^2 \cdot \mathcal{N}(0, \frac{1}{N})$.

We can conduct a similar analysis for the distribution of vector lengths. Let $u_i \sim \mathcal{N}(0, \sigma_u^2)$ and $v_i \sim \mathcal{N}(0, \sigma_v^2)$. In these cases, we want to understand the asymptotic distribution of the following length terms:

$$\langle u_i^2 \rangle = \frac{1}{N} \sum_{i=1}^{N} u_i^2 \quad \text{and} \quad \langle u_i \cdot v_i \rangle = \frac{1}{N} \sum_{i=1}^{N} u_i \cdot v_i, \tag{10}$$

as $N \to \infty$. Note that the variance of the Gaussian distribution in the length calculation is not divided by $N$ in contrast to the inner product version. Instead, the length includes a division by $N$. By applying the CLT, similar to the inner product case, we have:

$$\langle u_i^2 \rangle \to \sigma_u^2 \cdot \mathcal{N}(1, \frac{2}{N}). \tag{11}$$

Using this result, we can apply it to the cases of interest.

**Lengths** In the case of vector-vector multiplication, consider vectors $\mathbf{z}_1$, $\mathbf{z}_L$, and $\mathbf{b}_l$, where $l \in \{1, \ldots, L-1\}$. Each of these vectors is assumed to be sampled from a Gaussian distribution, i.e., each element is drawn from $\mathcal{N}(0, 1)$. The length defined by the relationship between these vectors, as $N \to \infty$, follows:

$$\langle u_i^2 \rangle \to \mathcal{N}(1, \frac{2}{N}), \tag{12}$$

while the cross-product between different vectors converges to:

$$\langle u_i \cdot v_i \rangle \to \mathcal{N}(0, \frac{1}{N}). \tag{13}$$

Consequently, the self-product (length) converges to 1, while the product with a different vector converges to 0 as $N \to \infty$. Finally, consider the length $l = \frac{1}{N} \mathbf{v}^\top A \mathbf{u}$, where each element of $A$, $A_{ij}$, is drawn from $\mathcal{N}(0, \frac{1}{N})$, and each element of $\mathbf{u}$ and $\mathbf{v}$ follows $\mathcal{N}(0, 1)$. The transformed vector $(A\mathbf{u})_i \sim \mathcal{N}(0, 1)$. Therefore, $\mathbf{v}^\top (A\mathbf{u}) \sim \mathcal{N}(0, 1)$. Thus, the length $l$ follows:

$$l \sim \mathcal{N}(0, \frac{1}{N^2}). \tag{14}$$

**Matrix-Matrix Multiplication** In the case of matrix-matrix multiplication, consider $C = A^\top A$, where each element of $A$, i.e., $A_{ij}$, is drawn from $\mathcal{N}(0, \frac{1}{N})$. The diagonal entries of $C$, $C_{ii}$, follow $\mathcal{N}(1, \frac{2}{N})$. The off-diagonal entries $C_{ij}$, where $i \neq j$, follow $\mathcal{N}(0, \frac{1}{N})$. Hence, $C$ approaches the identity matrix $I$ as $N \to \infty$. For the product of two matrices $D = AB$, where both $A_{ij}$ and $B_{ij}$ are sampled from $\mathcal{N}(0, \frac{1}{N})$, the resulting matrix $D_{ij}$ shares the same distribution as $A_{ij}$ and $B_{ij}$.

## C.3 INTERACTION MATRICES

For the analysis of length dynamics, we define several key variables as follows.

**Latent Self-Interaction** Let $P_{l+k,l}^t = \frac{1}{N}\mathbf{z}_{l+k}^{t\top}M_{l+k-1:l}\mathbf{z}_l^t$ for $1 \leq l, l+k \leq L$, where $M_{l+k:l} = M_{l+k}M_{l+k-1}\cdots M_l$ is the product of the series of matrices (as introduced in Section 3.1). By definition, we can observe that $P$ is symmetric, meaning that $P_{l,l+k}^t = P_{l+k,l}^t$. The length of the latent state at layer $l$ and time step $t$, $p_l^t$, can be represented as the diagonal elements of $P^t$, $p_l^t = \left\langle \left(z_{i,l}^t\right)^2 \right\rangle = \frac{1}{N}\sum_{i=1}^N \left(z_{i,l}^t\right)^2 = \frac{1}{N}\mathbf{z}_l^{t\top}\mathbf{z}_l^t = P_{l,l}^t$. With this definition, the matrix $P^t$ contains the length information and interactions between latent states at different layers at the inference step $t$. Since the input and output are fixed during the inference phase as $\mathbf{z}_1^{t+1} = \mathbf{z}_1^t$ and $\mathbf{z}_L^{t+1} = \mathbf{z}_L^t$, the interaction terms with the indices 1 and $L$ are constants as $P_{1,1} = P_{L,L} = 1$ and $P_{1,L} = P_{L,1} = 0$. Similarly, at $t = 0$, $P^0 = I$.

**Bias-Latent State Interaction** Let bias-latent state interaction $B_{l,l-k}^t = \frac{1}{N}\mathbf{b}_{l-1}^\top M_{l-1:l-k}\mathbf{z}_{l-k}^t$ be a bilinear term of interaction between the bias and latent states at layers $l$ and $l - k$ at inference step $t$ for $1 \leq l, l - k \leq L$. Likewise, let $B_{l-k,l}^t = \frac{1}{N}\mathbf{z}_l^{t\top}M_{l-1:l-k}\mathbf{b}_{l-k-1}$. Since the bias, the input ($\mathbf{z}_1$) and output ($\mathbf{z}_L$) are fixed during the inference phase, the interaction terms between these independent components, $B_{\cdot,1}$ and $B_{\cdot,L}$, are also fixed at 0. At $t = 0$, $B^0 = 0$.

**Bias Self-Interaction** The term $\Gamma_{l,l-k}$ represents $\frac{1}{N}\mathbf{b}_l^\top M_{l:l-k+1}\mathbf{b}_{l-k} = 0$ for $1 \leq l, l - k \leq L$. Since the bias terms are sampled from $\mathcal{N}(0, \sigma_b^2)$ and fixed during the inference phase, $\Gamma = \sigma_b^2 I$ is a constant matrix by the properties introduced in Appendix C.2.

## C.4 DYNAMICS OF INTERACTION MATRIX

We derive the update rule for $P$ using the definition of the interaction and the latent update rule in Equation 4. For an element of $P_{l,l-k}^t$, where $l - k > 1$ and $l < L$, the update equation can be described as follows:

$$
\begin{aligned}
P_{l,l-k}^{t+1} &= \frac{1}{N}\mathbf{z}_l^{t+1\top}M_{l-1:l-k}\mathbf{z}_{l-k}^{t+1} \\
&= \frac{1}{N}\left(\kappa \cdot \mathbf{z}_l^t + \nu \cdot M_{l-1}\mathbf{z}_{l-1}^t + \nu \cdot M_l^\top \mathbf{z}_{l+1}^t \right. \\
&\quad \left. +\eta \cdot \mathbf{b}_{l-1} - \nu \cdot M_l^\top \mathbf{b}_l\right)^\top \times M_{l-1:l-k} \\
&\quad \times \left(\kappa \cdot \mathbf{z}_{l-k}^t + \nu \cdot M_{l-k-1}\mathbf{z}_{l-k-1}^t \right. \\
&\quad + \nu \cdot M_{l-k}^\top \mathbf{z}_{l-k+1}^t + \eta \cdot \mathbf{b}_{l-k-1} \\
&\quad \left. -\nu \cdot M_{l-k}^\top \mathbf{b}_{l-k}\right)
\end{aligned}
$$

We want to expand this equation fully, showing all combinations of terms in the product. First, we identify the components of the vectors involved in the equation. The expression consists of a sum of transposed vectors, multiplied by a matrix $M_{l-1:l-k}$, and then multiplied by another sum of vectors. The components of the first sum of vectors are

$$
\begin{aligned}
\mathbf{u}_1 &= \nu \cdot M_{l-1}\mathbf{z}_{l-1}^t, \mathbf{u}_2 = \kappa \cdot \mathbf{z}_l^t, \mathbf{u}_3 = \nu \cdot M_l^\top \mathbf{z}_{l+1}^t, \\
\mathbf{u}_4 &= \eta \cdot \mathbf{b}_{l-1}, \quad \text{and} \quad \mathbf{u}_5 = -\nu \cdot M_l^\top \mathbf{b}_l.
\end{aligned}
$$

The components of the second sum of vectors are

$$\mathbf{v}_1 = \nu \cdot M_{l-k-1}\mathbf{z}_{l-k-1}^t, \mathbf{v}_2 = \kappa \cdot \mathbf{z}_{l-k}^t,$$
$$\mathbf{v}_3 = \nu \cdot M_{l-k}^\top \mathbf{z}_{l-k+1}^t, \mathbf{v}_4 = \eta \cdot \mathbf{b}_{l-k-1},$$
$$\text{and} \quad \mathbf{v}_5 = -\nu \cdot M_{l-k}^\top \mathbf{b}_{l-k}.$$

We can rewrite the original equation using the components we defined:

$$P_{l,l-k}^{t+1} = \frac{1}{N} \left(\mathbf{u}_1 + \mathbf{u}_2 + \mathbf{u}_3 + \mathbf{u}_4 + \mathbf{u}_5\right)^\top M_{l-1:l-k}\left(\mathbf{v}_1 + \right.$$
$$\left. \mathbf{v}_2 + \mathbf{v}_3 + \mathbf{v}_4 + \mathbf{v}_5\right)$$

We compute all possible products $\mathbf{u}_i^\top M_{l-1:l-k}\mathbf{v}_j$ for $i, j = 1$ to $5$.

- Terms involving $\mathbf{u}_1$:

$$\mathbf{u}_1^\top M_{l-1:l-k}\mathbf{v}_1 = \nu^2 \left(\mathbf{z}_{l-1}^t\right)^\top M_{l-1}^\top M_{l-1:l-k} M_{l-k-1}$$
$$\mathbf{z}_{l-k-1}^t = \nu^2 \cdot P_{l-1,l-k-1}^t$$
$$\mathbf{u}_1^\top M_{l-1:l-k}\mathbf{v}_2 = \nu\kappa \left(\mathbf{z}_{l-1}^t\right)^\top M_{l-1}^\top M_{l-1:l-k}\mathbf{z}_{l-k}^t$$
$$= \kappa\nu \cdot P_{l-1,l-k}^t$$
$$\mathbf{u}_1^\top M_{l-1:l-k}\mathbf{v}_3 = \nu^2 \left(\mathbf{z}_{l-1}^t\right)^\top M_{l-1}^\top M_{l-1:l-k} M_{l-k}^\top$$
$$\mathbf{z}_{l-k+1}^t = \nu^2 \cdot P_{l-1,l-k+1}^t$$
$$\mathbf{u}_1^\top M_{l-1:l-k}\mathbf{v}_4 = \nu\eta \left(\mathbf{z}_{l-1}^t\right)^\top M_{l-1}^\top M_{l-1:l-k}\mathbf{b}_{l-k-1}$$
$$= \nu\eta \cdot B_{l-k,l-1}^t$$
$$\mathbf{u}_1^\top M_{l-1:l-k}\mathbf{v}_5 = -\nu^2 \left(\mathbf{z}_{l-1}^t\right)^\top M_{l-1}^\top M_{l-1:l-k} M_{l-k}^\top$$
$$\mathbf{b}_{l-k} = -\nu^2 \cdot B_{l-k+1,l-1}^t$$

- Terms involving $\mathbf{u}_2$:

$$\mathbf{u}_2^\top M_{l-1:l-k}\mathbf{v}_1 = \kappa\nu \left(\mathbf{z}_l^t\right)^\top M_{l-1:l-k} M_{l-k-1}\mathbf{z}_{l-k-1}^t$$
$$= \kappa\nu \cdot P_{l,l-k-1}^t$$
$$\mathbf{u}_2^\top M_{l-1:l-k}\mathbf{v}_2 = \kappa^2 \left(\mathbf{z}_l^t\right)^\top M_{l-1:l-k}\mathbf{z}_{l-k}^t$$
$$= \kappa^2 \cdot P_{l,l-k}^t$$
$$\mathbf{u}_2^\top M_{l-1:l-k}\mathbf{v}_3 = \kappa\nu \left(\mathbf{z}_l^t\right)^\top M_{l-1:l-k} M_{l-k}^\top \mathbf{z}_{l-k+1}^t$$
$$= \kappa\nu \cdot P_{l,l-k+1}^t$$
$$\mathbf{u}_2^\top M_{l-1:l-k}\mathbf{v}_4 = \kappa\eta \left(\mathbf{z}_l^t\right)^\top M_{l-1:l-k}\mathbf{b}_{l-k-1}$$
$$= \kappa\eta \cdot B_{l-k,l}^t$$
$$\mathbf{u}_2^\top M_{l-1:l-k}\mathbf{v}_5 = -\kappa\nu \left(\mathbf{z}_l^t\right)^\top M_{l-1:l-k} M_{l-k}^\top \mathbf{b}_{l-k}$$
$$= -\kappa\nu \cdot B_{l-k+1,l}^t$$

- Terms involving $\mathbf{u}_3$:

$$\mathbf{u}_3^\top M_{l-1:l-k}\mathbf{v}_1 = \nu^2 \left(\mathbf{z}_{l+1}^t\right)^\top M_l M_{l-1:l-k} M_{l-k-1}$$
$$\mathbf{z}_{l-k-1}^t = \nu^2 \cdot P_{l+1,l-k-1}^t$$
$$\mathbf{u}_3^\top M_{l-1:l-k}\mathbf{v}_2 = \nu\kappa \left(\mathbf{z}_{l+1}^t\right)^\top M_l M_{l-1:l-k}\mathbf{z}_{l-k}^t$$
$$= \kappa\nu \cdot P_{l+1,l-k}^t$$
$$\mathbf{u}_3^\top M_{l-1:l-k}\mathbf{v}_3 = \nu^2 \left(\mathbf{z}_{l+1}^t\right)^\top M_l M_{l-1:l-k} M_{l-k}^\top$$
$$\mathbf{z}_{l-k+1}^t = \nu^2 \cdot P_{l+1,l-k+1}^t$$
$$\mathbf{u}_3^\top M_{l-1:l-k}\mathbf{v}_4 = \nu\eta \left(\mathbf{z}_{l+1}^t\right)^\top M_l M_{l-1:l-k}\mathbf{b}_{l-k-1}$$
$$= \nu\eta \cdot B_{l-k,l+1}^t$$
$$\mathbf{u}_3^\top M_{l-1:l-k}\mathbf{v}_5 = -\nu^2 \left(\mathbf{z}_{l+1}^t\right)^\top M_l M_{l-1:l-k} M_{l-k}^\top$$
$$\mathbf{b}_{l-k} = -\nu^2 \cdot B_{l-k+1,l+1}^t$$

- Terms involving $\mathbf{u}_4$:

$$\mathbf{u}_4^\top M^{l-1:l-k}\mathbf{v}_1 = \eta\nu \left(\mathbf{b}^{l-1}\right)^\top M^{l-1:l-k} M^{l-k-1}$$
$$\mathbf{z}^{l-k-1,t} = \nu\eta \cdot B_{l,l-k-1}^t$$
$$\mathbf{u}_4^\top M^{l-1:l-k}\mathbf{v}_2 = \eta\kappa \left(\mathbf{b}^{l-1}\right)^\top M^{l-1:l-k}\mathbf{z}^{l-k,t}$$
$$= \kappa\eta \cdot B_{l,l-k}^t$$
$$\mathbf{u}_4^\top M^{l-1:l-k}\mathbf{v}_3 = \eta\nu \left(\mathbf{b}^{l-1}\right)^\top M^{l-1:l-k} M^{l-k^\top}\mathbf{z}^{l-k+1,t}$$
$$= \nu\eta \cdot B_{l,l-k+1}^t$$
$$\mathbf{u}_4^\top M^{l-1:l-k}\mathbf{v}_4 = \eta^2 \left(\mathbf{b}^{l-1}\right)^\top M^{l-1:l-k}\mathbf{b}^{l-k-1}$$
$$= \eta^2 \cdot \gamma^{l-1,l-k-1}$$
$$\mathbf{u}_4^\top M^{l-1:l-k}\mathbf{v}_5 = -\eta\nu \left(\mathbf{b}^{l-1}\right)^\top M^{l-1:l-k} M^{l-k^\top}\mathbf{b}^{l-k}$$
$$= -\nu\eta \cdot \gamma^{l-1,l-k}$$

- Terms involving $\mathbf{u}_5$:

$$\mathbf{u}_5^\top M^{l-1:l-k}\mathbf{v}_1 = -\nu^2 \left(\mathbf{b}^l\right)^\top M^l M^{l-1:l-k} M^{l-k-1}$$
$$\mathbf{z}^{l-k-1,t} = -\nu^2 \cdot B_{l+1,l-k-1}^t$$
$$\mathbf{u}_5^\top M^{l-1:l-k}\mathbf{v}_2 = -\nu\kappa \left(\mathbf{b}^l\right)^\top M^l M^{l-1:l-k}\mathbf{z}^{l-k,t}$$
$$= -\kappa\nu \cdot B_{l+1,l-k}^t$$
$$\mathbf{u}_5^\top M^{l-1:l-k}\mathbf{v}_3 = -\nu^2 \left(\mathbf{b}^l\right)^\top M^l M^{l-1:l-k} M^{l-k^\top}$$
$$\mathbf{z}^{l-k+1,t} = -\nu^2 \cdot B_{l+1,l-k+1}^t$$
$$\mathbf{u}_5^\top M^{l-1:l-k}\mathbf{v}_4 = -\nu\eta \left(\mathbf{b}^l\right)^\top M^l M^{l-1:l-k}\mathbf{b}^{l-k-1}$$
$$= -\nu\eta \cdot \gamma^{l,l-k-1}$$
$$\mathbf{u}_5^\top M^{l-1:l-k}\mathbf{v}_5 = \nu^2 \left(\mathbf{b}^l\right)^\top M^l M^{l-1:l-k} M^{l-k^\top}\mathbf{b}^{l-k}$$
$$= \nu^2 \cdot \gamma^{l,l-k}$$

By systematically breaking down the original equation into its constituent components and computing all possible interactions between them, we have fully expanded the expression:

$$\begin{aligned}
P^{t+1}_{l,l-k} = {}& \nu^2 \cdot P^t_{l-1,l-k-1} + \kappa\nu \cdot P^t_{l-1,l-k} + \nu^2 \cdot P^t_{l-1,l-k+1} \\
& + \nu\eta \cdot B^t_{l-k,l-1} - \nu^2 \cdot B^t_{l-k+1,l-1} \\
& + \kappa\nu \cdot P^t_{l,l-k-1} + \kappa^2 \cdot P^t_{l,l-k} + \kappa\nu \cdot P^t_{l,l-k+1} \\
& + \kappa\eta \cdot B^t_{l-k,l} - \kappa\nu \cdot B^t_{l-k+1,l} \\
& + \nu^2 \cdot P^t_{l+1,l-k-1} + \nu^2 \cdot P^t_{l+1,l-k+1} + \kappa\nu \cdot P^t_{l+1,l-k} + \nu\eta \cdot B^t_{l-k,l+1} \\
& - \nu^2 \cdot B^t_{l-k+1,l+1} \\
& + \nu\eta \cdot B^t_{l,l-k-1} + \kappa\eta \cdot B^t_{l,l-k} + \nu\eta \cdot B^t_{l,l-k+1} \\
& + \eta^2 \cdot \gamma^{l-1,l-k-1} - \nu\eta \cdot \gamma^{l-1,l-k} \\
& - \nu^2 \cdot B^t_{l+1,l-k-1} - \kappa\nu \cdot B^t_{l+1,l-k} \\
& - \nu^2 \cdot B^t_{l+1,l-k+1} - \nu\eta \cdot \gamma^{l,l-k-1} + \nu^2 \cdot \gamma^{l,l-k}
\end{aligned} \tag{15}$$

On the other hand, when updating $P^t_{l,l-k}$, it is important to account for the cases where $l$ or $l-k$ are 1 or $L$, since the values of the latent states are fixed in such cases. For instance, the update equation for the interaction with the input layer, $P^{t+1}_{l,1}$, can be expressed as follows:

$$\begin{aligned}
P^{t+1}_{l,1} &= \frac{1}{N} \mathbf{z}^{l,t+1^\top} M^{l-1:1} \mathbf{z}^{1,t+1} \\
&= \frac{1}{N} \left( \kappa \cdot \mathbf{z}^{l,t^\top} M^{l-1:1} \mathbf{z}^{1,t} + \nu \cdot \mathbf{z}^{l-1,t^\top} M^{l-2:1} \mathbf{z}^{1,t} \right. \\
&\quad \left. + \nu \cdot \mathbf{z}^{l+1,t^\top} M^{l:1} \mathbf{z}^{1,t} \right) \\
&= \kappa \cdot P^t_{l,1} + \nu \cdot P^t_{l-1,1} + \nu \cdot P^t_{l+1,1}
\end{aligned} \tag{16}$$

Furthermore, by considering the symmetry of $p$, we have $P^t_{1,l} = P^t_{l,1}$. Similarly, the update equation for the interaction with the output layer, $P^{t+1}_{L,L-k}$, is as follows:

$$\begin{aligned}
P^{t+1}_{L,L-k} &= \frac{1}{N} \mathbf{z}^{L,t^\top} M^{L-1:L-k} \mathbf{z}^{L-k,t+1} \\
&= \frac{1}{N} \left( \kappa \cdot \mathbf{z}^{L,t^\top} M^{L-1:L-k} \mathbf{z}^{L-k,t} \right. \\
&\quad + \nu \cdot \mathbf{z}^{L,t^\top} M^{L-1:L-k-1} \mathbf{z}^{L-k-1,t} \\
&\quad \left. + \nu \cdot \mathbf{z}^{L,t^\top} M^{L-1:L-k+1} \mathbf{z}^{L-k+1,t} \right) \\
&= \kappa \cdot P^t_{L,L-k} + \nu \cdot P^t_{L,L-k-1} + \nu \cdot P^t_{L,L-k+1}
\end{aligned} \tag{17}$$

Moreover, $P_{L,L-k} = P_{L-k,L}$.

We now aim to express the above update rules, which involve many combination terms, in a more concise matrix and vector calculation form. Let us carefully examine the structure of the update equations for both the latent states and $p$. The update equation for $\mathbf{z}$ can be divided into two parts. The first part is the sum of the element-wise product of the latent states $[\mathbf{z}_{l-1}, \mathbf{z}_l, \mathbf{z}_{l+1}]^\top$ and the coefficients $\mathbf{c}_z = [\nu, \kappa, \nu]^\top$. The second part is the sum of the element-wise product of the bias terms $[\mathbf{b}_{l-1}, \mathbf{b}_l]$ and the coefficients $\mathbf{c}_b = [\eta, -\nu]^\top$. The update equation for $p$, which is derived from the update equation of $\mathbf{z}$, can be seen as the outer product of the latent updates of layer $l$ and another layer $l-k$. The coefficients are fixed, and the values of $l$ and $l-k$ correspond to the indices in the $P$ matrix. Utilizing this, we can rewrite the update rule from Equations 15 to 17 in matrix

form as follows:

$$
\begin{aligned}
P_{l,l-k}^{t+1} &= \mathbf{c}_z^\top P_{l-1:l+1,l-k-1:l-k+1}^t \mathbf{c}_z \\
&\quad + \mathbf{c}_b^\top B_{l-k:l-k+1,l-1:l+1}^t \mathbf{c}_z \\
&\quad + \mathbf{c}_b^\top B_{l:l+1,l-k-1:l-k+1}^t \mathbf{c}_z \\
&\quad + \mathbf{c}_b^\top \Gamma_{l-1:l,l-k-1:l-k} \mathbf{c}_b, \\
P_{1,l}^{t+1^\top} &= P_{l,1}^{t+1} = \mathbf{c}_z^\top P_{l-1:l+1,1}^t, \text{ and} \\
P_{L,l}^{t+1^\top} &= P_{l,L}^{t+1} = \mathbf{c}_z^\top P_{l-1:l+1,L}^t,
\end{aligned}
\tag{18}
$$

for $1 < l, l - k < L$.

The update rules for $B$ represent the evolution of the interaction between the latent states $\mathbf{z}$ and the bias terms $\mathbf{b}$.

$$
\begin{aligned}
B_{l+1,l-k}^{t+1} &= \frac{1}{N} \mathbf{b}^{l^\top} M^{l:l-k} \mathbf{z}^{l-k,t+1} \\
&= \frac{1}{N} \mathbf{b}^{l^\top} M^{l:l-k} \Big( \kappa \cdot \mathbf{z}^{l-k,t} + \nu \cdot M^{l-k-1} \mathbf{z}^{l-k-1,t} \\
&\quad + \nu \cdot M^{l-k^\top} \mathbf{z}^{l-k+1,t} \\
&\quad + \eta \cdot \mathbf{b}^{l-k-1} - \nu \cdot M^{l-k^\top} \mathbf{b}^{l-k} \Big) \\
&= \frac{1}{N} \Big( \kappa \cdot \mathbf{b}_l^\top M_{l:l-k} \mathbf{z}_{l-k}^t \\
&\quad + \nu \cdot \mathbf{b}_l^\top M_{l:l-k} M_{l-k-1} \mathbf{z}_{l-k-1}^t \\
&\quad + \nu \cdot \mathbf{b}_l^\top M_{l:l-k} M_{l-k}^\top \mathbf{z}_{l-k+1}^t \\
&\quad + \eta \cdot \mathbf{b}_l^\top M_{l:l-k} \mathbf{b}_{l-k-1} \\
&\quad - \nu \cdot \mathbf{b}_l^\top M_{l:l-k} M_{l-k}^\top \mathbf{b}_{l-k} \Big) \\
&= \frac{1}{N} \Big( \kappa \cdot \mathbf{b}_l^\top M_{l:l-k} \mathbf{z}_{l-k}^t \\
&\quad + \nu \cdot \mathbf{b}_l^\top M_{l:l-k-1} \mathbf{z}_{l-k-1}^t \\
&\quad + \nu \cdot \mathbf{b}_l^\top M_{l:l-k+1} \mathbf{z}_{l-k+1}^t \\
&\quad + \eta \cdot \mathbf{b}_l^\top M_{l:l-k} \mathbf{b}_{l-k-1} \\
&\quad - \nu \cdot \mathbf{b}_l^\top M_{l:l-k+1} \mathbf{b}_{l-k} \Big) \\
&= \kappa \cdot B_{l+1,l-k}^t + \nu \cdot B_{l+1,l-k-1}^t + \nu \cdot B_{l+1,l-k+1}^t \\
&\quad + \eta \cdot \Gamma_{l,l-k-1} - \nu \cdot \Gamma_{l,l-k}
\end{aligned}
\tag{19}
$$

We can simplify the update rule for $B$ as follows:

$$
B_{l,l-k}^{t+1} = B_{l,l-k-1:l-k+1}^t \mathbf{c}_z + \mathbf{c}_b^\top \Gamma_{l-k-1:l-k,l}.
\tag{20}
$$

Note that since $\Gamma$ is a symmetric matrix, swapping the column and row indices in the update equation for $B$ does not alter the result.

## C.5 DYNAMICS OF LENGTHS

**Lengths of the Latent States** As mentioned earlier, the diagonal elements of $P^t$ represent the lengths of the latent states. That is, $p_l^t = P_{l,l}^t$.

**Lengths of the Prediction Errors**   Let the length of the PE at layer $l$ be denoted by $q_l^t$. We can express it as follows:

$$q_l^t = \langle (\delta_{i,l}^t)^2 \rangle$$
$$= \frac{1}{N} \|\delta_l^t\|^2$$
$$= \frac{1}{N} (\mathbf{z}_l^t - \hat{\mathbf{z}}_l^t)^\top (\mathbf{z}_l^t - \hat{\mathbf{z}}_l^t), \tag{21}$$

where $\hat{\mathbf{z}}_l^t = W_{l-1}\mathbf{z}_{l-1}^t + \mathbf{b}_{l-1}$. By substituting the prediction term, $q_l^t$ can be further expanded as follows:

$$q_l^t = \frac{1}{N}\left(\mathbf{z}_l^t - \sigma_w \cdot M_{l-1}\mathbf{z}_{l-1}^t - \mathbf{b}_{l-1}\right)^\top$$
$$\left(\mathbf{z}_l^t - \sigma_w \cdot M_{l-1}\mathbf{z}_{l-1}^t - \mathbf{b}_{l-1}\right)$$
$$= \frac{1}{N}\left(\mathbf{z}_l^{t\top} - \sigma_w \cdot \mathbf{z}_{l-1}^{t\top}M_{l-1}^\top - \mathbf{b}_{l-1}^\top\right)$$
$$\left(\mathbf{z}_l^t - \sigma_w \cdot M_{l-1}\mathbf{z}_{l-1}^t - \mathbf{b}_{l-1}\right)$$
$$= \frac{1}{N}\left(\mathbf{z}^{l,t\top}(\mathbf{z}^{l,t} - \sigma_w \cdot M^{l-1}\mathbf{z}^{l-1,t} - \mathbf{b}^{l-1})\right.$$
$$- \sigma_w \cdot \mathbf{z}^{l-1,t\top}M^{l-1\top}(\mathbf{z}^{l,t} - \sigma_w \cdot M^{l-1}\mathbf{z}^{l-1,t} - \mathbf{b}^{l-1})$$
$$\left. - \mathbf{b}_{l-1}^\top(\mathbf{z}_l^t - \sigma_w \cdot M_{l-1}\mathbf{z}_{l-1}^t - \mathbf{b}_{l-1})\right)$$
$$= \frac{1}{N}\left(\mathbf{z}_l^{t\top}\mathbf{z}_l^t - \sigma_w \cdot \mathbf{z}_l^{t\top}M_{l-1}\mathbf{z}_{l-1}^t - \mathbf{z}_l^{t\top}\mathbf{b}_{l-1}\right.$$
$$- \sigma_w \cdot \mathbf{z}_{l-1}^{t\top}M_{l-1}^\top\mathbf{z}_l^t$$
$$+ \sigma_w^2 \cdot \mathbf{z}_{l-1}^{t\top}M_{l-1}^\top M_{l-1}\mathbf{z}_{l-1}^t$$
$$+ \sigma_w \cdot \mathbf{z}_{l-1}^{t\top}M_{l-1}^\top\mathbf{b}_{l-1}$$
$$\left. - \mathbf{b}_{l-1}^\top\mathbf{z}_l^t + \sigma_w \cdot \mathbf{b}_{l-1}^\top M_{l-1}\mathbf{z}_{l-1}^t + \mathbf{b}_{l-1}^\top\mathbf{b}_{l-1}\right)$$
$$= P_{l,l}^t - \sigma_w \cdot P_{l,l-1}^t - B_{l,l}^t$$
$$- \sigma_w \cdot P_{l-1,l}^t + \sigma_w^2 \cdot P_{l-1,l-1}^t + \sigma_w \cdot B_{l,l-1}^t$$
$$- B_{l,l}^t + \sigma_w \cdot B_{l,l-1}^t + \gamma_{l-1,l-1} \tag{22}$$

The above equation can be simplified as:

$$q_l^t = \mathbf{c}_q^\top P_{l-1:l,l-1:l}^t\mathbf{c}_q - 2B_{l,l-1:l}^t\mathbf{c}_q + \Gamma_{l-1,l-1}, \tag{23}$$

where the coefficient $\mathbf{c}_q = [-\sigma_w, 1]^\top$.

**Lengths of Weight Updates**   The length of the weight updates at layer $l$ is denoted by $r_l^t$, and is defined as:

$$r_l^t = \frac{1}{N^2}\|\Delta W_l^t\|_F^2, \tag{24}$$

where $\Delta W_l^t = \delta_{l+1}^t \mathbf{z}_l^{t\top}$, with $\delta_{l+1}^t$ representing the error signal at the next layer and $\mathbf{z}_l^t$ being the signal at the current layer.

Before proceeding further, we prove a simple lemma for the Frobenius norm:

**Lemma (Horn & Johnson, 2012):** $\|\mathbf{x}\mathbf{y}^\top\|_F^2 = \|\mathbf{x}\|^2 \cdot \|\mathbf{y}\|^2$, where $\mathbf{x}$ and $\mathbf{y}$ are vectors.

Proof:

$$\|\mathbf{x}\mathbf{y}^\top\|_F^2 = \sum_{i,j}(\mathbf{x}\mathbf{y}^\top)_{i,j}^2 = \sum_{i,j}(\mathbf{x}_i\mathbf{y}_j)^2 = \sum_i \mathbf{x}_i^2 \sum_j \mathbf{y}_j^2$$
$$= \|\mathbf{x}\|^2 \cdot \|\mathbf{y}\|^2. \tag{25}$$

Using the above lemma for the Frobenius norm, we can simplify $r_l^t$ as:

$$r_l^t = \frac{1}{N^2}\|\delta_{l+1}^t \mathbf{z}_l^{t\top}\|_F^2 = \frac{1}{N^2}\|\delta_{l+1}^t\|^2 \cdot \|\mathbf{z}_l^t\|^2$$
$$= q_{l+1}^t \cdot p_l^t \tag{26}$$

Since $\frac{1}{N}\|\Delta\mathbf{b}_l\|^2 = \frac{1}{N}\|\delta_l^t\|^2$, the length of the bias update is equivalent to $q_l^t$ and is therefore omitted.

## D    META-PCN TRAINING ALGORITHM

Algorithm 1 presents the complete training procedure for Meta-PCN. The algorithm consists of three main phases per mini-batch: feed-forward initialization, inference, and learning, followed by weight normalization.

**Feed-Forward Initialization (Lines 4–9).**    The input is set to the first layer (Line 4), and feed-forward predictions $\mathbf{c}_l$ are computed and frozen for subsequent use during inference (Lines 5–7). Latent states are initialized to these predictions. The output prediction $\mathbf{c}_L$ is computed (Line 8), and the output layer is set to the target (Line 9).

**Inference Phase (Lines 10–18).**    For $T$ iterations, the meta-prediction errors $\tilde{\boldsymbol{\delta}}_l^{(t)} = \mathbf{z}_l^{(t)} - \mathbf{c}_l$ are computed (Lines 11–13). The meta-PE objective $\mathcal{J}$ is then evaluated (Line 14), where $\tilde{\boldsymbol{\delta}}_{l+1}^{(t)}$ is treated as a constant (stop gradient) to serve as a stabilized error signal. Latent states are updated by gradient descent on $\mathcal{J}$ (Lines 15–17).

**Learning Phase (Lines 19–20).**    After inference converges, the parameter loss $\mathcal{L}(\theta)$ is computed based on the final meta-prediction errors $\tilde{\boldsymbol{\delta}}_l^{(T)}$ (Line 19). Parameters are updated using the optimizer (Line 20).

**Weight Normalization (Lines 21–24).**    After each parameter update, variance-based weight normalization is applied to maintain spectral control (Lines 21–24).

## E    EXPERIMENTAL SETUP

**Length Dynamics Analysis.**    The simulations described in Section 3 analyzed the length dynamics of latent states and PEs during the inference process in a random PCN ensemble. The dataset consisted of samples from a random unit Gaussian distribution $((x_i, y_i) \sim \mathcal{N}(\mathbf{0}, I))$. The dataset contained 128 samples processed in a single batch. The number of inference steps $(T)$ was mainly set to 2000 to track the iterative changes in length dynamics. The inference rate was set at 0.05. The model consisted of 30 layers to effectively show the exponential growth in PCNs. The latent dimension was set to 100.

**Baseline Comparisons.**    We compare three approaches under identical experimental conditions: (1) Standard backpropagation (BP), (2) Conventional PCN with only feed-forward initialization, and (3) The complete Meta-PCN framework. All methods use the same network architectures, optimization settings, and training procedures, differing only in their learning algorithms.

**Meta-PCN Components.**    Our framework is built on two core components: (1) **Meta-PE objective** with feed-forward initialization and (2) **Variance-based weight regularization**. In addition, all experiments include **blocked sweep updates** (Gauss-Seidel-like alternating layer updates), a theoretically motivated technique whose empirical contribution is analyzed in the ablation study (Appendix I).

**Training Configuration.**    We train all models for 50 epochs using the AdamW (Loshchilov & Hutter, 2019) optimizer with a learning rate of 0.0001 and a weight decay of 0.0005. The batch size

---

**Algorithm 1** Meta-PCN Training

---

**Input:** Dataset $\mathcal{D} = \{(\mathbf{x}, \mathbf{y})\}$, network with $L$ layers, parameters $\theta = \{W_l, \mathbf{b}_l\}$, inference steps $T$, inference rate $\eta$, learning rate $\alpha$, epochs $E$
**Output:** Trained parameters $\theta$

1. Initialize $\theta$ randomly

2. **for** epoch $= 1$ to $E$ **do**

3.     **for** each mini-batch $(\mathbf{x}, \mathbf{y}) \in \mathcal{D}$ **do**

4.         $\mathbf{z}_1 \leftarrow \mathbf{x}$                                                                 $\triangleright$ Input

5.         **for** $l = 2$ to $L - 1$ **do**

6.           $\mathbf{c}_l \leftarrow \phi(W_{l-1}\mathbf{z}_{l-1} + \mathbf{b}_{l-1}); \quad \mathbf{z}_l^{(0)} \leftarrow \mathbf{c}_l$                    $\triangleright$ Frozen prediction

7.         **end for**

8.         $\mathbf{c}_L \leftarrow \phi(W_{L-1}\mathbf{z}_{L-1} + \mathbf{b}_{L-1})$                         $\triangleright$ Output prediction

9.         $\mathbf{z}_L \leftarrow \mathbf{y}$                                                           $\triangleright$ Target

10.         **for** $t = 0$ to $T - 1$ **do**                                 $\triangleright$ Inference phase

11.           **for** $l = 2$ to $L$ **do**

12.              $\tilde{\boldsymbol{\delta}}_l^{(t)} \leftarrow \mathbf{z}_l^{(t)} - \mathbf{c}_l$

13.           **end for**

14.           Compute $\mathcal{J} = \frac{1}{2} \sum_{l=2}^{L-1} \|\tilde{\boldsymbol{\delta}}_l^{(t)} - g_l(\tilde{\boldsymbol{\delta}}_{l+1}^{(t)})\|^2$               $\triangleright$ stop gradient on $\tilde{\boldsymbol{\delta}}_{l+1}$

15.           **for** $l = 2$ to $L - 1$ **do**

16.              $\mathbf{z}_l^{(t+1)} \leftarrow \mathbf{z}_l^{(t)} - \eta \nabla_{\mathbf{z}_l} \mathcal{J}$

17.           **end for**

18.         **end for**

19.         Compute $\mathcal{L}(\theta) = \frac{1}{2} \sum_{l=2}^{L-1} \|\tilde{\boldsymbol{\delta}}_l^{(T)}\|^2$                      $\triangleright$ Learning phase

20.         $\theta \leftarrow \theta - \alpha \nabla_\theta \mathcal{L}(\theta)$

21.         **for** $l = 1$ to $L - 1$ **do**                                  $\triangleright$ Weight normalization

22.           $\sigma_w \leftarrow \sqrt{\text{Var}(W_l)}$

23.           $W_l \leftarrow W_l / ((\sqrt{m} + \sqrt{n}) \cdot \sigma_w)$

24.         **end for**

25.     **end for**

26. **end for**

27. **return** $\theta$

---

is set to 128 across all experiments to ensure fair comparison. For PC-specific parameters, we use an inference rate ($\eta$) of 0.2 and perform 20 inference steps ($T$) per learning step. For classification tasks, we replace the squared error at the output layer with a cross-entropy loss.

**Implementation Details.** All experiments were conducted using the PyTorch (Paszke et al., 2017) framework on CUDA-enabled GPUs rented from Vast.ai. Each experiment is repeated 5 times with different random seeds to ensure statistical reliability.

**Comparison with Prior PCN Configurations.** Our PCN baseline differs from the configuration reported in Pinchetti et al. (2024) in three key aspects: (i) network architecture (standard VGG with ReLU vs. custom VGG with GELU), (ii) inference rate ($\eta = 0.05$ vs. $\eta = 0.01$), and (iii) weight decay ($5 \times 10^{-4}$ vs. 0). A controlled study varying these factors on CIFAR-10 shows that accuracy can range from below 20% to above 70% depending on these choices, with high variance across runs (std up to 8.99%), indicating unstable training dynamics inherent to PCNs. Our primary focus is on stability and learning dynamics under a controlled, standardized setup, rather than on maximizing absolute baseline performance through method-specific hyperparameter tuning. A detailed comparison is provided in Table 1.

Table 1: Controlled comparison of PCN configurations on CIFAR-10. "pcx vgg7" refers to our implementation of the VGG7-style architecture from Pinchetti et al. (2024); "ours vgg7" is the standard VGG7 used in our main experiments. Results are mean $\pm$ std over three runs.

| Architecture | $\eta$ | Weight Decay | Test Acc. (%) |
|---|---|---|---|
| pcx vgg7 | 0.01 | 0.0 | $64.88 \pm 2.48$ |
| ours vgg7 | 0.01 | 0.0 | $40.48 \pm 2.44$ |
| pcx vgg7 | 0.01 | $5 \times 10^{-4}$ | $63.70 \pm 4.50$ |
| ours vgg7 | 0.01 | $5 \times 10^{-4}$ | $41.84 \pm 0.77$ |
| pcx vgg7 | 0.05 | 0.0 | $73.37 \pm 8.99$ |
| ours vgg7 | 0.05 | 0.0 | $16.33 \pm 4.18$ |
| pcx vgg7 | 0.05 | $5 \times 10^{-4}$ | $61.82 \pm 3.48$ |
| ours vgg7 | 0.05 | $5 \times 10^{-4}$ | $17.85 \pm 6.95$ |

## F    PARAMETER LEARNING IN META-PCN

**Separation of Inference and Parameter Objectives.** While the meta-PE objective $\mathcal{J}$ governs the inference dynamics over latent states, parameter updates follow the conventional PC loss:

$$\mathcal{L}(\theta) = \frac{1}{2} \sum_{l=2}^{L-1} \|\mathbf{z}_l^{(T)} - f_{l-1}(\mathbf{z}_{l-1}^{(0)}; \theta)\|_2^2. \tag{27}$$

We do not directly use the meta-PE objective for parameter updates for the following reasons. The first term $\tilde{\delta}_l$ in the meta-PE loss is subject to prediction freezing and thus does not directly involve parameters. To backpropagate through $g_l$ in the second term, one would need to compute second-order derivatives, which we explicitly avoid due to complexity and computational cost. Instead, the meta-objective reshapes the inference dynamics so that the resulting latent states and error signals are better aligned with the desired theoretical properties.

**Empirical Convergence.** Figure 8 shows that the training loss $\mathcal{L}(\theta)$ decreases smoothly during training, demonstrating stable convergence without pathological behavior.

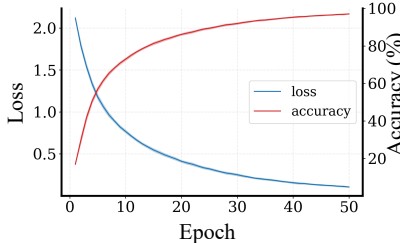

Figure 8: Training loss curves for Meta-PCN on CIFAR-10 with VGG-13 over 5 runs.

## G    COMPUTATIONAL OVERHEAD ANALYSIS

We empirically measured the computational overhead of Meta-PCN compared to conventional PCN. Using a representative configuration (batch size 1, $T = 20$ inference steps, averaged over 100 runs), we obtained:

- Conventional PCN: $127.930 \pm 0.637$ ms per inference step
- Meta-PCN: $129.821 \pm 0.622$ ms per inference step

This corresponds to a relative overhead of approximately **1.48%** per inference step.

Table 2: CIFAR-10 Top-1 accuracy (%): mean±std over $n=5$ runs for each method, with $p$-values from two-sided Mann-Whitney U tests. $p_1$: Meta-PCN vs BP; $p_2$: Meta-PCN vs PCN. **Bold** denotes the best mean accuracy per architecture. *Italic* $p$-values indicate statistical significance ($p \leq 0.05$).

| Architecture | BP | PCN | Meta-PCN | $p_1$ | $p_2$ |
|---|---|---|---|---|---|
| VGG-5 | 83.91±0.24 | 25.78±1.78 | **85.64±0.22** | *0.0079* | *0.0079* |
| VGG-7 | 88.09±0.47 | 20.01±3.44 | **88.99±0.40** | *0.0317* | *0.0079* |
| VGG-9 | 84.84±0.28 | 19.43±4.83 | **85.69±0.29** | *0.0079* | *0.0079* |
| VGG-11 | 84.50±0.50 | 17.96±3.23 | **85.04±0.23** | 0.0556 | *0.0079* |
| VGG-13 | 87.85±0.50 | 12.24±1.50 | **89.53±0.47** | *0.0079* | *0.0079* |
| ResNet-18 | 82.54±0.35 | 54.53±0.79 | **82.58±0.12** | 1.0000 | *0.0079* |

Table 3: CIFAR-100 Top-1 accuracy (%). Same format as Table 2.

| Architecture | BP | PCN | Meta-PCN | $p_1$ | $p_2$ |
|---|---|---|---|---|---|
| VGG-5 | 51.73±0.24 | 4.14±0.30 | **55.18±0.23** | *0.0079* | *0.0079* |
| VGG-7 | 51.52±0.82 | 7.01±1.22 | **55.49±0.49** | *0.0079* | *0.0079* |
| VGG-9 | 45.22±0.34 | 3.48±1.22 | **50.09±0.92** | *0.0079* | *0.0079* |
| VGG-11 | 51.13±0.88 | 1.32±0.76 | **52.21±0.48** | 0.1508 | *0.0079* |
| VGG-13 | 54.19±2.03 | 1.26±0.17 | **59.04±1.52** | *0.0079* | *0.0079* |
| ResNet-18 | **53.16±0.43** | 38.66±0.49 | 53.14±0.30 | 0.8413 | *0.0079* |

**Component-wise Analysis.** The overhead is attributable to the two core components:

- **Meta-objective computation:** The gradients of the meta-PE objective with respect to latent states can be efficiently computed in a single backward pass using automatic differentiation. The overhead is negligible compared to the overall inference cost.
- **Weight normalization:** Computing the variance of each weight tensor is substantially cheaper than spectral norm computation and benefits from GPU parallel reduction.

Additionally, blocked sweep updates are included in the Meta-PCN configuration but introduce no measurable overhead, as our implementation follows existing PC frameworks that do not parallelize updates across modules.

## H STATISTICAL SIGNIFICANCE TESTING

We performed Mann-Whitney U tests to verify whether Meta-PCN's performance improvements are statistically significant. This non-parametric test is suitable for comparing performance values as it does not assume normality of accuracy distributions. All comparisons use $n=5$ runs per method.

### H.1 MAIN METHOD COMPARISON: BP VS PCN VS META-PCN

The statistical analysis reveals several key findings across all datasets. Meta-PCN demonstrates statistically significant improvements over conventional PCNs in all architecture and dataset combinations. Against backpropagation, Meta-PCN shows statistically significant improvements in most VGG architectures, with the performance gap generally decreasing as architecture complexity increases. Notably, ResNet-18 shows minimal or non-significant differences compared to backpropagation.

The consistent pattern across Top-1 and Top-5 metrics on CIFAR-100 and TinyImageNet further validates Meta-PCN's robustness. The larger improvements observed in conventional PCN comparisons highlight the severity of the pathologies addressed by our framework, while the competitive performance against backpropagation supports the practical viability of locally learned algorithms.

**Conclusion**: The comprehensive statistical analysis provides strong evidence that Meta-PCN's performance improvements represent systematic rather than accidental gains. The framework successfully bridges the gap between local learning and competitive task performance, positioning predictive coding as a promising alternative to backpropagation-based learning.

Table 4: CIFAR-100 Top-5 accuracy (%). Same format as Table 2.

| Architecture | BP | PCN | Meta-PCN | $p_1$ | $p_2$ |
|---|---|---|---|---|---|
| VGG-5 | 79.71±0.26 | 14.83±0.13 | **81.94±0.24** | *0.0079* | *0.0079* |
| VGG-7 | 78.43±0.74 | 22.18±2.05 | **81.92±0.31** | *0.0079* | *0.0079* |
| VGG-9 | 75.71±0.28 | 13.18±3.84 | **79.15±0.81** | *0.0119* | *0.0079* |
| VGG-11 | 78.68±0.49 | 6.57±2.04 | **79.29±0.34** | 0.0952 | *0.0079* |
| VGG-13 | 81.06±1.81 | 5.70±0.59 | **85.12±1.49** | *0.0159* | *0.0079* |
| ResNet-18 | 79.81±0.19 | 68.29±0.44 | **80.36±0.17** | *0.0159* | *0.0079* |

Table 5: TinyImageNet Top-1 accuracy (%). Same format as Table 2.

| Architecture | BP | PCN | Meta-PCN | $p_1$ | $p_2$ |
|---|---|---|---|---|---|
| VGG-5 | 35.11±0.25 | 1.31±0.08 | **38.91±0.42** | *0.0079* | *0.0119* |
| VGG-7 | 39.35±0.33 | 0.90±0.05 | **42.28±0.51** | *0.0079* | *0.0079* |
| VGG-9 | 35.99±0.59 | 0.77±0.30 | **39.35±0.88** | *0.0079* | *0.0079* |
| VGG-11 | 38.06±0.42 | 0.72±0.28 | **39.18±0.35** | *0.0079* | *0.0079* |
| VGG-13 | 45.38±0.81 | 3.49±2.73 | **47.43±0.30** | *0.0079* | *0.0119* |
| ResNet-18 | 43.80±0.32 | 17.73±0.79 | **44.19±0.41** | 0.1508 | *0.0079* |

# I    ABLATION STUDY

## I.1    ABLATION STUDY ON META-PCN COMPONENTS

We conducted a systematic ablation study to quantitatively evaluate the contribution of each component in the Meta-PCN framework. Table 7 presents results from training on the CIFAR-10 dataset using the VGG-13 architecture for 50 epochs.

## I.2    COMPONENT ANALYSIS

The ablation results reveal that each Meta-PCN component contributes differently. Removing the meta-PE objective causes catastrophic performance degradation, dropping accuracy from 89.53% to 10.01% (79.52% decrease). This dramatic decline suggests that the standard free-energy objective suffers from severe pathologies, such as PE imbalance and EVPE, which prevent effective learning.

Removing weight regularization (normalization) leads to a 1.28% decrease (88.25% vs 89.53%), with statistical significance ($p = 0.0079$). This moderate but significant degradation indicates that variance control effectively addresses EVPE and contributes to training stability.

Removing blocked sweep updates shows minimal impact, with only a 0.20% decrease (89.33% vs 89.53%) and no statistical significance ($p = 1.0000$). While the convergence improvement from Gauss-Seidel-like updates appears modest in this configuration, the benefit may become more pronounced in deeper architectures.

## I.3    STATISTICAL SIGNIFICANCE AND COMPONENT RANKING

The Mann-Whitney U test results establish a clear hierarchy of component importance. The meta-PE objective is the most critical component, as its removal renders the model non-functional. Weight regularization provides statistically significant but moderate improvements, while blocked sweep updates show non-significant effects under current experimental conditions.

**Component Criticality Ranking**:

1. **Meta-PE Objective** (most critical): Essential for basic functionality (79.52% decrease upon removal)

2. **Weight Regularization** (moderately critical): Statistically significant (1.28% decrease upon removal, p = 0.0079)

3. **Blocked Sweep GS** (least critical): Non-significant (0.20% decrease upon removal, p = 1.0000)

Table 6: TinyImageNet Top-5 accuracy (%). Same format as Table 2.

| Architecture | BP | PCN | Meta-PCN | $p_1$ | $p_2$ |
|---|---|---|---|---|---|
| VGG-5 | 60.71±0.49 | 5.22±0.15 | **65.08±0.43** | *0.0079* | *0.0079* |
| VGG-7 | 64.67±0.40 | 3.35±0.28 | **67.79±0.91** | *0.0079* | *0.0119* |
| VGG-9 | 62.98±0.31 | 3.29±1.15 | **66.58±0.59** | *0.0079* | *0.0079* |
| VGG-11 | 65.79±0.73 | 2.95±0.57 | **66.24±0.45** | *0.3095* | *0.0079* |
| VGG-13 | 71.24±0.44 | 8.83±5.79 | **71.62±0.36** | *0.3095* | *0.0119* |
| ResNet-18 | 69.55±0.22 | 38.97±1.38 | **70.02±0.15** | *0.0159* | *0.0079* |

Table 7: Ablation study results showing the contribution of each Meta-PCN component. Performance is measured on CIFAR-10 using VGG-13 architecture across 5 independent runs. Statistical significance is evaluated using Mann-Whitney U tests against the full Meta-PCN framework.

| Method | Accuracy (%) | p-value |
|---|---|---|
| Meta-PCN | $89.53 \pm 0.47$ | – |
| – Blocked Sweep GS | $89.33 \pm 0.56$ | 1.0000 |
| – Normalization | $88.25 \pm 0.32$ | **0.0079** |
| – Meta-PE Objective | $10.01 \pm 0.02$ | **0.0167** |
| PCN (conventional) | $10.01 \pm 0.02$ | **0.0079** |
| + Prediction Freezing | $16.31 \pm 1.00$ | **0.0159** |
| + Normalization | $10.16 \pm 0.33$ | **0.0095** |

These results demonstrate that the meta-PE objective is fundamental to Meta-PCN's success, while weight regularization provides important stability benefits. The blocked sweep component, while theoretically motivated (Appendix I.5), showed no statistically significant effect in our experiments. Accordingly, the main text focuses on the two core components—the meta-PE objective and weight regularization—as the primary contributions. Nevertheless, all reported Meta-PCN results include blocked sweep updates to faithfully reflect the complete experimental configuration.

## I.4 Effect of Adding Individual Components to Conventional PCN

The lower half of Table 7 examines the effect of adding individual Meta-PCN components to conventional PCN. There are two conceptually distinct uses of feed-forward predictions in our work:

1. **Latent initialization via feed-forward prediction.** This is standard practice in the PC literature: feed-forward predictions initialize latent states $\mathbf{z}$. All variants in our ablation—conventional PC, Meta-PCN, and ablations—use this initialization.

2. **Freezing feed-forward predictions during inference.** This is our proposal: predictions are treated as fixed values $\mathbf{c}_l$ throughout inference, which is built into the meta-objective definition. Thus, freezing is inherently tied to using the meta-objective.

The "+ Prediction Freezing" row shows the effect of freezing predictions without the meta-objective, yielding the loss $\mathcal{J}(\tilde{\boldsymbol{\delta}}) = \frac{1}{2} \sum_{l=2}^{L-1} \|\mathbf{z}_l - \mathbf{c}_l\|_2^2$. Since $\mathbf{z}_l$ is initialized to $\mathbf{c}_l$, this setup produces minimal updates during inference. This variant achieved 16.31% accuracy, slightly better than conventional PCN (10.01%) but far below Meta-PCN (89.53%).

The "+ Normalization" row (10.16%) shows that weight regularization alone cannot resolve the structural issues. These results confirm that the meta-objective and normalization act complementarily: the former reshapes the objective to target the equilibrium residual, while the latter controls the spectral factors that drive EVPE.

## I.5 Blocked Sweep Updates for PC Inference

**Convergence Analysis through Classical Iterative Methods.** To understand the convergence limitations of PC inference, we frame the free-energy minimization problem in inference within the

context of classical iterative methods. PC inference can be viewed as solving the nonlinear stationarity system $\mathbf{F}(\mathbf{z}) = \nabla_{\mathbf{z}}\mathcal{F}(\mathbf{z}) = 0$ subject to boundary conditions $\mathbf{z}_1 = \mathbf{x}$ and $\mathbf{z}_L = \mathbf{y}$. The standard inference procedure follows the fixed-point iteration $\mathbf{z}^{t+1} = G_\eta(\mathbf{z}^t)$, where an equilibrium point $\mathbf{z}^*$ satisfies $\mathbf{z}^* = G_\eta(\mathbf{z}^*)$ if and only if $\nabla_{\mathbf{z}}\mathcal{F}(\mathbf{z}^*) = 0$.

This formulation reveals that the PCN inference performs simultaneous updates of all latent states, making it directly analogous to the Jacobi method in classical iterative solvers (Saad, 2003; Golub & Loan, 2013). This analogy provides crucial insights into the inherent limitations of PC inference, as the Jacobi method is well-known for its stringent convergence requirements and inefficient information propagation across network depth.

For linear systems $A\mathbf{u} = \mathbf{b}$, Jacobi convergence typically requires both $A$ and $2D - A$ to be symmetric positive definite (SPD), where $D$ denotes the diagonal of $A$. In contrast, the Gauss-Seidel method requires only that $A$ be SPD (see, e.g., Varga, 2009; Saad, 2003). More importantly, for consistently ordered SPD matrices, the celebrated convergence rate relationship $\rho(\text{GS}) = \rho(\text{Jacobi})^2$ demonstrates that Gauss-Seidel achieves asymptotically quadratic convergence improvement over Jacobi (Young, 1971; Saad, 2003; Varga, 2009), where $\rho(\cdot)$ denotes the spectral radius of the iteration matrix.

The practical implications for deep PCNs are significant. Because the standard PC update behaves analogously to the Jacobi method, its convergence in deep networks suffers from inherent inefficiencies: information propagates exclusively through simultaneous, layer-wise exchanges, resulting in slow convergence and potential divergence on ill-conditioned problem instances.

**PC-Compatible Blocked Sweep Method.** Standard PC inference employs simultaneous (Jacobi-like) updates of the form: $\mathbf{z}^{t+1} = \mathbf{z}^t - \eta\,\nabla_{\mathbf{z}}\mathcal{F}(\mathbf{z}^t)$, which updates all layers using neighbor information from iteration $t$. While this approach offers high parallelizability, it propagates information slowly across network depth, contributing to the convergence inefficiencies.

In contrast, Gauss-Seidel (GS)-type schemes achieve faster convergence by reusing the most recently computed values within the same iteration, though at the cost of reduced parallelism. Our blocked sweep update strategy offers a principled compromise that preserves the locality and modularity inherent to predictive coding, while leveraging newly computed neighbor states to accelerate information propagation.

**Theoretical Foundation and Convergence Analysis.** The blocked sweep method possesses a solid theoretical foundation rooted in classical iterative solver theory. The blocked sweep update is mathematically equivalent to a preconditioned gradient step $\mathbf{z}^{t+1} = \mathbf{z}^t - \eta\,\mathbf{M}^{-1}\nabla_{\mathbf{z}}\mathcal{F}(\mathbf{z}^t)$, where the preconditioner $\mathbf{M} = \mathbf{D} + \mathbf{L}$ has a block lower-triangular structure (with $\mathbf{D}$ representing the diagonal blocks and $\mathbf{L}$ the strictly lower block components). This corresponds precisely to a Gauss-Seidel iteration applied to the linearized system.

The convergence advantage is quantified through spectral analysis of the resulting iteration matrix $\mathbf{T}_\eta^{\text{BS}} = \mathbf{I} - \eta\,\mathbf{M}^{-1}\mathbf{H}_*$. For consistently ordered SPD problems, the classical result $\rho(\mathbf{T}_\eta^{\text{GS}}) = \rho(\mathbf{T}_\eta^{\text{Jac}})^2$ (Young, 1971; Saad, 2003; Varga, 2009) demonstrates that blocked sweeps achieve asymptotically quadratic convergence acceleration compared to simultaneous Jacobi-like updates. The method thus provides a theoretically motivated approach to address the depth-induced convergence slowdown, though its empirical impact may vary across configurations (see Section I).

## J   Weight Normalization for Backpropagation

We examined whether our variance-based normalization benefits backpropagation (BP). On CIFAR-10 with VGG-13, standard BP achieves $87.85 \pm 0.50\%$ accuracy, while BP with our normalization achieves $89.07 \pm 0.46\%$. Although normalization provides a modest improvement for BP, the resulting accuracy remains below Meta-PCN ($89.53 \pm 0.47\%$), indicating that the full Meta-PCN framework offers benefits beyond weight regularization alone.

## K   Length Dynamics with Nonlinear Activations

Figure 9 explores the dynamics of latent state lengths ($p^{l,t}$), PE lengths ($q^{l,t}$), and weight update lengths ($r^{l,t}$) in nonlinear PCNs across different activation functions. The analysis focuses on common nonlinearity types such as `tanh`, `ReLU`, `SELU`, and `SiLU`, each applied to layers with varying

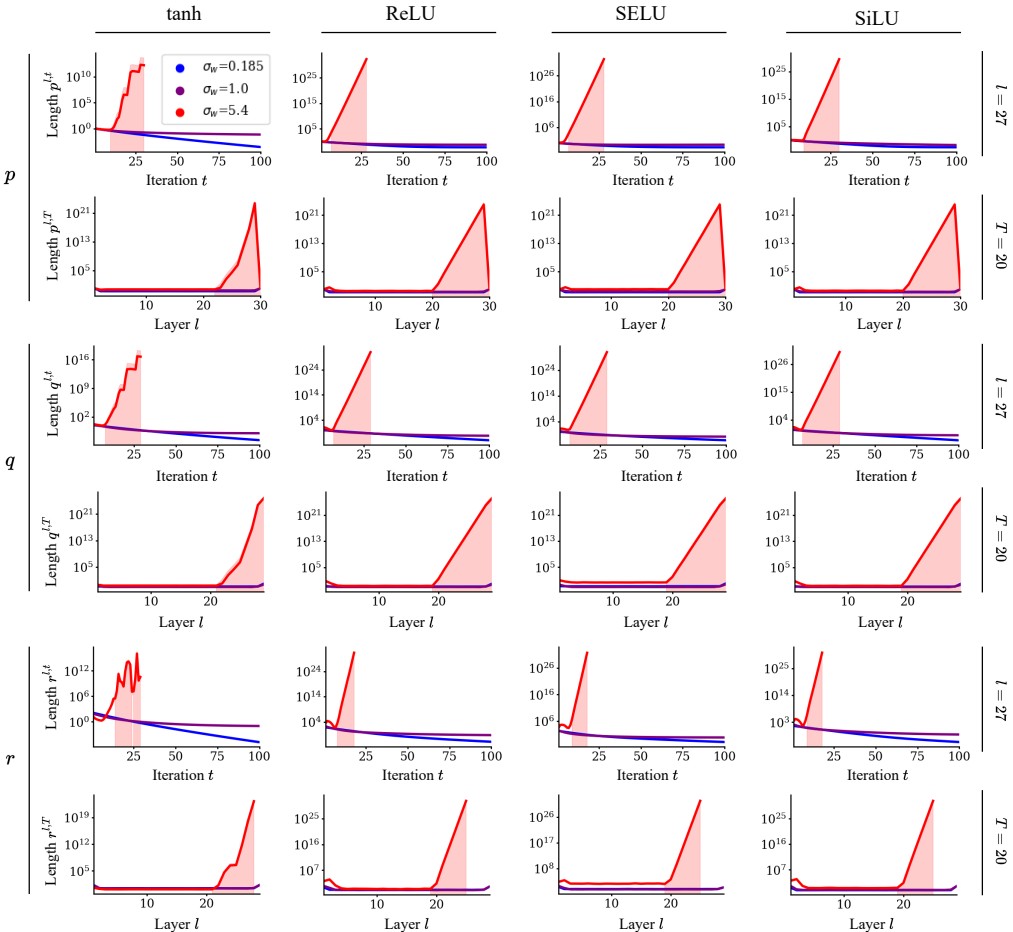

Figure 9: The dynamics of $p^{l,t}$, $q^{l,t}$, and $r^{l,t}$ and their layer-wise results for nonlinear PCNs. In all subfigures, the results are shown for the cases of $\sigma_w \in \{0.185, 1.0, 5.4\}$ with different colors, and $\sigma_b = 0.1$. Settings not mentioned or indicated are identical to those in Figure 2. Each column represents the applied nonlinear function. The odd rows are the dynamics of $p$, $q$, and $r$, respectively ($l = 27$). The even rows are the layer-wise distribution of $p$, $q$, and $r$, respectively ($T = 20$).

weight variances ($\sigma_w$). The results show that the dynamics for $p$, $q$, and $r$ are highly sensitive to $\sigma_w$, even under nonlinear activations. The odd rows depict the temporal evolution of $p$, $q$, and $r$ at layer $l = 27$, while the even rows display the layer-wise distribution of these values at the $T = 20$ inference step. These subfigures illustrate two key phenomena that occur regardless of the applied nonlinear activation function:

1. In the odd rows, we observe that even with nonlinearity applied, $p$, $q$, and $r$ exhibit exponential growth near the output layer when $\sigma_w$ is large (e.g., $\sigma_w = 5.4$). This suggests that while nonlinear activations are typically expected to provide some degree of constraint on the predicted latent state dynamics by squashing the outputs (e.g., `tanh`), the latent state length growth persists for larger $\sigma_w$. This pattern holds across all activation functions examined, indicating that nonlinearity alone is insufficient to counteract the destabilizing effects of high weight variance.

2. The even rows reveal that these exponential growth patterns can emerge early in the inference phase, even at $T = 20$, particularly in deeper layers. The layer-wise distributions of $p$, $q$, and $r$ show that the effects of large $\sigma_w$ extend throughout the network, with PEs ($q$) and weight updates ($r$) becoming increasingly concentrated toward the output layer. This observation underscores a key challenge in training deep PCNs with nonlinearity. While early inference stages may seem stable, instability can rapidly accumulate in deeper layers due to the interplay between nonlinearity and large weight variances.

Importantly, this analysis highlights the need for regularization strategies, even in networks with nonlinear activations. The exponential growth seen here mirrors the behavior in linear PCNs, suggesting that length regularization and weight variance control are critical to preventing runaway dynamics in both linear and nonlinear architectures. Regularization techniques, such as those introduced in Meta-PCN, become essential for maintaining stability, particularly when nonlinearity alone is insufficient to prevent the excessive growth of latent states and PEs.

## L    ADDITIONAL RESULTS

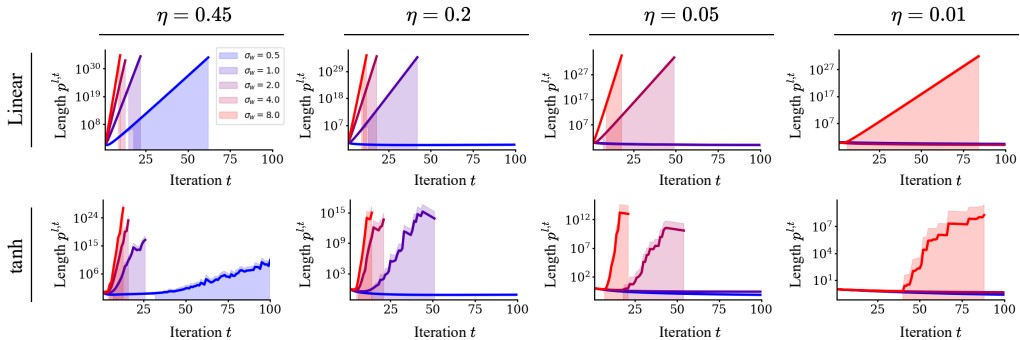

Figure 10: The dynamics of $p^{l,t}$ for PCNs ($l = 26$ and $L = 30$). *Settings:* In all subfigures, the results are shown for the cases of $\sigma_w \in \{0.5, 1.0, 2.0, 4.0, 8.0\}$ with different colors, and $\sigma_b = 0.3$. Settings not mentioned or indicated are identical to those in Figure 3. *Subfigures:* (a)-(d) Dynamics of $p^{l,t}$ of linear PCNs over the 100 inference steps. (e)-(h) Dynamics of $p^{l,t}$ of nonlinear PCNs (tanh) over the 100 inference steps.

Figure 10 shows the dynamics of the latent state lengths $p^{l,t}$ for PCNs across varying $\sigma_w$ values. The results demonstrate how the network's stability depends heavily on the initialization of the weights and inference rate. In both the linear and nonlinear PCN cases, we observe that as $\sigma_w$ or $\eta$ increases, the system becomes more prone to instability, with the exponential growth of the latent state lengths becoming apparent. This is especially visible for higher values of $\sigma_w$ (e.g., 8.0), where the growth accelerates drastically. This behavior aligns with the theoretical predictions discussed in the paper, where weight variance $\sigma_w$ significantly influences the dynamics of the latent states. For smaller values of $\sigma_w$, such as 1.0, the growth is more contained, allowing the network to maintain more stable latent states across inference steps. However, larger values lead to a divergence in $p^{l,t}$, which necessitates additional regularization techniques, as suggested in our proposed framework.

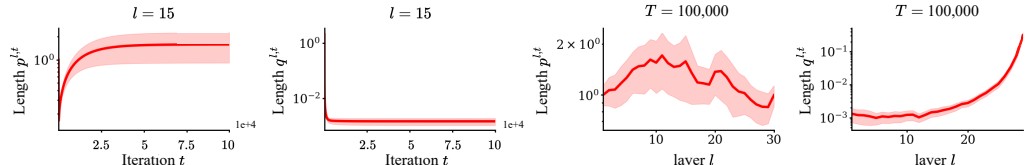

Figure 11: The dynamics of $p^{l,t}$ and $q^{l,t}$ for PCNs ($T = 100,000$). Settings not mentioned or indicated are identical to those in Figure 3.

Figure 11 explores the effect of extremely large inference steps ($T = 100,000$) on the dynamics of $p^{l,t}$ and $q^{l,t}$. Despite the large number of steps, the latent states and PEs stabilize after sufficient inference steps when $\sigma_w = 1.0$. However, we also observe that PEs tend to concentrate near the output layer, a phenomenon consistent with earlier findings that show concentrated PEs as a major challenge in deep PCNs. This stability over extended inference periods suggests that while PCNs can converge in theory, the issue of error concentration near the output layer persists. The results emphasize the need to balance PEs to prevent output-layer dominance, a feature crucial in deep networks for robust training.

Figure 12 provides a heatmap visualization showing the effects of $\sigma_w$ and $\sigma_b$ on the latent state lengths. For both linear and nonlinear PCNs, we observe that $\sigma_w$ has a much more significant

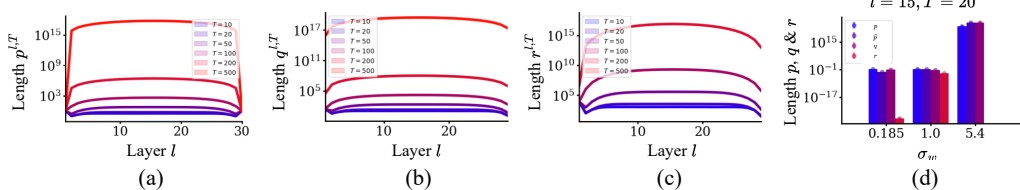

Figure 12: Heatmap plot of length $p^{l,t}$ for linear and nonlinear PCNs. ($\sigma_w \in \{0.6, 1.2, ..., 6.0\}$ and $\sigma_b \in \{0.6, 1.2, ..., 6.0\}$). The total number of inference steps $T = 10$ and the layer index $l = 15$. Settings not mentioned or indicated are identical to those in Figure 3.

impact on the length dynamics than $\sigma_b$. This supports the notion that the variance of the weights is the primary driver of instability, while the bias variance has a more subdued role. The heatmap also reveals that larger $\sigma_w$ values result in increasingly longer latent state lengths. These findings underline the necessity of controlling weight variance during initialization, as unchecked variance can lead to runaway growth in latent states.

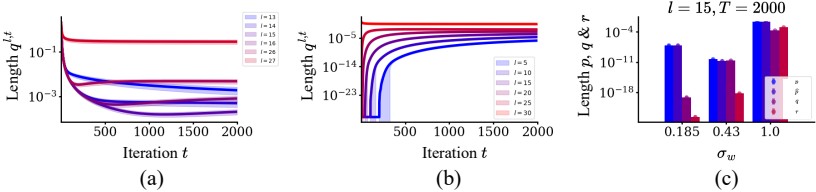

Figure 13: (a)-(c) The layer-wise distribution of $p^{l,t}$, $q^{l,t}$, and $r^{l,t}$ for linear PCNs ($\sigma_w = 5.4$ and $\sigma_b = 0.1$). The results are shown for the cases of $t \in \{10, 20, 50, 100, 200, 500\}$ with different colors. (d) Direct comparison of $p$, $\hat{p}$, $q$, and $r$ for $l = 15$ and $t = 20$. $\hat{p}$ represents the length of the prediction. Settings not mentioned or indicated are identical to those in Figure 3.

Figure 13 presents a detailed examination of the layer-wise distribution of $p^{l,t}$, $q^{l,t}$, and $r^{l,t}$ in linear PCNs for different inference steps $t$. These subfigures aim to capture how the latent state lengths, PEs, and weight update magnitudes evolve across different layers and with varying $t$. In Figure 13a-c, for $\sigma_w = 5.4$, we observe an exponential growth pattern in the values of $p$, $q$, and $r$ across all layers, particularly as $t$ increases. This growth is expected, given that larger weight variances typically result in larger latent state dynamics, leading to a cascading effect on PEs and weight updates. The increase in $p$, $q$, and $r$ with inference steps indicates that the internal representations become increasingly unstable as the inference phase progresses without proper regularization. Figure 13d highlights a direct comparison between $p$, $\hat{p}$ (the length of predictions), $q$, and $r$ for layer $l = 15$ at $t = 20$. Across all values of $\sigma_w$, we observe that $p$, $\hat{p}$, and $q$ remain within a similar range, though their values become more exaggerated for higher $\sigma_w$ values. Notably, $r$, which represents the weight update length, shows explosive growth when $\sigma_w = 5.4$, making it impractical to display fully. This behavior confirms that the higher values of $\sigma_w$ without regularization lead to unstable weight updates. Interestingly, for lower $\sigma_w$ values (e.g., $\sigma_w = 0.185$), $r$ remains small, indicating that proper initialization can contain these dynamics. However, $\sigma_w = 1$ shows a more moderate, controllable behavior in $r$. This figure emphasizes the need for length regularization and highlights the trade-off between network capacity (as influenced by $\sigma_w$) and the necessity of stability through regularization techniques.

Figure 14: (a) & (b) The dynamics of $q^{l,t}$ for linear PCNs ($\sigma_w = 1$ and $\sigma_b = 0.1$). The results are shown for the cases of different layer index $l$ with different colors. (c) Direct comparison of $p$, $\hat{p}$, $q$, and $r$ for $l = 15$ and $T = 2000$. $\hat{p}$ represents the length of the prediction. Settings not mentioned or indicated are identical to those in Figure 2.

Figure 14 illustrates the dynamics of $q^{l,t}$ (PE lengths) in linear PCNs, with $\sigma_w = 1$ and $\sigma_b = 0.1$, across different layer indices and inference steps. In Figure 14a and b, we see that the PE length ($q^{l,t}$) increases significantly as we approach the output layer (indicated by red lines). This trend is consistent with the concentration of PEs in deeper layers, a challenge observed in deep PCNs that affects the learning capacity of intermediate layers. Conversely, the PE length in earlier layers (indicated by blue lines) starts small and grows gradually with further inference steps, reinforcing the observation that early layers tend to stabilize more effectively than deeper layers. Figure 14c compares $p$, $\hat{p}$, $q$, and $r$ for layer $l = 15$ at $T = 2000$. The comparison shows how the dynamics of prediction lengths ($\hat{p}$), latent state lengths ($p$), and the magnitude of $r$ (weight update length) become highly dependent on $\sigma_w$. As noted earlier, the growth in $r$ with larger $\sigma_w$ values can lead to instability, stressing the need for controlled weight updates via regularization mechanisms.

