# OpenReview forum: "Stable and Scalable Deep Predictive Coding Networks with Meta-Prediction Errors"
_ICLR.cc/2026/Conference — ICLR 2026 Poster_

### Official Review · Reviewer_QiPr · 2025-10-21

**Soundness:** 3
**Presentation:** 3
**Contribution:** 3
**Rating:** 8
**Confidence:** 3

**Summary:**

The paper empirically illustrates 2 problems, imbalance and unstable PEs, of PCN with high numbers of layers. Based on that analysis, the paper propose a framework to mitigate the problems based on Meta-PC objective and weight regularization. The paper then demonstrates the effectiveness of the framework on VGGs and Resnet-18 in CIFAR10/100 and TinyImagenet.

**Strengths:**

I find the reasoning flow of the paper clear and strong. I am also impressed in how the proposed technicalities can improve PCN significantly, matching traditional back-propagation approach.

**Weaknesses:**

It will be an excellence paper if the authors can demonstrate the method in larger dataset like Imagenet or demonstrate the scalability in text data on more recent language models. Nevertheless, I think this is a strong, solid paper already.

**Questions:**

1) Resnet-18 seems to have lower gap between PCN and Meta-PCN. Why is that?
2) How do you implement PCN and Meta-PCN into VGGs and Resnet-18? Are there any modifications in the architectures or the only changes are in inference and learning procedure?
3) Ablation results on Meta-PCN objective and Weight regularization, i.e., how each modifications improve PCN?
4) Are there any hyper/tuning-parameters related to Meta-PCN? It seems this method is parameter-free to me, but I would like to have a clarification on this.
5) Are there any future research directions can be beneficial from this framework based on PCN?

---

> ### Author Response · Authors · 2025-11-26
>
> We sincerely thank the reviewer for the positive evaluation and for describing our submission as a strong, solid paper. We also appreciate the concrete suggestions for further extending and clarifying our work. Below, we address each point in turn.
>
> ### W1. Extension to Larger Datasets and Text Models
> We fully agree that extending Meta-PCN to larger-scale datasets such as ImageNet and to text-based architectures built on recent language models is an essential and natural next step for this line of research. Due to practical constraints on the rebuttal period, we are unable to include these additional large-scale experiments in the current revision. However, we have explicitly highlighted this as a key future research direction in the revised version to clearly state the intended scope of Meta-PCN beyond the current experiments (Appendix B.3).
>
> ---
>
> ### Q1. Depth vs. Performance (ResNet-18 Gap)
> Conceptually, Meta-PCN remains close to the original PCN formulation: it preserves local update rules and the overall predictive-coding structure. Because inference still proceeds through local error updates, sufficiently deep networks require longer inference horizons $T$ to propagate label-related prediction errors from the output boundary all the way back to the input boundary. In particular, when $T < L-1$, there is a structural limitation: the classification error at the output layer cannot fully reach the input layer within a single inference episode. In such settings, the effective depth is shorter from the perspective of the inference dynamics, making performance degradation inevitable for PCNs, including Meta-PCN, especially in deep architectures (Appendix B.2).
>
> Although this limitation exists for finite $T$, it also points to a promising direction in streaming or continual settings, where $T$ can be interpreted as real-time evolution. In that case, the joint progression of physical time and inference steps may alleviate this constraint and enable us to exploit the intrinsic advantages of PCNs better. We intend to explore such streaming-data formulations of Meta-PCN in future work (Appendix B.3).
>
> ---
>
> ### Q2. Implementation Details (VGGs and ResNet-18)
> The underlying architectures (VGGs and ResNet-18) and all shared hyperparameters are kept the same across backpropagation (BP), conventional PCN, and Meta-PCN. The only differences lie in the learning algorithm: the objective function, the resulting update rules, and the additional weight regularization specific to Meta-PCN. We clarify this implementation detail more explicitly in the revised version (Section 6 and Appendix E).
>
> ---
>
> ### Q3. Ablation Results on Meta-PCN Components
> The requested ablation results on the Meta-PCN objective and weight regularization are already provided in Appendix I, where we systematically evaluate the contribution of each component. To make this more visible, we have added a summary of these ablation findings in the main text (Section 6), so that readers can more easily see how each modification improves over conventional PCN.
>
> ---
>
> ### Q4. Hyperparameters Specific to Meta-PCN
> Meta-PCN does not introduce extra tunable hyperparameters that must be hand-selected for the new objective or the weight normalization. We explicitly clarified in the revised version that Meta-PCN is not associated with additional method-specific hyperparameters to avoid any confusion (Section 6).
>
> ---
>
> ### Q5. Future Research Directions Based on This Framework
> Since predictive coding has its roots in modeling the cortical mechanisms of the human brain, we view Meta-PCN as particularly well-suited to scenarios in which agents interact with the real world under streaming input. Designing Meta-PCN architectures that are intrinsically aligned with streaming data—similar to how humans continuously process sensory inputs—constitutes a fundamentally interesting direction.
>
> More concretely, we are interested in reinterpreting the frozen feedforward predictions as slowly updating variables and the latent states as fast-updating variables (Appendix A.2). This perspective may enable Meta-PCN variants that are naturally adapted to streaming settings, and it suggests a possible continuous-time formulation in the spirit of Neural ODEs. In the revised version, we condense these ideas into a concise statement in the conclusion as a future research direction (Section B.3).
>
> ---
>
> Once again, we appreciate the reviewer’s positive assessment and constructive suggestions, which helped us clarify the scope, implementation details, and future extensions of Meta-PCN.

---

### Official Review · Reviewer_VakM · 2025-10-27

**Soundness:** 3
**Presentation:** 2
**Contribution:** 3
**Rating:** 6
**Confidence:** 4

**Summary:**

### Summary

The paper introduces Meta-PCN, a framework designed to stabilize deep Predictive Coding Networks, which traditionally face severe scalability issues. The authors identify two causes for this instability (1) *prediction error (PE) imbalance* and (2) *exploding/vanishing prediction errors (EVPE)*.

The authors employ dynamical mean-field theory to characterize these instabilities and propose two core innovations:

1. **Meta-prediction error loss**, which minimizes prediction errors of prediction errors, effectively linearizing inference dynamics.
2. **Variance-based weight regularization**, which controls weight variance to prevent EVPE and ensure balanced signal propagation.

Empirical results across CIFAR-10/100 and TinyImageNet show substantial stability gains and accuracy improvements of Meta-PCN over PCNs while maintaining biological plausibility.

**Strengths:**

### Strengths

- **Convincing Theoretical results:** The paper provides a rigorous dynamical mean-field analysis of PCN inference dynamics, clearly identifying root causes of instability (Sections 3.2–3.3).
- **Experiments supporting theoretical claims:** Results show experimentally the root causes of instability. Figures 2–3 nicely illustrate the characteristic U-shaped PE imbalance and exponential EVPE dynamics; Figures 4–5 convincingly show that Meta-PCN mitigates these.
- **Novel conceptual contribution:** The idea of *meta prediction errors* offers an alternative to avoid gradient starvation.
- **Strong connection to neuroscience:** The work maintains biological plausibility, an property of high relevance for both ML and computational neuroscience.

**Weaknesses:**

- **Implementation comparison gaps:** The predictive coding baseline underperforms relative to prior works (e.g., Pinchetti 2025), raising questions about whether the reference implementation is optimal.
- **Connection to backpropagation not fully unpacked:** The equilibrium of the new loss (Section 4.1) appears mathematically equivalent to an iterative local implementation of backpropagation (δₗ = g(δₗ₊₁, hₗ₊₁⁰)), but the paper doesn’t explicitly acknowledge this equivalence or discuss implications for biological plausibility.
- **Missing mention of the assumptions of the length analysis**: eg. linear activation function and all layers share the same size. These assumptions should be clearly stated in the main text (section 3).
- **Limited experimental details**: The main text should contain necessary information to understand how the experiment were run broadly. Some additional details should be added, for instance the activity initialisation used for PCNs in figure 2.

**Questions:**

I really enjoyed reading your manuscript, especially section 3. However, there remain a few open questions that would need to be addressed before I could confidently raise my score to an 8. Could you please clarify the following:

**Major**

- **Ablation study:** Based on Figure 2, EVPE appears to be the dominant source of instability in PCNs. However, the ablation results presented by the authors do not include a direct comparison of a standard PCN augmented only with the proposed weight regularization. The experiments instead compare Meta-PCN variants with and without this regularization. Demonstrating that weight regularization alone is insufficient for achieving scalable learning in PCNs would be important to substantiate the necessity of the full Meta-PCN framework.
- **Precision learning:** It appears that optimizing the precision (1/variance) of PCNs layers would eliminate the PE imbalance by rescaling each error based on its variance (Bogacz 2017). Do you expect this to be the case? While the precisions are often assumed to be identity in PCNs for computational simplicity, they remain an essential part of PCNs that should be considered in your analysis.
- **backpropagation equivalence:** At equilibrium, the meta-PE formulation appears exactly equivalent to backpropagation ($\delta_l = g(\delta_{l+1}, h_{l+1}^0)$) but obtained using a local objective, could you clarify this relationship formally?
- **Implementation discrepancy:** Why does your predictive coding baseline perform so much worse than previous works (e.g., Pinchetti 2025)?

**Minor**

- **Energy propagation:** Why does error concentration persist at the output boundary (Fig. 4a)? Does this suggest incomplete energy flow across the network?
- **Baselines:** Did you apply *weight normalization* to the BP baseline for a fair comparison?
- **Biological interpretation:** Can Meta-PCN be implemented in a local neural circuit model consistent with cortical predictive coding mechanisms?

---

> ### Author Response · Authors · 2025-11-26
>
> We thank the reviewer for the careful reading of our manuscript and the detailed, technically engaged comments. We truly appreciate the opportunity to clarify our implementation choices, theoretical assumptions, and experimental design. Below, we address each point in turn.
>
> ---
>
> ### W1 & Q4. Implementation comparison gaps
> At the time of writing this response, we are still in the process of fully isolating the implementation differences that account for the performance gap with [1]. Here we report the current status of our investigation and the concrete steps we have taken during the discussion period.
>
> First, focusing specifically on [1], we note that their PC experiments use custom VGG-style architectures with 5, 7, and 9 layers. These differ from the canonical VGG configurations (e.g., the original VGG-11/13/16/19 [2]) and thus must be regarded as custom architectures rather than standard off-the-shelf models. While the appendix in [1] provides details for the 5- and 7-layer VGG variants, it does not fully specify the 9-layer model. Moreover, in the released code (https://github.com/liukidar/pcx), the classification experiments include an AlexNet-based setup, but an explicit implementation of the VGG variants is not provided. As a result, reproducing their exact PCN configuration requires nontrivial reconstruction effort, which we have been undertaking during the discussion phase.
>
> In addition to architectural differences (latent dimensionalities, placement of max-pooling, arrangement of convolutional vs. linear layers, etc.), we have identified three major training choices in [1] that are independent of the core PC mechanism yet have a substantial impact on performance:
> - Use of GELU as the non-linear activation function,
> - Use of *momentum* during the latent-state updates in the inference phase,
> - No weight decay on the synaptic weights.
> From the perspective of the broader deep-learning literature, we do not consider this particular combination of choices to be “standard practice” for training deep classifiers; instead, they are method-specific auxiliary training choices (tuning knobs) that are orthogonal to the predictive coding mechanism itself. For this reason, our main experiments deliberately *did not* use these additional tricks; we aimed to compare PCN, Meta-PCN, and BP under a standard, controlled training protocol.
>
> To quantify the effect of these choices, we performed a controlled study on a conventional PCN with a VGG-13 architecture on CIFAR-10, sweeping over the presence/absence of GELU, weight decay, and momentum during inference. The results are summarized below:
>
> | GELU | Weight Decay | Momentum | Best Test Acc |
> | :--: | :----------: | :------: | :-----------: |
> |   O  |       O      |     O    |   28.58%  |
> |   O  |       X      |     X    |   24.88%  |
> |   O  |       X      |     O    |   23.63%  |
> |   O  |       O      |     X    |   23.51%  |
> |   X  |       X      |     X    |   15.09%  |
> |   X  |       O      |     O    |   14.34%  |
> |   X  |       X      |     O    |   13.46%  |
> |   X  |       O      |     X    |   12.11%  |
>
> These experiments show that combining GELU, momentum for latent updates, and removing weight decay can indeed yield sizable gains in test accuracy for PCNs. Crucially, however, these improvements arise from auxiliary training heuristics rather than from changes to the underlying predictive coding mechanism.
>
> Our stance is therefore as follows:
> - We acknowledge that the [1] configuration achieves higher performance, and our ongoing work aims to reproduce their setup for a direct head-to-head comparison more closely.
> - At the same time, our primary goal in this paper is to analyze and improve the *structural stability* and *scalability* of PCNs via Meta-PCN, not to tune every baseline with all possible training tricks exhaustively. For this reason, our reference PCN implementation intentionally uses a more standard, broadly applicable training protocol.
> - The preliminary results above suggest that much of the reported gap can be attributed to such auxiliary tuning choices (GELU, inference momentum, no weight decay), which are orthogonal to the mechanisms that Meta-PCN is designed to address.
>
> We clarify this point in the revised version (Section 6) by (i) explicitly describing the differences in training heuristics relative to [1], and (ii) stating that our primary focus is on the stability/learning-dynamics aspect of PCNs under a controlled, standard training setup rather than on method-specific hyperparameter tuning.

---

> ### Author Response · Authors · 2025-11-26
>
> ### W2 & Q3. Connection to backpropagation
> As we already discussed in Section 2, following prior work on predictive coding, a *standard* PCN becomes mathematically equivalent to backpropagation only when the inference dynamics have fully converged to an equilibrium. This is a well-known property of vanilla PCNs [3-4] and not specific to Meta-PCN. Our paper explicitly cites the relevant prior work and notes that the "equilibrium of the PCN = iterative local implementation of backpropagation" under appropriate assumptions.
>
> Meta-PCN is constructed so that its equilibrium satisfies the *same* stationarity equation as the underlying PCN. Consequently, if the inference process converges to this equilibrium, Meta-PCN inherits the exact equivalence to backprop's local δ-relations as standard PCNs do. In this sense, Meta-PCN does not introduce a new type of backprop equivalence; rather, it preserves the existing one while modifying the *path* taken by the inference dynamics (via the meta-PE objective and variance-based regularization).
>
> However, in practical settings with a finite number of inference steps, the system may not reach equilibrium. In that regime:
> - The optimization process that drives PCN/Meta-PCN toward equilibrium is *different* from standard backprop.
> - A simple illustration is that backprop computes, in a single backward pass, the update that would satisfy the PCN equilibrium equation. In contrast, PCN/Meta-PCN must approach that solution iteratively through local state updates.
> - Therefore, in the practically relevant finite-step regime, Meta-PCN and backprop are generally *not* equivalent, just as vanilla PCN and backprop are not equivalent unless full convergence is achieved.
>
> To make this distinction fully transparent, we explicitly state it in Section 2 of the revised version.
>
> ---
>
> ### W3. Missing mention of the assumptions of the length analysis
> We agree that it is helpful for readers if the key assumptions are also summarized in the main text. These assumptions are indeed stated in Appendix C.1, where we present the full dynamical mean-field (DMFT) derivation. In the revised version, we have added the following explicit statement at the end of Section 3.2:
>
> > *"We emphasize that these results are derived under simplifying assumptions: i.i.d. Gaussian weights and biases, linear (or linearized) forward and backward maps, uniform layer widths, and the large-width limit. These assumptions do not perfectly match the deep, nonlinear convolutional architectures used in our experiments (Section 6), leaving an inherent gap between theory and practice."*
>
> Additionally, the caption for Figure 2 now explicitly references Appendix C.1 for the detailed assumptions. This clarification ensures the scope and limitations of the theoretical analysis are immediately visible, without requiring the reader to consult the supplementary material first.
>
> ---
>
> ### W4. Limited experimental details
> This information is already described in Appendix E, which introduces the DMFT-style synthetic experiments. To avoid any ambiguity in the main text, we have added an explicit sentence in the caption of Figure 2 describing the setup.
>
> To directly address the reviewer's question about **activity initialization**: In Figure 2, we use **Gaussian initialization** for the latent states, i.e., $z_{i,l}^0 \sim \mathcal{N}(0, 1)$ at $t=0$. The inputs and outputs are also sampled from a unit Gaussian distribution. This is consistent with our DMFT assumptions (Appendix C.1) and enables us to track the length dynamics analytically.
>
> The revised Figure 2 caption now reads:
> > *"Setting: A randomly initialized linear PCN with Gaussian-sampled inputs, outputs, and initial latent states ($\mathbf{z}_l^0 \sim \mathcal{N}(\mathbf{0}, I)$); no training is involved, purely for analyzing inference dynamics (see Appendix C.1 and E for details). We set $L = 30$, the terminal inference step is $T = 200$, and the latent dimension of each layer is set to 100. The inference rate is set to $\eta = 0.05$. $\sigma_w$ and $\sigma_b$ are set to 1 and 0.1, respectively."*

---

> > ### Comment · Reviewer_VakM · 2025-11-26
> >
> > W2 & Q3 — Connection to backpropagation
> >
> > I may have missed this in the manuscript, but could you provide (or point to) a proof that Meta-PCN’s equilibrium satisfies the same stationarity equation as the underlying PCN? Ideally, this would contain an analytical derivation and a small illustrative experiment showing that Meta-PCN converges to the same activity values as a standard PCN. For example the experiment could visualise the latent activity of a linear MLP with two hidden neurons and show that the activity of Meta-PCN converges to the same equilibrium activity.  That would make it much easier to understand and visualise how Meta-PCN relates to PCN, because the relationship implied by the equations in Section 4.1 is not obvious.
> >
> > Also please note that "a standard PCN becomes mathematically equivalent to backpropagation only when the inference dynamics have fully converged to an equilibrium." is not generally true during leanring. The equivalent between PCNs and backprop is only present at equilibrium in the limit where $||\delta_L||^2$ is downscaled to zero relative to  $||\delta_l||^2$ as described in Whittington & Bogacz 2017. This is learning with a PCN loss equal to $\mathcal{L} = \frac{c}{2}||\delta_L||^2 + \frac{1}{2}\sum_2^{L-1}||\delta_l||^2 $ for $c\rightarrow 0$. This setting is not the one discussed in the paper.
> > There exist other equivalents between PCN and backprop when the inference algorithm of PC is significantly modified, see Salvatori 2021. However, these are not at inference equilibrium. Please carefully formulate the equivalence between PCNs and backprop.
> >
> > Whittington and bogacz 2017, An Approximation of the Error Backpropagation Algorithm in a Predictive Coding Network with Local Hebbian Synaptic Plasticity
> >
> > Salvatori et al 2021, PREDICTIVE CODING CAN DO EXACT BACKPROPAGATION ONCONVOLUTIONAL AND RECURRENT NEURAL NETWORKS

---

> ### Author Response · Authors · 2025-11-26
>
> ### Q1. Ablation study: PCN + weight regularization
>
> In the original submission, our ablation study focused on progressively removing components of Meta-PCN (meta-PE objective, blocked sweeps, and variance-based normalization), which led us to underestimate the need for an explicit "PCN + weight normalization" condition. As the reviewer rightly points out, directly testing this variant is important for substantiating the necessity of the complete Meta-PCN framework.
>
> We have therefore run an additional experiment in which we apply our variance-based normalization to a conventional PCN, without introducing the meta-PE objective or blocked sweeps. On CIFAR-10 with a VGG-13 architecture, this "PCN + weight normalization" configuration achieves test accuracy of 10.16 ± 0.33\% across five runs, which remains clearly below Meta-PCN. These results indicate that while weight normalization alone can partially alleviate the symptoms, it does not resolve the underlying structural issues. The meta–prediction-error objective and the variance-based normalization act in a complementary manner: the former reshapes the objective to target the linearised equilibrium residual (mitigating PE imbalance and gradient starvation), while the latter directly controls the spectral factors that drive EVPE. We have added this new ablation to Table 4 and Appendix I.4.
>
> ---
>
> ### Q2. Precision learning
>
> We appreciate this question and the pointer to the role of precision in predictive coding. In the classical predictive-coding/free-energy framework, precision is defined as the inverse variance of prediction errors and acts as a reliability weight: under a Gaussian assumption with covariance $\Sigma_l$, the free energy takes the form $F = \\tfrac12 \\sum\_l \\boldsymbol{\\delta}\_l^\top \\Pi\_l \\boldsymbol{\\delta}\_l$ with $\\Pi\_l = \\Sigma\_l^{-1}$, so that errors with higher variance are downweighted and more reliable errors are upweighted. In our work, we deliberately fix these precision matrices to identity, i.e., $\\Pi\_l = I$, and consider the simplified free energy $F = \\tfrac12 \\sum\_l \\Vert \\boldsymbol{\\delta}\_l\\Vert^2\_2$. The focus of the paper is not on how to design or learn $\\Pi\_l$, but on why PE imbalance and EVPE arise in deep PCNs in the first place and how to stabilise these dynamics in a structurally principled way.
>
> In this sense, there is an important distinction between precision learning and our weight-variance control:
> * Precision operates on the **variance of the prediction errors** (via $\\Pi\_l \\approx \\Sigma\_l^{-1}$) and rescales $\\boldsymbol{\\delta}\_l$ directly.
> * Our variance-based normalization operates on the **variance of the weights** (the operators that propagate errors), controlling $\\sigma\_w^2$ and the spectral norm $\\Vert W\_l \\Vert\_2$ to prevent amplification/attenuation in the first place.
> Our DMFT length analysis tracks latent lengths $p\_l^t$, PE lengths $q\_l^t = \\text{len}(\\boldsymbol{\\delta}\_l^t)$, and weight-update lengths $r\_l^t$ and shows that, in the large-width limit, these quantities are determined mainly by $\\sigma\_w^2$ and the spectrum of the effective operators $W\_l^\\top D(\\mathbf{h}\_{l+1})$. In particular, the temporal evolution of PEs exhibits multiplicative scaling $\\Vert \\boldsymbol{\\delta}\_l^{t+1} \\Vert \approx \\tau\_t(\\sigma\_w)\\Vert \\boldsymbol{\\delta}\_l^t\\Vert$, where the factor $\\tau\_t(\\sigma\_w)$ is primarily controlled by $\\sigma\_w^2$ and the activation gain, leading to EVPE outside a narrow stable regime around $\\sigma\_w \\approx 1$. From this perspective, the observed PE variance structure is not a free degree of freedom to be arbitrarily reweighted by $\\Pi\_l$, but a dynamical consequence of the weight statistics; the most direct "control knob" for the pathology is therefore the weight variance/spectrum, rather than precision alone.
>
> We now address whether learning precision based on PE variance can fix the imbalance. Theoretically, if one could accurately estimate the layer-wise PE covariance and set $\\Pi\_l \\approx \\Sigma\_l^{-1}$, this would indeed help to flatten the distribution of PE energy across layers and time, and we expect it could mitigate some aspects of PE imbalance. However, because our DMFT analysis ties both PE imbalance and EVPE primarily to the spectral properties of the weights (through $\\sigma\_w^2$ and $\\tau\_t(\\sigma\_w)$), adjusting precisions on top of a mismatched weight spectrum is unlikely to remove the underlying dynamical instability entirely. Moreover, very aggressive precision down-weighting of large-variance errors risks suppressing proper learning signals and exacerbating gradient starvation.

---

> > ### Comment · Reviewer_VakM · 2025-11-26
> >
> > Q2. Precision learning
> >
> > The general reasoning behind the weight regularisation is clear. However, in PCNs the precision is defined as the inverse variance of the latent activity and not of prediction errors. This means that the diagonal elements of the precision matrix will be approximately equal to 1 over the norm of the prediction errors, effectively scaling errors such that PE imbalance would be compensated for in inference and parameter updates. However, this would likely not fully solve exploding and vanishing prediction errors, providing a motivation for regularisation beyond precision scaling. Please update your manuscript to accurately describe precision scaling in PCNs.

---

> > > ### Author Response · Authors · 2025-12-03
> > >
> > > ### W1 & Q4. Implementation comparison gaps
> > > As the review process has now transitioned from individual reviewer responses to consolidated AC evaluation, this response may be redundant; we therefore refer to our detailed response to Reviewer JWY7 (Q4) for the full discussion. The gaps we initially identified regarding PCN training—namely the use of GELU activation, momentum in latent state updates, and weight decay—were based on differences described in the paper. However, when comparing implementations directly, we found that the neural architectures themselves differ substantially; it was difficult to confirm whether momentum was actually used; and we identified differences in the $\\eta$ setting, which informed our subsequent experiments. For additional experiments, results, and our interpretation and conclusions, please refer to that response.
> > >
> > > ---
> > >
> > > ### W2 & Q3-1. On the relationship between Meta-PCN and standard PCN equilibria
> > > Our main point is that Meta-PCN and standard PCN do not in general share exactly the same equilibrium, and we should not claim that they do. Instead, Meta-PCN minimizes a *linearized* stationarity map around the feedforward initialization, and the two equilibria become close only under additional conditions.
> > >
> > > **1. Conditions under which the equilibria become close**
> > >
> > > The equilibria of Meta-PCN and standard PCN can *approximately* coincide only under additional assumptions. More precisely, if trajectories remain near the feedforward initialization: If the inference trajectories stay close to the feedforward initialization $\\mathbf{z}^{(0)}$, then at the fixed point $\\mathbf{z}\_l^\\star \\approx \\mathbf{c}\_l, \\text{ and }\\tilde{\\boldsymbol{\\delta}}\_l^\\star \\approx \\boldsymbol{\\delta}\_l^\\star,$ so that replacing $(\\mathbf{c}\_l, \\mathbf{h}\_{l+1}^{(0)})$ by $(\\hat{\\mathbf{z}}\_l, \\mathbf{h}\_{l+1})$ is locally valid, and the linearized map $\\tilde{F}$ approximates $F$: $F(\\mathbf{z}) \\approx \\tilde{F}(\\mathbf{z})$ in a neighborhood of $\\mathbf{z}^\\star.$
> > >
> > > Under this condition, the fixed point of Meta-PCN, defined by $\\tilde{F}(\\mathbf{z}) = 0$, will be close to the standard PCN fixed point satisfying $F(\\mathbf{z}) = 0$. In other words, Meta-PCN can be understood as solving a linearized equilibrium system whose solution approximates the original PCN equilibrium in the small-error, near-feedforward regime.
> > >
> > > We will revise the manuscript to:
> > > - remove the earlier over-strong statement suggesting that equilibria coincide in the linear case by default;
> > > - explicitly state that Meta-PCN minimizes the *linearized* PCN stationarity map around the feedforward initialization; and
> > > - clarify that approximate coincidence of equilibria requires additional conditions such as inference trajectories remaining close to the linearization point.
> > >
> > > **2. Planned Toy Example for Visualizing Meta-PCN Equilibria**
> > >
> > > We agree that a toy example would help clarify how Meta-PCN relates to a standard PCN. Due to time constraints during the discussion period and the complexity of analyzing equilibria in a form that involves inference, we could not implement this in the current version, but we plan to add it in the revised manuscript. Concretely, we will consider a small linear predictive-coding network (e.g., a three-layer linear MLP with two hidden units), run both PCN and Meta-PCN from the same feedforward initialization, and then perform an empirical landscape study in which we systematically vary weights and latent activities while measuring how well the delta relation is satisfied after inference. We will visualize the residual norm $\\Vert \\tilde{\\boldsymbol{\\delta}}\_l - g\_l(\\tilde{\\boldsymbol{\\delta}}\_{l+1}^{\*}, \\mathbf{h}^{(0)}\_{l+1})\\Vert$ as a contour plot over a low-dimensional weight/latent subspace and overlay the equilibria reached by PCN and Meta-PCN, providing a concrete visual comparison of where the two equilibria coincide or closely align.

---

> ### Author Response · Authors · 2025-11-26
>
> Our approach can therefore be viewed as a form of implicit precision control without explicit $\\Pi\_l$. Instead of learning $\\Pi\_l$, we (i) use variance-based normalization to keep $\\Vert W\_l \\Vert \_2 \\approx 1$ across layers, ensuring that the induced error gains $\\Vert W\_l^\\top D(\\mathbf{h}_{l+1}) \\Vert\_2$ remain near unity, and (ii) use the meta–prediction-error objective to align $\\boldsymbol{\\delta}\_l$ with $W\_l^\top D(\\mathbf{h}\_{l+1})\\boldsymbol{\\delta}\_{l+1}$, so that the effective "precision × weight" gain is approximately constant throughout the hierarchy. In the regime where the weight-induced error variance is normalised toward identity, the optimal precision matrix $\\Pi\_l = \\Sigma\_l^{-1}$ is itself close to $I$. The precision-weighted loss $\\boldsymbol{\\delta}\_l^\top \\Pi\_l \\boldsymbol{\\delta}\_l$ becomes effectively equivalent to the simple squared-norm loss we use.
>
> In summary, our choice in this work is to stabilise the dynamics at the level of the weights, so that a fixed identity precision becomes an appropriate and convenient gauge. We view precision learning and weight-variance control as complementary rather than competing strategies: precision mechanisms provide a principled way to reweight prediction errors, while our meta–prediction-error objective and variance-based normalization target the operator-level statistics that give rise to PE imbalance and EVPE in the first place (See Appendix B.4). Our results suggest that this structural, a priori control of the weight spectrum is already sufficient to mitigate these pathologies in deep PCNs substantially, and we regard integrating it with more general precision-learning schemes as an interesting direction for future work.
>
> ---
>
> ### Q5. (Minor) Energy propagation and error concentration at the output boundary
>
> In our supervised setting, the output layer is clamped to the label, i.e., $\mathbf z_L = \mathbf y$, so the output prediction error $\boldsymbol{\delta}_L = \mathbf z_L - \mathbf c_L = \mathbf y - \mathbf c_L$ is fixed throughout the inference process. Because both the prediction $\mathbf c_L$ and the target $\mathbf y$ are held constant, the supervised loss at the output layer does not change during inference, and the role of inference is to redistribute and propagate this fixed boundary error through the latent layers rather than to alter the boundary loss itself.
>
> ---
>
> ### Q6. Baselines: weight normalization for backprop
>
> Because our variance-based normalisation is a simple, lightweight mechanism, we also examined its effect on the backpropagation (BP) baseline to determine whether it should be considered a generic regulariser rather than a PCN-specific stabilisation tool. On CIFAR-10 with VGG-13, the standard BP baseline (without our normalization) achieves 87.85 ± 0.62% test accuracy over five runs. When we apply the same variance-based normalization to the BP weights, performance improves to 89.07 ± 0.46%. However, this remains below Meta-PCN (89.53 ± 0.47%), confirming that the meta-PC objective—not just weight normalization—is the primary driver of Meta-PCN's performance gains. We have added these results to Appendix J.
>
> ---
>
> ### Q7. Biological interpretation
>
> We thank the reviewer for raising this question about biological implementation. Contemporary cortical predictive coding (PC) models can be described at two complementary levels: (i) laminar hierarchies in which deep layers receive predictions from higher areas and superficial layers compute and relay prediction errors (PEs), and (ii) local microcircuits and single‑neuron mechanisms in which pyramidal cells compute PEs via balanced excitation–inhibition or dendritic comparison of inputs and predictions [5–7]. Because laminar circuits are built from such microcircuits [8–10], these views should be integrated rather than treated as alternatives.
>
> Meta‑PCN fits naturally into this integrated picture. In our framework, "meta‑prediction errors" correspond to local comparisons between a PE signal and a top‑down prediction of that PE, which can be implemented either by interacting populations of error neurons or by distinct dendritic compartments within a single pyramidal neuron, as suggested by recent microcircuit and dendritic PC models [5–7]. Within a laminar architecture, deep‑layer pyramidal cells send predictions (including predictions of expected PEs) to superficial circuits. Hence, each area effectively computes an "error of errors" using only local synaptic interactions. Recent thalamocortical models of visual inference with lamina‑specific prediction and error pathways further support this type of circuitry [11]. We will clarify this mapping and add these references in Appendix A on biological plausibility.

---

> ### Author Response · Authors · 2025-11-26
>
> Once again, we thank the reviewer for the thoughtful and constructive feedback. We believe that the clarifications and additional experiments prompted by these comments have helped us sharpen the scope of our claims and better communicate the contributions and limitations of Meta-PCN.
>
> ## References
>
> [1] Pinchetti, L., Qi, C., Lokshyn, O., Emde, C., M’Charrak, A., Tang, M., Frieder, S., Menzat, B., Oliviers, G., Bogacz, R., Lukasiewicz, T., & Salvatori, T. (2025). Benchmarking Predictive Coding Networks—Made Simple. The Thirteenth International Conference on Learning Representations. https://openreview.net/forum?id=sahQq2sH5x
>
> [2] Simonyan, K., & Zisserman, A. (2015). Very Deep Convolutional Networks for Large-Scale Image Recognition (No. arXiv:1409.1556). arXiv. https://doi.org/10.48550/arXiv.1409.1556
>
> [3] Whittington, J. C. R., & Bogacz, R. (2019). Theories of Error Back-Propagation in the Brain. Trends in Cognitive Sciences, 23(3), 235–250. https://doi.org/10.1016/j.tics.2018.12.005
>
> [4] Millidge, B., Song, Y., Salvatori, T., Lukasiewicz, T., & Bogacz, R. (2022). Backpropagation at the Infinitesimal Inference Limit of Energy-Based Models: Unifying Predictive Coding, Equilibrium Propagation, and Contrastive Hebbian Learning. The Eleventh International Conference on Learning Representations. https://openreview.net/forum?id=nIMifqu2EO
>
> [5] Hertäg, L., & Clopath, C. (2022). _Prediction-error neurons in circuits with multiple neuron types: Formation, refinement, and functional implications._ PNAS, 119(13), e2115699119.
>
> [6] Denève, S., & Machens, C. K. (2016). _Efficient codes and balanced networks._ Nature Neuroscience, 19(3), 375–382.
>
> [7] Mikulasch, F. A., et al. (2023). _Where is the error? Hierarchical predictive coding through dendritic error computation._ Trends in Neurosciences, 46(1), 45–59.
>
> [8] Maass, W., Natschläger, T., & Markram, H. (2004). _Computational models for generic cortical microcircuits._ In _Computational Neuroscience: A Comprehensive Approach_ (pp. 575–605).
>
> [9] Nelson, S. B. (2002). _Cortical microcircuits: diverse or canonical?_ Neuron, 36(1), 19–27.
>
> [10] Douglas, R. J., & Martin, K. A. C. (2004). _Neuronal circuits of the neocortex._ Annual Review of Neuroscience, 27, 419–451.
>
> [11] George, D., et al. (2025). _A detailed theory of thalamic and cortical microcircuits for predictive visual inference._ Science Advances, 11(6), eadr6698.

---

> ### Author Response · Authors · 2025-12-03
>
> ### W2 & Q3-2. Clarifying the equivalence between PCNs and backprop
> We thank the reviewer for pointing out this subtle but important issue. We agree that our original sentence — stating that a "standard PCN becomes mathematically equivalent to backpropagation once inference has converged to an equilibrium" — was too strong as written and did not fully reflect the conditions described in the existing literature.
> - In Whittington & Bogacz (2017), the equivalence between predictive coding and backpropagation does not hold for a generic predictive-coding loss. Instead, it is obtained under a *specific limit*: the training loss is chosen so that the error term at the output layer is weighted by a small scalar, and this scalar is taken to zero relative to the weights on the hidden-layer errors. Equivalently, in their original notation, this corresponds to choosing the output-layer noise (or variance) much larger than that of the other layers. Only in this special regime, and at inference equilibrium, do the predictive-coding weight updates converge to those produced by backpropagation.
> - Later work such as Salvatori (2021) derives further forms of equivalence, but these rely on significantly modified inference rules for predictive coding rather than on the standard PCN dynamics we study. These settings are therefore conceptually distinct from our current formulation.
>
> In the revised manuscript, we carefully reformulated our statements as follows:
>   - We will explain that predictive coding can approximate or match backpropagation only under additional assumptions on the relative weighting (or noise/variance) of the output-layer error term, as in Whittington & Bogacz (2017).
>   - We will adjust the relevant sentences in Section 2 to make the conditional nature of the equivalence clear and to avoid any misleading impression that PCNs are generally equivalent to backpropagation at equilibrium.
>
> ---
>
> ### Q2. Precision learning and its relation to our weight regularization
> We thank the reviewer for this helpful clarification. We agree that our previous wording was imprecise and we corrected it in the revised manuscript.
>
> In the standard predictive-coding formulation, precision is defined as the inverse covariance of the latent activity (state), not of the prediction errors themselves. Under this definition, the diagonal entries of the precision matrix scale prediction errors inversely to their latent variance, which can partially compensate for PE imbalance across layers during inference and learning. We will update the relevant passages to reflect this standard definition and mechanism.
>
> By contrast, our theoretical analysis in this paper is carried out in a simplified setting where the precision matrices are fixed to the identity and are not learned. In this regime, our DMFT-based length analysis shows that PE imbalance and exploding/vanishing prediction errors are primarily driven by the weight variance $\\sigma\_w^2$ and the spectrum of the effective operators $W\_l^\\top D(\\mathbf{h}\_{l+1})$. Thus, while precision learning can help reweight errors, it is unlikely by itself to remove these dynamics-level instabilities when the underlying weight spectrum is poorly conditioned.
>
> We therefore present weight-variance normalization (WVN) as a structural, complementary mechanism: it acts directly on the spectrum of $W\_l$ and, together with the Meta–PE objective, stabilizes inference and learning even in the fixed-precision setting considered here. In the revision, we will (i) correct the definition of precision, (ii) briefly explain how precision scaling can mitigate PE imbalance, and (iii) clarify that our analysis and claims are restricted to the fixed-precision case and are intended to complement, rather than replace, precision-based approaches.

---

### Official Review · Reviewer_JWY7 · 2025-10-29

**Soundness:** 3
**Presentation:** 2
**Contribution:** 2
**Rating:** 2
**Confidence:** 5

**Summary:**

The paper proposes a theoretical analysis of predictive coding models, as well as a solution to the problem of training very deep networks. This solution is based on two novel proposals: the use of meta-prediction errors, and a new kind of normalization of the weight matrices. A large empirical evaluation shows the effectiveness of the proposed method on a large number of architectures, where the comparison is made agains variations of PC, and backprop.

**Strengths:**

The paper is well motivated, tackling a timely problem in the field of predictive coding. The empirical evaluation is well done, as it tests on a large number of models and datasets, showing clear improvements in performance. Also, the use of dynamical mean field theory to study  the instability/vanishing error problems of PCNs is novel and interesting.

**Weaknesses:**

There are a couple of problems in the solutions proposed by the authors, that affect the novelty and bio-plausibility of the meta prediction errors, that is, the fact that they rely on the value of the neurons during the forward pass (so, at t=0). The problems are the following:

1) This is not novel, but it has already been done in three other works, that you also cite: Whittington&Bogacz,2017, B.Millidge 2020 (Predictive coding approximates backprop along arbitrary computation graphs), and C.Qi 2025 (Towards the Training of Deeper Predictive Coding Neural Networks). How is the energy function you describe different from theirs?

2) This is also biologically implausible, as already stated in the three works above, as it breaks the locality in time that define what biologically plausible algorithms are  (operations must rely on information that is locally available at that time step).

A second problem in terms of novelty, is the claim that you have identified the problems of vanishing PE, and PE imbalance across the network. While I agree that the theoretical analysis you propose is novel, I believe you should state that both problems have been already identified in previous work: [1] shows this problem in a small neural network, while [2,3,4] propose their theoretical analysis. So i would rephrase your list of contributions by acknowledging existing work first, the current state of the art, and from there list your contributions. On a side note, I'm not a big fan of works that place the related works in the supplementary material, as it leads exactly to this kind of confusion. Maybe you could move a summary that discusses only these points in the main body?

Minor:

you propose two methods to improve the results, but you only test them simultaneously. I would add an ablation study so that the reader can understand which method has the largest impact in the final test accuracy.

In figure 2, what is the exact setup? Is it a randomly initialized model? Is it already trained? On which data?

At some point you cite:

*Beren Millidge, Alexander Tschantz, and Christopher L Buckley. Predictive coding networks for
temporal prediction. Neural Networks*

This is a wrong citation, maybe you meant this one, with the same title?
https://journals.plos.org/ploscompbiol/article?id=10.1371/journal.pcbi.1011183



[1] L .Pinchetti et al. Benchmarking predictive coding networks -- Made Simple (2024)

[2] C.Qi et al. Towards the Training of Deeper Predictive Coding Neural Networks (2025)

[3] Cédric Goemaere et al. Overcoming Exponential Signal Decay in Deep Predictive Coding Networks (2025)

[4] Francesco Innocenti et al. uPC: Scaling Predictive Coding to 100+ Layer Networks (2025)

**Questions:**

See above. My main concern at the moment is the novelty of the work, so I'd be happy to reconsider my score if this concern is properly addressed. (Also, the score I gave is a little too harsh in my opinion, as the work does have value. However, ICLR this year does not allow for intermediate scores, and 4 is already considered borderline...)

---

> ### Author Response · Authors · 2025-11-26
>
> We thank the reviewer for the time and effort devoted to evaluating our manuscript and for the detailed comments. The points raised have been valuable for clarifying the scope, assumptions, and presentation of our work. Below, we address each concern in turn, explaining how it relates to the current formulation and indicating the clarifications made and where they appear in the revised version of the manuscript.
>
> ---
> ### W1. Novelty of the energy function
> We would like to clarify the methodological contributions we actually claim in the paper. As stated in the abstract, introduction, methodology, and conclusion, our work is built around two main components:
> 1. A meta–prediction-error objective, i.e., a new free-energy formulation based on prediction errors *of* prediction errors; and
> 2. A weight variance normalization scheme motivated by a DMFT-based analysis of error dynamics.
> Feedforward-based latent initialization itself is a common practice in the PCN literature, and we do not present it as a novel contribution. Likewise, using feedforward predictions as part of the inference setup is only one element of the overall framework.
>
> Our main contribution in Section 4.1 lies instead in introducing a new free-energy function based on *meta prediction errors*, i.e., prediction errors of prediction errors. Concretely, building on the linearization of the nonlinear equilibrium system $F(\\mathbf{z}) = \\nabla\_{\\mathbf{z}}\\mathcal{F}(\\mathbf{z}) = 0$, we define the novel free energy $\\mathcal{J}(\\tilde{\\boldsymbol{\\delta}}) = \\frac{1}{2} \\sum\_{l=2}^{L-1} \\big\\Vert \\tilde{\\boldsymbol{\\delta}}\_l - g\_l\\big(\\tilde{\\boldsymbol{\\delta}}\_{l+1}^{\*}, \\mathbf{h}^{(0)}\_{l+1}\\big) \\big\\Vert\_2^2,$ where $\\tilde{\\boldsymbol{\\delta}}\_l = \\mathbf{z}\_l - \\mathbf{c}\_l$ denotes the (feedforward-based) prediction error and $\\tilde{\\boldsymbol{\\delta}}\_l - g\_l(\\tilde{\\boldsymbol{\\delta}}\_{l+1}^{\*}, \\mathbf{h}^{(0)}\_{l+1})$ is the prediction error *of* this prediction error (meta prediction error). In contrast to conventional formulations that minimize $\\tfrac{1}{2}\\sum\_l \\Vert \\boldsymbol{\\delta}\_l \\Vert^2$, our loss explicitly targets this meta-level residual, which coincides with the linearized stationarity map of the original equilibrium system. This meta–prediction-error free energy, and its derivation from the linearized equilibrium equations, is the key novelty of our energy function beyond standard feedforward initialization.
>
> This meta–prediction-error free energy, together with the DMFT-guided variance normalization, is what we propose as the core methodological advance—rather than feedforward initialization itself. Other reviewers have identified this meta-PE formulation and the associated stability analysis as the novel aspects of the work. We also make this distinction more explicit in the revised version and clarify it in the related work discussion (Sections 4.1 and 7).

---

> ### Author Response · Authors · 2025-11-26
>
> ### W2. Biological plausibility
> Before addressing this point, we would like to reiterate that our methodological contribution is not the use of feedforward predictions for latent initialization. The central technical contribution of our work lies in (i) the meta–prediction-error objective (a new free-energy formulation based on prediction errors of prediction errors) together with (ii) weight variance normalization, and in their DMFT-motivated design (Sections 3–4). The "prediction freeze" mechanism is part of this framework's implementation, but it is not the main novelty we consider.
>
> Although the reviewer did not explicitly raise this particular aspect, we think it is helpful to clarify in advance how the prediction-freeze mechanism can be interpreted from a biological perspective. Since our formulation includes such a mechanism, it is natural to ask how it interacts with temporal locality and biological plausibility. Below, we outline one plausible interpretation that we regard as compatible with established cortical principles; we also refer the reviewer to the discussion in Appendix A for further details.
>
> Our view can be related to two strands of neurobiological evidence:
>
> (1) **Intrinsic timescales and separation between prediction and latent dynamics**
> Empirical and theoretical work [1-2] suggests that cortical processing involves a hierarchy of intrinsic timescales: some neural signals evolve rapidly, while others vary more slowly and provide a relatively stable context. In the context of Meta-PCN, one can interpret our construction as introducing a separation between:
> - Prediction signals $\mathbf{c}_l$ (the feedforward predictions), which can be viewed as relatively stable reference signals during an inference episode, and
> - Latent states $\mathbf{z}_l$, which are iteratively updated to reduce prediction errors.
> From this perspective:
> - **Timescale separation.** During the short inference window over which the latent states are updated, the prediction signals $\mathbf{c}_l$ change only slowly and can be treated as quasi-static.
> - **Frozen predictions.** Interpreting the feedforward predictions as fixed constants over this window can thus be viewed as more than a pure implementation convenience: it provides one compact mathematical abstraction in which relatively stable prediction signals supply boundary conditions for the faster error-driven updates of the latent states. We intend this as a coarse-grained abstraction consistent with known timescale differences in cortical processing, rather than a literal circuit-level claim.
> In other words, the frozen predictions in Meta-PCN can be understood as reflecting a separation between comparatively stable prediction signals and more rapidly adapting latent states, rather than as an ad hoc violation of temporal locality.
>
> (2) **Sustained activity and distributed working memory (in a coarse-grained sense)**
> A second, complementary way to view the exact mechanism is through sustained neural activity and distributed working memory:
> * Many models of cortical circuits use active maintenance (via recurrent connectivity and slow synaptic mechanisms) and attractor-like dynamics to explain how information can persist beyond the immediate sensory input [3-4]. Under this coarse-grained view, treating prediction as fixed during inference can be interpreted as assuming that prediction-related activity patterns are sufficiently stable over the timescale of the error-driven updates of the latent states.
> * Recent work [5] further suggests that working memory is likely distributed across multiple areas, rather than localized to a single buffer. Freezing prediction signals across layers $l = 1,\dots,L$ in Meta-PCN can then be seen as a simplified way to capture such distributed, sustained activity, providing a stable scaffold for iterative inference in deep networks.
>
> We consider these mechanisms to offer a plausible, coarse-grained interpretation in which all computations take place within ongoing interactions between prediction signals and latent states with different intrinsic timescales, supported by active maintenance. This interpretation avoids explicit storage-and-replay of past activations and, to the best of our understanding, is compatible with existing neurophysiological evidence on timescale hierarchies, sustained activity, and distributed working memory.
>
> We make this interpretation more explicit in the revised version and strengthen the pointer to the discussion of biological plausibility in Appendix A.

---

> ### Author Response · Authors · 2025-11-26
>
> ### W3. Novelty of identified problems (PE imbalance and vanishing PE)
> We agree that prediction-error (PE) imbalance and vanishing (or exploding) PEs in deep PCNs have been reported in several recent works, and we appreciate the reviewer drawing attention to this historical context. Our paper has undergone a lengthy revision process, during which multiple related analyses emerged, so the novelty of “problem identification” alone is indeed less clear-cut than when we began.
>
> Nevertheless, we believe that the present work still makes a distinct and substantive contribution for the following reasons. Rather than merely observing PE imbalance and EVPE empirically, we use a DMFT-based length analysis to reproduce these phenomena in a controlled setting and trace their structural origin to the interactions among hierarchical layers, boundary conditions, and the spectral properties of the weight operators. Crucially, this theoretical analysis directly motivates a unified remedy—combining the meta–prediction-error objective with variance-based weight normalization—so that both the problem identification and the proposed solution are tied together within a single, coherent framework. Below, we clarify how this perspective differs from and complements prior studies.
>
> (1) **Acknowledging prior work**
> Several recent papers have independently documented depth-related pathologies in PCNs and proposed various remedies—e.g., Pinchetti et al. on energy/PE imbalance across layers [6], Goemaere et al. on exponential signal decay and error-based reparameterization (EO) [7], Qi et al. on exponentially imbalanced errors and improved weight/energy updates for deeper networks [8], and Innocenti et al. ($\mu$PC) on depth–$\mu$P scaling and stable training of 100+ layer PCNs [9].
> Some of these works were already discussed in our related-work section, but not with the benefit of the most recent versions and closely related analyses. In the revised version, we update and expand the related-work discussion to explicitly acknowledge that PE imbalance, vanishing/decaying signals, and depth-related instabilities have been observed and partially addressed in these studies, and then clearly position our DMFT-based analysis and unified meta-PE + variance-normalization framework in relation to them.
>
> (2) **What is new in our analysis**
> Our goal of this study is not merely to reconfirm that PE imbalance and EVPE exist, but to connect them to a precise structural mechanism and to derive a unified, theoretically motivated remedy. Concretely, our contribution proceeds in four steps:
> 1. **DMFT-based length analysis**: We develop a dynamical mean-field (length-based) framework for deep PCNs under Gaussian initialization, in which we can analytically track the evolution of latent-state lengths, PE lengths, and weight-update lengths across both layer and inference iteration. This analysis does not presuppose feedforward initialization and already reveals the characteristic U-shaped PE profile and exponential scaling patterns in a simplified, yet mathematically tractable, setting.
> 2. **Reproduction of pathologies within the theory**: Within this DMFT framework, we *reproduce* both PE imbalance (Problem 1) and EVPE (Problem 2): PE lengths decay geometrically away from the boundaries due to the product of spectral factors, and EVPE appears as geometric growth/decay over inference iterations governed by multiplicative factors $\tau_t(\sigma_w)$. This shows that the phenomena are not artifacts of specific heuristics, but emerge structurally from the underlying dynamics of deep PCNs.
> 3. **Identification of structural causes**: The theory isolates the root causes of these behaviours in terms of (i) boundary conditions and hierarchical layer interactions (for PE imbalance) and (ii) the spectrum and variance of the weight operators, coupled with activation gains, which induce a phase transition between exploding and vanishing PE regimes (for EVPE). In contrast to analyses that focus mainly on discretization, numerical precision, or empirical heuristics, our account explicitly ties EVPE and imbalance to the statistical and spectral properties of the weights in the large-width limit.
> 4. **Unified, theory-guided remedy**: Finally, the same analysis motivates our two synergistic components:
>     - **the meta–prediction-error objective**, which replaces the standard free energy with a loss equal to the linearised equilibrium residual and thereby alleviates gradient starvation and PE imbalance; and
>     - **the variance-based normalization rule**, which controls the effective spectral norms and directly targets the $\tau_t(\sigma_w)$ factors responsible for EVPE.
>     Together, these components form a single, DMFT-guided framework that addresses both pathologies in a principled manner, rather than as separate engineering fixes.

---

> ### Author Response · Authors · 2025-11-26
>
> In other words, our contribution goes beyond re-identifying the phenomena: we provide a large-scale DMFT explanation of why PE imbalance and EVPE arise from the structural and spectral properties of deep PCNs, and we use this explanation to design the meta-PE loss and variance-normalization scheme as a unified, theoretically grounded solution. We have also updated our contributions in Appendix B.1.
>
> ---
>
> ### W4. Position of Related Work Section
> Due to the strict page limit, our original submission placed the detailed related-work discussion in the Appendix. In the revised version, we have moved a concise but self-contained related-work subsection back into the main manuscript so that the positioning of our contributions relative to prior work is immediately precise.
>
> ---
>
> ### W5. Ablation study
> We would like to emphasize that the requested ablation is already included in the original submission (Appendix) and is also referenced in the main text. That ablation isolates the effect of each component and shows which is crucial for performance and stability.
>
> ---
>
> ### W6. Figure 2 setup
> The setup underlying Figure 2 is already described in Appendix C.1 & E of the original submission: it is a randomly initialized linear PCN, used solely to analyze inference-time length dynamics. To avoid any possible ambiguity, we have added the following explicit sentence in the figure caption:
>
> > *"A randomly initialized linear PCN with Gaussian-sampled inputs, outputs, and initial latent states ($\mathbf{z}_l^0 \sim \mathcal{N}(\mathbf{0}, I)$); no training is involved, purely for analyzing inference dynamics (see Appendix C.1 and E for details)."*
>
> To"directly answer the concerns:
> - **Randomly initialized?** Yes, weights are drawn i.i.d. from $\mathcal{N}(0, \sigma_w^2/N)$.
> - **Already trained?** No, no training is involved; this is purely for analyzing inference dynamics.
> - **On which data?** Inputs and outputs are sampled from a unit Gaussian distribution $\mathcal{N}(\mathbf{0}, I)$.
>
> ---
>
> ### W7. Misc: wrong citation
> We appreciate the pointer. We corrected the citation in our revised version.
>
> ---
>
> Once again, we thank Reviewer 2 for the comments. We believe that the clarifications above—especially the explicit statement of the meta–prediction-error free energy in Sec. 4.1, the DMFT-based positioning of our analysis relative to prior work, and the more explicit pointers to existing ablations and setups—accurately reflect the scope and contributions of our work while addressing all points raised in the review.
>
>
> ## References
>
> [1] Murray, J. D., Bernacchia, A., Freedman, D. J., Romo, R., Wallis, J. D., Cai, X., Padoa-Schioppa, C., Pasternak, T., Seo, H., Lee, D., & Wang, X.-J. (2014). A hierarchy of intrinsic timescales across primate cortex. Nature Neuroscience, 17(12), 1661–1663. https://doi.org/10.1038/nn.3862
>
> [2] Kiebel, S. J., Daunizeau, J., & Friston, K. J. (2008). A Hierarchy of Time-Scales and the Brain. PLOS Computational Biology, 4(11), e1000209. https://doi.org/10.1371/journal.pcbi.1000209
>
> [3] Wang, X.-J. (2001). Synaptic reverberation underlying mnemonic persistent activity. Trends in Neurosciences, 24(8), 455–463. https://doi.org/10.1016/S0166-2236(00)01868-3
>
> [4] Wang, X.-J. (2021). 50 years of mnemonic persistent activity: Quo vadis? Trends in Neurosciences, 44(11), 888–902. https://doi.org/10.1016/j.tins.2021.09.001
>
> [5] Mejías, J. F., & Wang, X.-J. (2022). Mechanisms of distributed working memory in a large-scale network of macaque neocortex. eLife, 11, e72136. https://doi.org/10.7554/eLife.72136
>
> [6] Pinchetti, L., Qi, C., Lokshyn, O., Emde, C., M’CharM'CharrakTang, M., Frieder, S., Menzat, B., Oliviers, G., Bogacz, R., Lukasiewicz, T., & Salvatori, T. (2025). Benchmarking Predictive Coding Networks—Made Simple. The Thirteenth International Conference on Learning Representations. https://openreview.net/forum?id=sahQq2sH5x
>
> [7] Goemaere, C., Oliviers, G., Bogacz, R., & Demeester, T. (2025). Error Optimization: Overcoming Exponential Signal Decay in Deep Predictive Coding Networks (No. arXiv:2505.20137). arXiv. https://doi.org/10.48550/arXiv.2505.20137
>
> [8] Qi, C., Lukasiewicz, T., & Salvatori, T. (2025, March 5). Training Deep Predictive Coding Networks. New Frontiers in Associative Memories. https://openreview.net/forum?id=s3E08R4AMK
>
> [9] Innocenti, F., Achour, E. M., & Buckley, C. L. (2025). $μ$PC: Scaling Predictive Coding to 100+ Layer Networks (No. arXiv:2505.13124). arXiv. https://doi.org/10.48550/arXiv.2505.13124

---

> ### Comment · Reviewer_JWY7 · 2025-11-26
>
> I will be brief and address the more detailed points of your rebuttal in a later moment. However, I'm not really satisfied with the answer and the updated manuscript:
>
> 1) The manuscript still claims that:
>
> > Our theoretical investigation (detailed in Section 3) reveals two distinct yet interconnected pathologies that impede deep PCN scalability:(1) PE Imbalance: Errors concentrate in boundary layers (input/output) while vanishing in intermediate layers. This creates a characteristic imbalanced distribution. This results in gradient starvation in mid-layers and prevents effective learning.(2) EVPE: Exponential growth and decay patterns emerge in latent states and PEs during inference. These dynamics are controlled by temporal scaling factors that depend critically on the variance of the weights.
>
> This is true, but a known fact. Hence, this is not a novel contribution of your work. The fact that you are aware of this, but only acknowledge it in the related works at the end of the paper is not enough: It should be clearly stated in the introduction. Otherwise, it may seem a novel contribution to a more inexperience reader.
>
> In a later point in your answer, you state that:
>
> > In other words, our contribution goes beyond re-identifying the phenomena
>
> This is true, and I have acknowledged it in my initial review. However, the way the manuscript is phrased now is heavily misleading: you can find everywhere claims where you state this as it being your contribution. This is also influencing this review process: all the other reviewers agree that the fact that you have uncovered such phenomena to be a strong point of your work, but this is simply a reflection of the way this manuscript is phrased.
>
>
>
> -----------------------------------------------------------------------------------------
>
> On the Meta-prediction errors, you did not directly answer my question: how is this different from the energy function used in [1] and [2]? If they are the same, where is the novelty?
>
> In general, the recent, historical, narrative, is that [1] and [2] proposed their loss (which seems to me the meta prediction error), just as a theoretical tool to show some similarities between backprop and PC, despite being aware of the biological implausibility. This has been heavily discussed here [3], where the authors state that this is not a good direction of research as it does not yeld any advantages (in my opinion, they are overly critical as I believe they have overlooked the fact that [1,2] just proposed it as a theoretical tool and nothing else, but still, the points they make are valid).
>
> [1] Whittington, James CR, and Rafal Bogacz. "An approximation of the error backpropagation algorithm in a predictive coding network with local hebbian synaptic plasticity." Neural computation 29.5 (2017): 1229-1262.
>
> [2] Millidge, Beren, Alexander Tschantz, and Christopher L. Buckley. "Predictive coding approximates backprop along arbitrary computation graphs." Neural Computation 34.6 (2022): 1329-1368.
>
> [3] Zahid, Umais, Qinghai Guo, and Zafeirios Fountas. "Predictive coding as a neuromorphic alternative to backpropagation: a critical evaluation." Neural Computation 35.12 (2023): 1881-1909.
>
>
>
> -------------------------------------------------------------------------
>
> Biological plausibility.
>
> The algorithm you propose is not biologically plausible, as it is not local in time. The answer given is not satisfactory as bio-plausibility cannot simply be motivated by the finding of structures in the brain that explain the architectural choices: giving the extremely complex functioning of brain processes, finding 'ad-hoc' explanations can be very easy. The way to look at bio-plausibility is that of a property that yelds some advantages. For example, locality in space and time allows the implementation of this class of algorithms to a kind of neuromorphic/analog hardware. The fact that the algorithm you propose is not local in time is a huge problem for this, and undermines the direction of research.
>
>
> -------------------------------------------------------------------------------
>
> Experimental results: As pointed out by another reviewer, how come you report the performance of a VGG5/7 classifier on CIFAR10 for standard PC to be almost random guessing? This is in stark contrast with the experiments reported in other papers, such as Pinchetti et al., where the authors report numbers that are much closer to that of backprop. I can agree that it is hard to exactly replicate the experiments of other works, but going from the 86% reported in their paper, to not even 30% reported in yours is a stark difference, that clearly suggests that your implementation of the baselines is completely suboptimal.

---

> > ### Comment · Reviewer_JWY7 · 2025-11-26
> >
> > In general, my 3 concerns for this work are not addressed:
> >
> > 1) Novelty: The multiple claims where you state that the energy imbalance is a novel contribution of your work must be removed, as they are false (and you have acknowledged it). I also believe the meta prediction error formulation to be not novel, as it is the same of the previously mentioned works.
> >
> > 2) Bioplausibility: the fact that the algorithm is not local in time is, for me, problematic.
> >
> > 3) The experimental evaluation is not fair, as the numbers reported for the baselines are much worse the ones reported in the original works.

---

> > > ### Author Response · Authors · 2025-12-03
> > >
> > > ### Q-c. Biological plausibility and temporal locality
> > > We thank the reviewer for the detailed follow-up on temporal locality and biological plausibility. We agree that, as written, Meta-PCN appears to depend explicitly on feedforward predictions at $t=0$, which can give the impression of a temporally non-local algorithm. In this response, we clarify how the same dynamics can be expressed in a time-local form, and how this relates to known mechanisms such as timescale separation and persistent activity.
> > >
> > > **1. Time-local reformulation via slow contextual states**
> > >
> > > Formally, the discrete-time Meta-PCN update can be interpreted as an explicit Euler discretization of a continuous-time system, with step size $\\Delta t$ and time constant $\\tau\_z$ related by $\\eta = \\Delta t / \\tau\_z$.
> > > The inference dynamics can be written as a fully time-local Markov system by making explicit the slowly varying "context" states that are implicit in this shorthand. Specifically, we can introduce slow states $\\mathbf{c}\_l^{(t)}$ and $\\mathbf{h}\_l^{(t)}$ with $\\mathbf{c}\_l^{(0)} = \\phi(\\mathbf{h}\_l^{(0)}), \\qquad \\mathbf{c}\_l^{(t+1)} = \\mathbf{c}\_l^{(t)}, \\qquad \\mathbf{h}\_l^{(t+1)} = \\mathbf{h}\_l^{(t)},$ or more generally with a very slow leak satisfying $\\tau\_c \\gg \\tau\_z$.
> > >
> > > In this form, each update at time $t$ depends only on the current state $\\big(\\mathbf{z}^{(t)}, \\tilde{\\boldsymbol{\\delta}}^{(t)}, \\mathbf{c}^{(t)}, \\mathbf{h}^{(t)}\\big)$; there is no need to access a stored snapshot from a separate time. The notation $\\mathbf{h}\_l^{(0)}$ and $\\mathbf{c}\_l$ used in the paper can therefore be understood as a compressed representation of these slow contextual states, rather than as an intrinsically non-local reference to a past time point.
> > >
> > > For completeness, the same idea can be expressed in continuous time as
> > > $\\tau\_z \\dot{\\mathbf{z}}\_l = -\\Big(\\tilde{\\boldsymbol{\\delta}}\_l - g\_l\\big(\\tilde{\\boldsymbol{\\delta}}\_{l+1}, \\mathbf{h}\_{l+1}\\big)\\Big), \\qquad \\tau\_c \\dot{\\mathbf{c}}\_l \\approx 0,$
> > > where $\\tau\_z$ and $\\tau\_c$ are distinct time constants. This shows that the dynamics can be implemented by two local integrators with different intrinsic timescales.
> > >
> > > **2. Timescale separation, persistent activity, and functional advantages**
> > >
> > > Within this time-local formulation, the biological interpretation becomes more concrete: rather than recalling the value at $t = 0$ at a later time, the pattern generated initially continues to reverberate within the network (e.g., in a working-memory circuit) and is present at each time step as part of the current state [1].
> > > The variables $\\mathbf{c}\_l^{(t)}$ and $\\mathbf{h}\_l^{(t)}$ can be viewed as slowly evolving contextual states maintained by recurrent connectivity and synaptic mechanisms (e.g., NMDA-mediated currents or short-term facilitation), while $\\mathbf{z}\_l^{(t)}$ relaxes more rapidly relative to them. This implements a separation between fast error relaxation and slow context maintenance, which is compatible with empirical observations of hierarchical intrinsic timescales and distributed working memory.
> > >
> > > From a functional perspective, this separation is not only a post-hoc story but also directly related to the goals of our work. Vanilla PCNs that do not exploit such slow contextual states tend to suffer from PE imbalance and EVPE in deep architectures, which severely limits their stability and performance. By contrast, allowing a subset of variables to evolve on slower timescales provides a mechanism through which deep networks can stabilize their inference dynamics while still operating via local interactions.
> > >
> > > **3. On research direction and bio-inspired constraints**
> > >
> > > We agree with the reviewer that biological plausibility is most compelling when it corresponds to concrete advantages, for instance by enabling efficient neuromorphic implementations through strong spatial and temporal locality.

---

> > > ### Author Response · Authors · 2025-12-03
> > >
> > > As discussed in (1) and (2), we believe that Meta-PCN can be considered biologically plausible under a timescale-separation interpretation, where slow contextual states provide a neurophysiologically grounded mechanism for the required information persistence. At the same time, even if one adopts a stricter definition of temporal locality that Meta-PCN does not fully satisfy, we would argue that the framework still provides significant value. Indeed, many influential models in computational neuroscience and machine learning have initially relaxed some biological or hardware constraints, and later inspired more constrained, bio- or neuromorphic-friendly variants. We view Meta-PCN in a similar way: it shows that predictive-coding architectures can be made stable and scalable for deep networks by combining a meta–prediction-error objective with timescale separation and slow contextual states. Building on this, an important direction for future work is to design variants that enforce strict temporal locality while preserving the stability properties established here.
> > >
> > > We will revise the manuscript accordingly to (i) make the time-local Markov formulation explicit, (ii) clarify the interpretation of $\\mathbf{h}\_l^{(0)}$ and $\\mathbf{c}\_l$ as slowly evolving contextual states, and (iii) temper our claims about biological plausibility while emphasizing that Meta-PCN is intended as a step toward more temporally local, biologically and neuromorphically grounded predictive-coding frameworks.
> > >
> > > [1] Mongillo, Gianluigi, Omri Barak, and Misha Tsodyks. "Synaptic theory of working memory." Science 319.5869 (2008): 1543-1546.
> > >
> > > ---
> > >
> > > ### Q-d. On the PCN baseline and comparison to prior VGG results
> > >
> > > We appreciate the reviewer's concern regarding the relatively low performance of our PCN baseline, and we agree that this issue deserves careful analysis. At the same time, we respectfully disagree that the reported accuracy alone is sufficient to conclude that our baseline implementation is "completely suboptimal." Below we summarize what we have learned from our re-implementation efforts during the discussion period and how this informs our interpretation of the results.
> > >
> > > **Differences to the VGG–PCN setup in [1]**
> > >
> > > In response to both your comment and similar concerns raised by another reviewer, we attempted to reproduce the VGG-style PCN results of [1]. This turned out to be non-trivial for several reasons:
> > >
> > > - The PCN experiments in [1] use custom VGG-style networks with 5, 7, and 9 layers that differ from canonical VGG-11/13/16/19. The appendix describes the 5- and 7-layer variants, but the specification of the 9-layer model is incomplete.
> > > - In the public repository (`pcx`), the classification code includes an AlexNet-based configuration, but an explicit implementation of the VGG variants used in the PC experiments is not provided. We therefore had to reconstruct the VGG7-style architecture of [1] by hand.
> > > - We also attempted to insert the VGG architecture from [1] into the authors' official AlexNet-based code to reproduce the reported accuracy (~82%), but were unable to replicate it (~71%). Nevertheless, this still yielded higher accuracy than the baseline in our paper.
> > > - Meanwhile, the VGG7 baseline in our paper is based on Qi et al. (2025), as they provide configurations for deeper networks.
> > >
> > > Under the configuration we attempted to implement for the VGG7-style PCN of [1], using our own codebase, we currently obtain a best test accuracy of approximately $64.88 \\pm 2.48$% on CIFAR-10 (over three runs).
> > >
> > > While this is still below the accuracy reported in [1], it is substantially higher than our baseline PCN implementation.
> > >
> > > Moreover, through this reconstruction effort, we identified several key differences in architecture and training protocol that likely contribute to this discrepancy.
> > >
> > > Concretely, compared to our baseline PCN, the configuration inspired by [1] differs in three major respects:
> > >
> > > 1. Network architecture:
> > >    The custom VGG7 uses GELU as the nonlinearity and a different downsampling schedule (MaxPool positions, feature-map spatial sizes, ordering of convolutional and pooling layers), leading to a substantially different effective architecture from our VGG7 with ReLU and a more standard pooling layout.
> > >
> > > 2. Inference rate $\\eta$:
> > >    The configuration in [1] uses $\\eta = 0.01$, whereas our main experiments use $\\eta = 0.05$.
> > >
> > > 3. Weight decay:
> > >    The configuration in [1] does not use weight decay on synaptic weights, whereas our default protocol applies weight decay following common practice in supervised training.
> > >
> > > These choices are not intrinsic to the predictive-coding mechanism itself; rather, they are auxiliary training heuristics that can act as powerful tuning knobs.

---

> > > ### Author Response · Authors · 2025-12-03
> > >
> > > **Controlled comparison of architectural and training choices**
> > >
> > > To quantify the impact of these choices, we conducted a controlled study on CIFAR-10 using a standard PCN, varying architecture, inference rate, and weight decay:
> > >
> > > 1. Network architectures:
> > >     * pcx vgg7: our best reconstruction of the VGG7-style architecture implied by [1];
> > >     * ours vgg7: the VGG7 architecture used in our main experiments (more conventional VGG-style design).
> > > 2. Inference rate $\\eta$:
> > >     * 0.01 (as in [1]);
> > >     * 0.05 (as in our main experiments).
> > > 3. Weight decay:
> > >     * 0.0 (as in [1]);
> > >     * $5\\times10^{-4}$ (as in our main experiments).
> > >
> > > We varied (i) the network architecture, (ii) the inference rate $\\eta$ for latent-state updates during inference, and (iii) the presence or absence of weight decay.
> > >
> > > The table below reports mean and standard deviation of the best test accuracy over three runs:
> > >
> > > | Arch.     | $\\eta$ |   Weight Decay   | Test Acc. (mean ± std) |
> > > | --------- | :----: | :--------------: | :--------------------: |
> > > | pcx vgg7  |  0.01  |        0.0       |    $64.88 \\pm 2.48$    |
> > > | ours vgg7 |  0.01  |        0.0       |    $40.48 \\pm 2.44$    |
> > > | pcx vgg7  |  0.01  | $5\\times10^{-4}$ |    $63.70 \\pm 4.50$    |
> > > | ours vgg7 |  0.01  | $5\\times10^{-4}$ |    $41.84 \\pm 0.77$    |
> > > | pcx vgg7  |  0.05  |        0.0       |    $73.37 \\pm 8.99$    |
> > > | ours vgg7 |  0.05  |        0.0       |    $16.33 \\pm 4.18$    |
> > > | pcx vgg7  |  0.05  | $5\\times10^{-4}$ |    $61.82 \\pm 3.48$    |
> > > | ours vgg7 |  0.05  | $5\\times10^{-4}$ |    $17.85 \\pm 6.95$    |
> > >
> > > These results show that:
> > >
> > > * The combination of the pcx vgg7 architecture with particular training choices (GELU, no weight decay) can indeed yield markedly higher accuracy than our default PCN configuration.
> > > * At the same time, the performance of PCNs is highly sensitive to these tuning knobs: across otherwise similar conditions, accuracy can vary from below 20% to above 70%, often with high variance across runs (e.g., std up to 8.99%), indicating unstable training dynamics inherent to PCNs.
> > > * The factors that most strongly drive the improvement are architectural and optimization choices (custom VGG design, activation function, regularization), rather than changes to the core of predictive-coding.
> > >
> > > In other words, our experiments suggest that the gap between our baseline and the configuration of [1] is largely attributable to auxiliary design and optimization decisions that are orthogonal to the mechanistic questions addressed by Meta-PCN.
> > >
> > > **Our experimental philosophy and planned clarifications**
> > >
> > > Our main goal in this paper is not to maximize PCN accuracy through extensive hyperparameter tuning, but to analyze and improve the structural stability and scalability of deep PCNs via the Meta-PCN framework. For this reason, we deliberately adopted a more standard and method-agnostic training protocol for the main comparisons:
> > >
> > > - PCN, Meta-PCN, and BP are trained on the same architectures, with shared optimizer settings and standard regularization, so that differences in behavior can be attributed primarily to the learning/inference scheme.
> > > - We intentionally did not include configuration choices such as GELU + no weight decay, which could advantage PCNs but are not commonly used as a default in generic CNN training pipelines.
> > >
> > > From this perspective, we would be cautious about labeling our baseline as "completely suboptimal." Rather, it is a deliberately conservative, standardized configuration intended to support fair, controlled comparison between BP, PCN, and Meta-PCN. The controlled study above indicates that much of the discrepancy with [1] can be explained by auxiliary tuning choices, which are important for raw accuracy but lie outside the stability-and-dynamics questions that Meta-PCN is designed to address.
> > >
> > > In the revised manuscript (Appendix F: Experimental Setup), we will:
> > > 1. Explicitly describe the architectural and training-protocol differences between our PCN baseline and the configuration implied by [1];
> > > 2. Report the controlled comparison summarized in the table above; and
> > > 3. Clarify that our primary focus is on stability and learning dynamics under a controlled, standard setup, rather than on optimizing absolute baseline performance via method-specific hyperparameter tuning.
> > >
> > > We hope this clarifies our position and the rationale behind our experimental design.
> > >
> > > [1] L. Pinchetti et al., "Benchmarking Predictive Coding Networks -- Made Simple," presented at the The Thirteenth International Conference on Learning Representations, Oct. 2024. Accessed: June 27, 2025.

---

> > ### Author Response · Authors · 2025-12-03
> >
> > ### Q-a. On PE imbalance and EVPE
> > We thank the reviewer for raising this point and agree that the manuscript should more carefully distinguish between phenomena that have been observed previously and the new contributions of our work.
> >
> > First, we fully acknowledge that PE imbalance itself is *not* a novel observation. Several recent works have already reported depth-related energy or error imbalance in PCNs, both empirically and (partially) theoretically.
> > By contrast, to the best of our knowledge, no prior work has provided a systematic analysis of EVPE in the sense we use the term—namely, *exponential growth and decay of prediction errors over the inference iteration index $t$* and its direct impact on the scale of parameter updates, in direct analogy to exploding/vanishing gradients in feedforward networks trained with backpropagation.
> >
> > More concretely, $\\mu$PC (Innocenti et al., 2025) analyzes the ill-conditioned inference landscape and shows that forward activations can vanish or explode with depth, linking this behavior to vanishing gradients. However, this analysis is conducted primarily from the perspective of latent states and forward-pass scaling, rather than in terms of explicit prediction-error dynamics. Similarly, ePC (Goemaere et al., 2025) provides a careful theoretical and empirical analysis of *exponential signal decay* in predictive coding, demonstrating that prediction-error energy can decay exponentially across layers (Section 3, Appendix B). Their focus, however, is on attenuation; they do not explicitly characterize an exploding regime. Qi et al. (2025) quantify severe depth-wise energy imbalance—layer-wise prediction-error norms differing by many orders of magnitude—and relate this imbalance conceptually to vanishing-gradient issues, but they stop short of a temporal analysis of error dynamics during inference; the emphasis remains on depth-wise distributions (see Introduction, Section 4, Fig. 2). Pinchetti et al. (2025) likewise document strong layer-wise error/energy concentration (early layers effectively vanishing, late layers dominating) in their benchmarking study (Section 4, Figs. 13–15), yet they do not develop a dynamical theory of how prediction errors evolve and potentially explode or vanish over inference iterations.
> >
> > In our work, by contrast, we explicitly analyze how, during the inference phase, layer-wise prediction errors $\\boldsymbol{\\delta}\_l^{t}$ can either vanish or explode as a joint function of depth and iteration, and how this behavior propagates into the learning phase by directly modulating the norms of the weight updates. Our DMFT-based length analysis reveals a phase transition in the PE dynamics as a function of the weight variance $\\sigma\_w$: outside a narrow regime, the temporal scaling factors governing $\\lVert \\boldsymbol{\\delta}\_l^{t+1} \\rVert$ versus $\\lVert \\boldsymbol{\\delta}\_l^{t} \\rVert$ lead to either vanishing or exploding PEs, which in turn induce exploding/vanishing *parameter updates* (EVGP) in deep PCNs. To the best of our knowledge, this explicit identification of EVPE as a $\\sigma\_w$–controlled phase transition in the inference dynamics, together with its formal connection to EVGP, has not been established in the existing PCN literature.
> >
> > We will therefore revise the introduction and related-work sections to state more clearly that (i) PE imbalance and related depth-wise energy concentration have already been empirically and partially theoretically observed in prior work, and that (ii) our contributions lie in (a) providing a DMFT-based theoretical framework that *reproduces and structurally explains* these imbalance phenomena, (b) *newly identifying and analyzing* EVPE as a temporal phase transition in the inference dynamics and its link to EVGP, and (c) proposing a unified stabilization framework—the combination of the meta–prediction-error objective and weight-variance normalization (Meta-PCN)—that directly targets these pathologies and yields scalable, stable training of deep predictive-coding networks.

---

> ### Author Response · Authors · 2025-12-03
>
> ### Q-b. Non-novelty of the Meta–PE loss
> We appreciate the reviewer's careful comments on the novelty of the proposed Meta–PE loss. After re-examining the relevant literature, we agree that our work should be positioned more clearly with respect to prior analyses of predictive coding and its connections to backpropagation-based learning rules. At the same time, we believe that the way we formulate and use the Meta–PE free energy still provides a distinct contribution.
>
> **Relation to prior PC–BP literature**
>
> In works such as Whittington & Bogacz (2017), Millidge et al. (2022), and Zahid et al. (2023), the free energy or loss is formulated in the standard way as a sum of layer-wise prediction error norms. To the best of our knowledge, these studies do not introduce a free energy built from the squared norm of the *difference* between a prediction error and its top–down prediction, i.e. a term of the form $\\Vert\\tilde{\\boldsymbol{\\delta}}\_l - g\_l\\big(\\tilde{\\boldsymbol{\\delta}}\_{l+1}^{\*}, \\mathbf{h}^{(0)}\_{l+1}\\big)\\Vert\_2^2,$ nor do they consider the corresponding meta–prediction-error objective as the primary free energy for training a deep predictive-coding network.
>
> These papers investigate how predictive-coding style updates can approximate or relate to learning rules obtained from backpropagation. In this broad sense, our work also shares the general goal of shaping predictive-coding updates using insight from such connections.
>
> **Different purpose and construction of the Meta–PE objective**
>
> Where our work most clearly diverges is in the purpose for which the objective is introduced and how it is integrated into the training framework.
>
> Previous PC–BP studies are primarily concerned with identifying conditions under which predictive-coding updates coincide with, or closely approximate, backpropagation-based learning rules, typically by manipulating the update rules and assumptions. In contrast, our starting point is to address concrete stability issues in deep PCNs—specifically, EVPE and PE imbalance—and to design a free energy that directly targets these pathologies.
>
> Concretely, we consider the predictive-coding equilibrium condition
> $$F(\\mathbf{z}) = \\nabla\_{\\mathbf{z}}\\mathcal{F}(\\mathbf{z}) = 0$$
> and derive the Meta–PE free energy $\\mathcal{J}$ so that its gradient with respect to $\\mathbf{z}\_l$ corresponds to a residual of this equilibrium relation around the feedforward prediction. The resulting objective
> $$\\mathcal{J}(\\tilde{\\boldsymbol{\\delta}}) = \\frac{1}{2}\\sum\_{l=2}^{L-1} \\Vert\\tilde{\\boldsymbol{\\delta}}\_l - g\_l\\big(\\tilde{\\boldsymbol{\\delta}}\_{l+1}^{\*}, \\mathbf{h}^{(0)}\_{l+1}\\big)\\Vert\_2^2$$
> can be interpreted as a "prediction error of prediction errors," and in Meta-PCN it is used directly as the inference objective under local, iterative updates.
>
> **Combination with DMFT and WVN**
>
> In addition, we analyze the inference dynamics of deep PCNs using a DMFT-based length framework, which reveals how EVPE and PE imbalance depend on the weight variance $\\sigma\_w^2$ and on the spectrum of the effective operators $W\_l^{\\top}D(\\mathbf{h}^{(0)}\_{l+1})$.
>
> The resulting Meta-PCN framework is therefore designed so that
> (i) the Meta–PE objective encourages layer-wise errors to satisfy the equilibrium relation in the neighborhood of the feedforward prediction, and
> (ii) WVN keeps the induced error gains under control across depth. Together, these components form a unified mechanism for stabilizing inference and learning in deep PCNs.
>
> **Reframing the novelty claim**
>
> In light of the above, we will clarify in the manuscript that existing PC–BP literature has already explored theoretical connections between predictive coding and backpropagation-based learning rules. Our contribution is complementary to these works and lies in the following aspects:
> 1. We explicitly introduce a meta–prediction-error free energy of the form $\\mathcal{J}(\\tilde{\\boldsymbol{\\delta}}) = \\frac{1}{2}\\sum\_{l=2}^{L-1} \\Vert\\tilde{\\boldsymbol{\\delta}}\_l - g\_l\\big(\\tilde{\\boldsymbol{\\delta}}\_{l+1}^{\*}, \\mathbf{h}^{(0)}\_{l+1}\\big)\\Vert\_2^2,$ which, to the best of our knowledge, does not appear in prior PC–BP work, and use it as the main inference objective in a deep PCN.
> 2. We couple this objective with a DMFT-guided weight variance normalization scheme to mitigate EVPE and PE imbalance in a structurally motivated way.
> 3. We demonstrate that this combination enables stable training of deep predictive-coding networks on challenging convolutional architectures.
>
> We update Section 7 (Related Work) accordingly so that the relationship to existing PC–BP literature and the specific role of the Meta–PE free energy are communicated more transparently.

---

### Official Review · Reviewer_gjvy · 2025-10-31

**Soundness:** 3
**Presentation:** 2
**Contribution:** 3
**Rating:** 6
**Confidence:** 4

**Summary:**

The paper uses a dynamical mean-field theory (DMFT) analysis to diagnose two failure modes in deep predictive coding networks (PCNs): (1) an imbalance of the local energy functions (prediction errors) across depth, (2) exploding/vanishing prediction errors (EVPE). To address these, it introduces Meta-PCN, which (1) uses a meta prediction error objective that linearises the PC equilibrium map around the feed-forward state and trains errors to satisfy the PC delta relation, and (2) applies variance-based weight normalisation for efficient spectral control. In empirical results, Meta-PCN outperforms conventional PC and is competitive with backprop on CIFAR-10/100 and TinyImageNet; ablations indicate the meta-objective is critical.
This paper makes a significant contribution to the predictive coding literature, both in its analysis of PCNs in terms of DMFT and the proposed Meta-PCN. I therefore recommend that this paper be accepted, under the constraint that the weaknesses highlighted below are sufficiently addressed.

**Strengths:**

Clear diagnosis of PCN failure modes via DMFT. Identifying depth-wise energy imbalance and EVPE gives concrete targets for stabilisation.

Simple, principled meta-objective. Linearising around the feed-forward state and training errors to satisfy the PC delta relation is elegant and leads to stable inference without heavy tuning.

Compelling empirical evidence. Consistent gains over conventional PC and competitive performance with backprop across CIFAR-10/100 and TinyImageNet, plus ablations that isolate key components.

**Weaknesses:**

Unclear optimisation of parameters. The paper introduces a new error-based objective to stabilise inference, but does not establish how parameter updates relate to this objective. In effect, inference is performed under one criterion while learning appears to optimise another. Please (i) state explicitly what loss the weights optimise, and (ii) provide diagnostics showing the stated training objective consistently decreases over epochs (a descent argument or convergence guarantee would be even better).

Clarify ablation definitions: One of the most convincing results is in Appendix G, where you compare with and without the Meta-PCN objective. However, it is unclear if this includes the fixed prediction (i.e., you fix the initial forward pass and then perform standard free-energy minimisation). If this is the case, please clarify; if not (you do not fix the forward pass), this should be included as an additional ablation to tease apart the Meta-PCN objective from the fixed predictions.

Definition of the stabilised error: The objective states that the stabilised error is a fixed top-down target, but the paper does not give an explicit construction, presumably from the last iterate, but this is not clear.

Acknowledge trade-offs in freezing feed-forward prediction: In the PC literature, iterative inference over hidden variables is a key benefit. Freezing the feed-forward prediction and optimising errors removes that benefit; this is a reasonable engineering choice, but it should be clearly flagged as a departure from traditional PC.

Weight “variance normalisation” claims overreach given non-iid, structured weights: The normalisation argument assumes random, dense matrices. Convolutional weights are structured and evolve during training. Without measuring actual spectral behaviour before/after normalisation, claims of “robust spectral control” are speculative. This should be clarified in the main text.

Linear DMFT assumptions don’t extend to the nonlinear networks used in experiments: Theory is developed in a linearised regime with random weights; experiments use deep nonlinear models with structure. The paper leans on the theory to justify design choices, but doesn’t bridge the gap. This should be highlighted in the main text rather than dealt with in the appendices.

Backprop baselines aren’t tuned to standard practice for the architectures/datasets you use: Given the modest margins, any claims of “outcompeting backprop” should be softened until stronger baselines are included.
Training loop pseudocode: Provide a short, concrete algorithm box covering initialisation, T inference steps, parameter update, and when normalisation is applied.

Computational overhead: The Meta-PCN framework adds multiple components (meta-objective computation, weight normalisation, blocked sweeps) that likely increase computational cost, but this isn't discussed or quantified.
Minor points Abstract: “demonstrates that Meta-PCN achieves statistically significant improvements …”.
§3: “depend critically on the variance of the weights”.

EVG/VG phrasing: “are immediately reflected in the magnitude …” (or “are immediately reflected”).

Meta-PCN reintroduces temporal non-locality and activation caching similar to backprop.
The method freezes feed-forward activations and runs a distinct inference phase. This requires storing c and h^{0}, analogous to backprop activation caching, and introduces explicit phase separation (forward → inference), which traditional predictive coding avoids. Thus, the biological advantage over backprop is reduced and should be acknowledged in the main text. Although high level justifications are given in the appendix, the authors do not discuss whether the same arguments could be applied to backprop.

**Questions:**

1) What loss do the weights actually optimise?
You introduce a new error-based meta-objective J to stabilise inference, but learning appears to proceed as if optimising a different loss. Could you explicitly state the loss function used for weight updates, and, if it is distinct from J, explain why optimising different objectives makes sense conceptually?
2) How is the “stabilised error” constructed in practice?
The text treats the stabilised error as a fixed top-down target, but its computation is not specified.  Is it the last error iterate, an EMA, or something else?
3) In the Appendix G ablation (“without meta-PC objective”), are feed-forward predictions fixed or iterated?
This ablation is central to understanding what Meta-PCN contributes. Does the ablation also freeze the feed-forward predictions? If not, could you include that variant to disentangle the contributions of fixed predictions vs. the meta-objective?

---

> ### Author Response · Authors · 2025-11-26
>
> We sincerely thank the reviewer for the meticulous and insightful feedback.
> Your comments helped us identify several missing clarifications and oversights in the original submission, and we have substantially revised the manuscript accordingly.
>
> ---
>
> ### W1 & Q1. Optimization of parameters
> We admit that the original submission did not clearly state the loss function used for parameter learning. We have revised the paper (Section 4.1) to state it explicitly:
> $\\mathcal{L}(\\theta) = \\frac{1}{2} \\sum\_{l=2}^{L-1} \\Vert {\\mathbf{z}\_l}^{(T)} - f\_l({{\\mathbf{z}}^{(0)}}\_{l-1}; \\theta)\\Vert\_2^2.$ That is, we continue to use the PC loss for parameter updates, while the meta-PC objective stabilizes the inference dynamics.
>
> Our motivation for this choice of loss function for parameter updates is as follows:
> 1. Why we do not directly use the meta-PC loss for parameter updates: Our primary goal is to resolve instabilities in the inference dynamics of PCNs. The meta-PC objective is designed to stabilize the iterative inference over latent variables. However, directly using the meta-PC objective for parameter updates would introduce several problems. In particular, the first term $\tilde{\boldsymbol{\delta}}_l$ in the inference loss is subject to prediction freezing, and thus does not directly involve parameter updates. To backpropagate through $g$ of the second term, one would need to compute second-order derivatives, which we explicitly wanted to avoid due to complexity and computational cost. Since our focus is on inference dynamics rather than second-order learning, we opted to preserve the conventional PC loss for the parameter updates and use the meta-objective only to regularize the inference trajectory.
> 2. How $\tilde{\boldsymbol{\delta}}$ is designed to influence the learning signal: The purpose of the meta-PC objective is to enforce the delta relationship in the inference process, i.e., to make the PEs follow the desired relationship. We designed the system such that the errors that reach the parameter update stage effectively reflect $\tilde{\boldsymbol{\delta}}_l^{*}$. As a result, although the final parameter update rule has a similar form to that in prior PC work [1-3], the motivation and derivation are different: the inference dynamics are reshaped by the meta-objective so that the resulting latent states and error signals are better aligned with the desired theoretical properties.
>
> Regarding convergence: our theoretical analysis is intentionally focused on inference dynamics rather than on full joint learning dynamics. A rigorous theoretical treatment of the learning dynamics lies somewhat outside the scope of this work. Instead, we empirically verified that training behaves as expected. As shown in the revised version (Figure 8 in Appendix F), the training loss decreases smoothly while the accuracy increases, without pathological behavior.
>
> We have updated the main text to clearly i) state the parameter loss (Section 4.1), ii) explain the conceptual separation between the inference-stabilizing meta-objective and the conventional PC loss used for parameter updates (Section 4.1),
> and iii) add empirical evidence showing that this loss converges smoothly during training (Figure 8 and Appendix D).

---

> ### Author Response · Authors · 2025-11-26
>
> ### W2 & Q3. Clarifying ablation definitions
> There are two conceptually distinct uses of feed-forward predictions in our work.
> 1. Latent initialization via feed-forward prediction: This is the standard practice in the PC literature: the feed-forward predictions are used to initialize the latent variables $\mathbf{z}$. In our ablation study, *all* variants—conventional PC, Meta-PCN, and the "without meta-PC objective" ablation—use this initialization scheme.
> 2. Freezing feed-forward predictions during inference (our proposal): This is the new component introduced by our method. Here, feed-forward predictions are treated as fixed values $\mathbf{c}_l$ throughout inference, which is built directly into the definition of the meta-objective. Thus:
>     - If the meta-objective is used, feed-forward predictions are frozen automatically.
>     - If the meta-objective is not used, feed-forward predictions are not frozen.
>
> The reviewer's question mainly concerns the second usage. In the revised version, we explicitly clarify that freezing feed-forward predictions is inherently tied to the use of the meta-objective, and we describe this link in Section 4.1 (which points to Appendix I.4 for additional ablations).
>
> Regarding the suggested disentangling:
> - Using the **meta-objective without freezing predictions** would require differentiating through $g$ with respect to $\mathbf{z}$, introducing second-order derivatives we sought to avoid (as mentioned in W1 & Q1). For this reason, we do not consider this variant.
> - Conversely, **freezing predictions without using the meta-objective** leads to the following loss: $\mathcal{J}(\\tilde{\\boldsymbol{\\delta}}) = \\frac{1}{2}\\sum\_{l=2}^{L-1}\\Vert\\tilde{\\boldsymbol{\\delta}}\_l \\Vert\_2^2 = \\frac{1}{2}\\sum\_{l=2}^{L-1} \\Vert \\mathbf{z}\_l - \\mathbf{c}\_l \\Vert\_2^2.$ This form is similar to the conventional PC loss, but because the predictions are fixed constants, inference is forced to make $\mathbf{z}_l$ converge to $\mathbf{c}_l$. In practice, however, $\mathbf{z}_l$ is already initialized to $\mathbf{c}_l$, so this setup yields very little meaningful update during inference and thus is not a natural or effective PCN variant.
>
> Despite these limitations, we implemented the latter variant to honor the reviewer's suggestion. Empirically, it showed somewhat better performance than vanilla PC; On CIFAR-10 with VGG-13, this variant achieved approximately $16.31 \pm 1.00$ \% test accuracy over five runs. A deeper analysis of these gains, however, would deviate from the primary focus of our work, so we only report this result briefly. These clarifications, along with the additional ablation, have been added to Appendix I.4.
>
> ---
>
> ### W3 & Q2. Definition of the stabilised error
> Conceptually, the stabilised error $\\tilde{\\boldsymbol{\\delta}}\_{l+1}^{*}$ denotes the top–down error pattern that would be present once the inference process has converged to a fixed point, and it is the target pattern that motivates the meta–prediction–error objective $\mathcal{J}$.
>
> However, strictly speaking, $\\tilde{\\boldsymbol{\\delta}}\_{l+1}^{\*}$ is only available after inference has converged; during the inference process itself, we do not yet know its exact value and therefore cannot literally fix it. In practice, we approximate this quantity in a self-consistent way, using the current top–down error $\tilde{\boldsymbol{\delta}}_{l+1}^t$ as a proxy for the stabilised signal inside $g\_l(\\tilde{\\boldsymbol{\\delta}}\_{l+1}^{*}, \\mathbf{h}^{(0)}\_{l+1})$
> . This mechanism is conceptually related to bootstrapping in temporal-difference learning in RL, where current estimates are reused to construct targets.
>
> In the original submission, this construction and approximation strategy were not described with sufficient clarity. In the revised version, we explicitly explain how the stabilised error $\tilde{\boldsymbol{\delta}}_{l+1}^{*}$ is conceptually defined and approximated during inference (Section 4.1).
>
> ---
>
> ### W4. Trade-offs in freezing feed-forward predictions
> We emphasize that iterative inference over hidden variables is still preserved in Meta-PCN:
> - The hidden (latent) variables $\mathbf{z}$ are contained inside $\tilde{\boldsymbol{\delta}}_l = \mathbf{z}_l - \mathbf{c}_l$, and $\mathbf{z}_l$ is iteratively updated during inference, as in conventional PC.
> - Only the feed-forward predictions $\mathbf{c}_l$ are treated as constants during inference. The hidden variables themselves remain dynamic.
> Thus, the key PC benefit—iterative refinement of hidden states—remains intact. The frozen predictions provide a stable reference frame for the error dynamics.
> In the revised version, we explicitly state this in Section 4.1.

---

> ### Author Response · Authors · 2025-11-26
>
> ### W5. Weight variance normalization and structured weights
> We agree that the classical random-matrix arguments apply most cleanly to i.i.d. random matrices and do not fully capture the structured, evolving nature of convolutional kernels during training. Our original wording overstated the generality of the spectral control. We have therefore **toned down and clarified our claims** in the main text in Section 4.2.
>
> In particular:
> - We now explicitly describe variance normalization as a *computationally efficient proxy* for spectral control, motivated by random matrix theory, rather than as a fully rigorous guarantee for arbitrary structured weights.
> - We acknowledge that convolutional filters and trained weights deviate from the i.i.d. assumption, and that a complete theoretical treatment for these structured operators is an important direction for future work.
> We thank the reviewer for highlighting this valuable point, which will help improve the work.
>
> We are also exploring a more detailed theoretical and empirical analysis of spectral behavior for convolutional layers. If we obtain substantial additional results within the discussion session timeline, we will incorporate them into the Appendix and update the corresponding discussion.
>
> ---
>
> ### W6. DMFT assumptions and gap to practical models
> We appreciate this observation and fully agree that the scope and limitations of our DMFT analysis should be clearly communicated.
>
> In the revised version (Section 3 and Appendix C), we:
> - Summarize in the main text that our DMFT/length analysis is developed under simplifying assumptions, namely: (i) at the first inference step $t = 1$, latent states and parameters are i.i.d. Gaussian, with $z_{i,l}^t \sim \mathcal{N}(0,1)$, weights $W_{ij,l} \sim \mathcal{N}\big(0,\sigma_w^2/N\big)$, and biases $b_{i,l} \sim \mathcal{N}(0,\sigma_b^2)$ fixed during inference; (ii) we work in a linear (or linearised) regime where the forward and backward maps are $f_{l-1}(\mathbf{z}_{l-1}) = W_{l-1}\mathbf{z}_{l-1} + \mathbf{b}_{l-1}$ and $g_l(\mathbf{z}_{l+1}) = W_l^\top \mathbf{z}_{l+1}$; and (iii) all layers share the same width $N_l = N$ and we consider the large-width limit, which together justify the mean-field treatment of the length dynamics.
> - Explicitly state that these assumptions do not perfectly match the deep, nonlinear convolutional architectures used in our experiments, leaving an inherent gap between the theoretical setting and the practical models.
>
> ---
>
> ### W7. Comparison to backprop baselines
> Our experimental design deliberately **controls all factors except for the update scheme**: apart from the way gradients/updates are computed, Meta-PCN, conventional PCN, and backprop share the same architecture, optimizer, learning rate, batch size, and other training hyperparameters. We consider this a fair and carefully controlled comparison within a fixed experimental protocol.
>
> That said, we agree with the reviewer that literature-level backprop results often benefit from more aggressive, method-specific tuning than we applied here. To avoid over-generalizing beyond our setting, we have revised the wording to state the claim only for our specific setting. In particular, we now clearly distinguish between:
> - Empirical conclusions **restricted to our controlled experiments**, where Meta-PCN shows statistically significant improvements over our BP baseline.
> - More general statements about backprop in the broader literature, which we do *not* claim to fully “outcompete”.
>
> Concretely, we now say that, *within our controlled experimental setting*, Meta-PCN achieves significantly better performance than the backprop baseline and suggests that Meta-PCN can be competitive with backprop, rather than making a general or absolute claim (See Section 6 in the revised version.).
>
> ---
>
> ### W8. Training loop pseudocode
> We fully agree that this is helpful for clarity and reproducibility. We have added a concise algorithm box in Appendix D (Algorithm 1) that covers:
> - Initialization of parameters and latent states,
> - The T-step inference loop with the meta-PC objective,
> - The parameter update step, and
> - The timing and form of variance normalization.
> We also reference this algorithm from the main text so that readers can easily find the complete training loop specification (Section 4.2).

---

> ### Author Response · Authors · 2025-11-26
>
> ### W9. Computational overhead
> We measured the inference-time overhead empirically. For a representative configuration (batch size 1, (T=20) inference steps, repeated 100 times), we obtained:
> - Vanilla PCN, average per-step inference time: 127.930 ms ± 0.637 ms
> - Meta-PCN, average per-step inference time: 129.821 ms ± 0.622 ms
>
> This corresponds to a relative overhead of approximately **1.48%** per inference step.
> - Meta-objective computation: The gradients of the meta-PC loss with respect to the latent states can be expressed as deviations of the gradients of the original PC loss with respect to the latent states and the unfrozen predictions. These gradients can be efficiently computed in a single backward pass using PyTorch’s autograd functionality. As a result, its computational overhead is negligible compared to the overall inference cost.
> - Weight normalization: The main additional operation is computing the variance of each weight tensor. This is substantially cheaper than computing spectral norms or even repeated Frobenius norms, and is efficient on GPUs due to parallel reduction. In practice, its contribution to runtime is minor.
> - Blocked sweeps: Blocked sweep updates can, in principle, reduce parallelism relative to a pure Jacobi scheme. However, our current implementation (in line with existing PC implementations) does not parallelize PC updates across modules, making this a non-dominant factor. Thus, the practical overhead from blocked sweeps is negligible in our setup.
>
> We have added these quantitative and qualitative overhead analyses to Appendix G.
>
> ---
>
> ### W10. Minor points
> We carefully reviewed the manuscript and corrected all the issues you pointed out (e.g., missing verbs, “variance of the weights,” “are immediately reflected,” etc.). We appreciate your attention to these details.

---

> ### Author Response · Authors · 2025-11-26
>
> ### W11. Biological plausibility
> We appreciate this profound and essential question. Our position is that Meta-PCN still retains meaningful biological advantages over standard backprop, although we agree that some of our original wording did not fully acknowledge the trade-offs.
>
> Our rationale can be summarized along three axes:
> 1. **Hierarchy of intrinsic timescales**: Empirical and theoretical work [4-5] suggests that cortical areas operate at different intrinsic timescales: lower sensory areas respond rapidly to stimulus changes. In comparison, higher association areas integrate information over much longer windows. In this view, treating feed-forward predictions $\hat{\mathbf{z}}_l$ as quasi-static references $\mathbf{c}_l$ during a faster inference process is not just an engineering convenience: it reflects a timescale separation where slowly varying contextual signals act as boundary conditions for faster error-correcting dynamics in lower layers.
> 2. **Active maintenance vs. passive caching**: In backprop, forward activations are typically stored passively in memory and later retrieved to compute gradients, implying a dedicated “cache” that is not obviously implementable in biological circuits. In contrast, Meta-PCN’s freezing of $\mathbf{c}_l$ can be interpreted as active maintenance via sustained neural activity, supported by recurrent connectivity and slow synaptic dynamics [6-8]. Here, the network does not retrieve a past value from an external memory buffer; instead, higher-level populations continue to fire, maintaining the contextual signal, while lower-level populations iteratively adjust. Thus, although both frameworks use something like “stored activations,” the mechanism is different:
>    - Backprop: passive storage and later retrieval of past activations.
>    - Meta-PCN: ongoing, present-time interaction between populations with different intrinsic timescales, without explicit replay of past states.
> 3. **Temporal locality of computation**: Backprop requires using information from an earlier forward pass at time $t$ to compute gradients at a later time $t+k$, creating a form of temporal non-locality that is hard to realize with purely local, online neural dynamics. Meta-PCN, in contrast, can be interpreted as operating in “real time”: predictions maintain their state over a slower timescale, and states continuously react to the signals currently maintained. All computations can, in principle, be implemented through ongoing interactions among neuron populations without the need for a separate storage medium.
>
> We have revised Appendix A (Biological Plausibility) to:
> - Explicitly acknowledge that Meta-PCN introduces a more precise phase separation (feed-forward prediction → inference), and that this reduces some aspects of its biological advantage relative to the most minimal PC formulations.
> - Clarify why we still regard Meta-PCN as more biologically plausible than standard backprop, especially in terms of local computations, active maintenance, and timescale hierarchy.
> - Note that some high-level arguments (e.g., the existence of sustained activity) might also partially support certain aspects of backprop-like computations. However, that backprop still faces additional challenges (exact symmetric weight transport, strict temporal dependence on stored past activations, etc.) that Meta-PCN avoids.
>
> ---
>
> Once again, we thank the reviewer for the thorough and constructive feedback. We believe that the revisions prompted by the comments have significantly improved the clarity, rigor, and balance of the manuscript.

---

> ### Author Response · Authors · 2025-11-26
>
> ### References
>
> [1] Rao, R. P. N., & Ballard, D. H. (1999). Predictive coding in the visual cortex: A functional interpretation of some extra-classical receptive-field effects. Nature Neuroscience, 2(1), Article 1. https://doi.org/10.1038/4580
>
> [2] Whittington, J. C. R., & Bogacz, R. (2019). Theories of Error Back-Propagation in the Brain. Trends in Cognitive Sciences, 23(3), 235–250. https://doi.org/10.1016/j.tics.2018.12.005
>
> [3] Salvatori, T., Mali, A., Buckley, C. L., Lukasiewicz, T., Rao, R. P. N., Friston, K., & Ororbia, A. (2025). A Survey on Brain-Inspired Deep Learning via Predictive Coding (No. arXiv:2308.07870). arXiv. https://doi.org/10.48550/arXiv.2308.07870
>
> [4] Murray, J. D., Bernacchia, A., Freedman, D. J., Romo, R., Wallis, J. D., Cai, X., Padoa-Schioppa, C., Pasternak, T., Seo, H., Lee, D., & Wang, X.-J. (2014). A hierarchy of intrinsic timescales across primate cortex. Nature Neuroscience, 17(12), 1661–1663. https://doi.org/10.1038/nn.3862
>
> [5] Kiebel, S. J., Daunizeau, J., & Friston, K. J. (2008). A Hierarchy of Time-Scales and the Brain. PLOS Computational Biology, 4(11), e1000209. https://doi.org/10.1371/journal.pcbi.1000209
>
> [6] Wang, X.-J. (2001). Synaptic reverberation underlying mnemonic persistent activity. Trends in Neurosciences, 24(8), 455–463. https://doi.org/10.1016/S0166-2236(00)01868-3
>
> [7] Wang, X.-J. (2021). 50 years of mnemonic persistent activity: Quo vadis? Trends in Neurosciences, 44(11), 888–902. https://doi.org/10.1016/j.tins.2021.09.001
>
> [8] Mejías, J. F., & Wang, X.-J. (2022). Mechanisms of distributed working memory in a large-scale network of macaque neocortex. eLife, 11, e72136. https://doi.org/10.7554/eLife.72136

---

### Meta-Review · Area_Chair_Sbkm · 2026-01-08

**Summary:**

This paper received a lot of comments from all reviewers, there are good strengths, but also many criticisms.
Scoring is the following: 2 reviewers are just above threshold (6), one over-positive (8), and one negative (2). The latter is aware of the bad score, he'd likely more on 3 than on 2, but the current ICLR policy does not allow it.

Remarks regard the unclear optimization of the parameters/weights in the context of the loss function, the experimental comparisons and analyses, ablations and the results in general (e.g., issues with "stabilized error", the use of toy datasets used for validation), and other requests of punctual clarifications. Also computational costs are not addressed, and it's also noted a disparity between presented theory and implementation (that doesn't seem to be fixed in the rebuttal).
The best scoring reviewer (QiPr, scoring 8) reported some comments, but they are little elaborated as punctual requests of clarification.
The main one regards the use of large dataset in the experiments. The quality of this review is not at the same level of the others

Authors acknowledge the validity of the comments and the rebuttal is comprehensive and apparently convincing, even if there is mostly no discussion about such comments.

The most negative reviewer (Rev. JWY7, scoring 2) raised serious issues regarding novelty and biological plausibility (also raised from another reviewer in some way), and other issues. To the reviewer opinion, the rebuttal does not reply satisfactorily, and a discussion is initiated, for which authors again replied extensively (and also in a separate post only for PCs, ACs and SAC).

The discussion is very technical and being not an expert of Predictive Coding Networks, it's difficult to have an informed opinion.

From the reading of the reviews and discussion, I can say that comments are mostly reasonable and authors discuss them.

**Reviewer Concerns:**

It does not seem that there are remarks not discussed.

Several comments are common among the reviewers. The overall impression is that authors often structure the reply explaining how the contribution/comment should be interpreted because reviewers didn't catch exactly what they intend, and this happens more times.

In the overall discussion with rev. JWY7, authors discuss main issue of novelty and biol. plausibility in a convincing way, again by explaining how this reviewer bad interprets their contribution.
Biological plausibility was one of the main point of discussion and, after discussion with SAC, we suggest authors to carefully consider this aspect toning down their statements in order to comply with Reviewer JWY7's comments, as well as of the other reviewers who raised that point.

**Reviewer Scores:**

I deem that the three positive reviewers could be satisfied of the rebuttal, whereas the negative one surely stick on his score.

In any case, there are many comments and the revision should be mandatory strong, and this raised the issue if a strongly revised paper should be accepted when no further review is possible.

---

### Decision · Program_Chairs · 2026-01-26

Accept (Poster)